# From Coefficients to Directions: Rethinking Model Merging with Directional Alignment

## Abstract

Model merging is a recently emerging paradigm that aims to integrate multiple independently trained models into a single model without joint retraining. Previous studies have demonstrated the effectiveness of combining parameters from multiple models through strategies such as parameter decomposition, coefficient optimization and subspace learning. These methods significantly reduce the need for expensive joint training and have shown strong empirical performance across diverse tasks. However, these approaches predominantly treat merging as a problem of parameter space decomposition or fusion coefficient optimization, while largely overlooking the role of directional information in both parameter space and feature space. In practice, naïve merging introduces inconsistencies in dominant parameter directions and disrupts structural coherence across models, which can degrade performance. Moreover, coefficient-based optimization methods implicitly assume compatible feature-space directions across models. However, Neural Collapse indicates that class features follow structured directional patterns, which may differ across independently trained models, making coefficient optimization alone insufficient. In this work, we emphasize the importance of *directional alignment* and introduce a unified geometric framework, *Merging with Directional Alignment* (`MDA`), which aligns directional structures consistently in both the parameter and feature spaces. Our theoretical analysis demonstrates that directional alignment improves structural coherence and tightens generalization bounds. Extensive experiments across benchmarks, model scales, and task configurations confirm the effectiveness of our approach.

## 1 Introduction

Fine-tuning has become the standard approach for adapting large pretrained models to downstream tasks (Hu et al., 2022; Muqeeth et al., 2024). However, as the number of tasks grows, maintaining a separate set of tuned parameters or adapters for each task leads to storage costs that scale linearly with the task count. In practical scenarios such as edge deployment and multi-task services, this overhead quickly becomes prohibitive. To address this issue, model merging (Ilharco et al., 2022) has emerged. By combining task-specific fine-tuned weights into a single shared model, it significantly reduces storage requirements while retaining task performance.

Existing model merging methods can be broadly categorized into two families: the data-free approach and the data-based approach. Data-free approaches mostly exploit the algebraic properties of the parameter space, employing strategies such as weighted averaging (Ilharco et al., 2022; Matena & Raffel, 2022), subspace projection (Wei et al., 2025), or matrix decomposition (Gargiulo et al., 2025) to alleviate task interference. These methods do not require access to task data, which makes them attractive in terms of privacy and scalability. In contrast, data-based approaches leverage real or synthetic data to learn or optimize fusion coefficients (Yang et al., 2023a; Tang et al., 2023), thus preserving the performance of every task after merging. Although both paradigms have clear advantages, the former is broadly applicable as it only leverages the intrinsic properties of parameters, and the latter offers stronger performance guarantees, most of the existing work has primarily focused on fusion coefficient optimization. What has been largely overlooked is the role of directional information, i.e., how the direction of high-dimensional parameter or feature space fundamentally shapes the behavior of the merged model. This raises an important question:

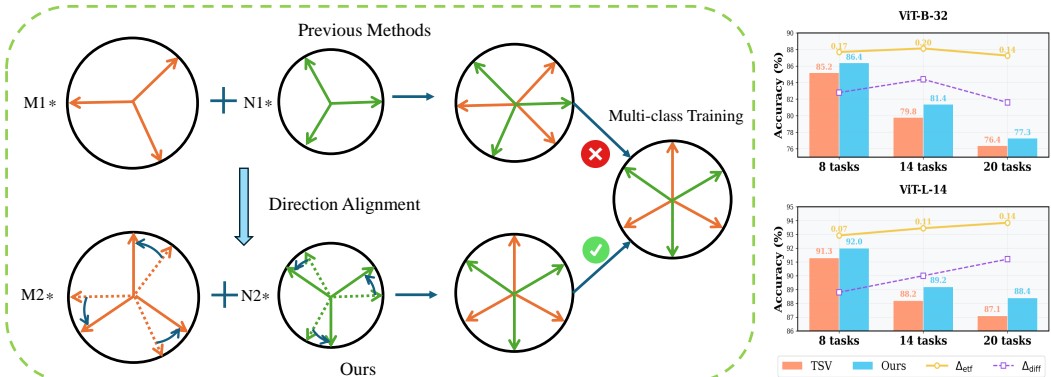

Figure 1: **Left Figure:** A comparison between our method and previous methods. Our method constructs a simplex Equiangular Tight Frame (ETF) as a geometric basis to align task-specific parameter directions, thereby achieving directional consistency that better approximates the behavior of multi-class joint training. M1, N1, M2 and N2 are fusion coefficients, rather than model or classifier weights. **Right Figure:** Correlation between the directional deviation $\Delta_{\text{ETF}}$ (from the ideal ETF geometry) and the performance gap $\Delta_{\text{diff}}$ across tasks. Compared with *task singular vectors* (TSV) (Gargiulo et al., 2025), our approach reveals a clear correlation between the directional deviation $\Delta_{\text{etf}}$ from the simplex ETF structure and the observed performance $\Delta_{\text{diff}}$ gap: larger deviations from simplex ETF structure reliably indicate larger performance gaps across tasks.

> *Should we jointly optimize both aspects for more effective model fusion, since vectors in the parameter or feature space are fundamentally characterized by both magnitude and direction?*

In this work, we argue the importance of direction and address this limitation by optimizing the directional components of the task vectors, both in the parameter space and in the feature space, to achieve superior merging performance. *direction* is used to represent the orientation of parameter space or feature space. Following prior work (Ilharco et al., 2022), we adopt the standard definition of *task vector*, obtained by subtracting the model parameters before fine-tuning from those after fine-tuning on a given task. From an empirical perspective, a well-trained single-task model tends to encode task-specific information in a small number of dominant directions in the parameter space. When models are trained independently, these dominant directions generally do not align, and naïvely combining their parameters can lead to destructive interference. Since parameter space is finite-dimensional, misaligned task directions reduce the separability of the merged model and increase unwanted interactions across tasks. In this work, we regard the simplex *Equiangular Tight Frame* (ETF) as an idealized form of maximally separable directional structure in such finite-dimensional spaces, providing a geometric reference for understanding why preserving or recovering directional consistency is beneficial. In the feature space, we leverage the phenomenon of Neural Collapse (Papyan et al., 2020), which shows that under balanced data and mild training conditions, both classifier weights and class-mean features tend to organize into structured directional templates. This provides geometric motivation for enforcing directional consistency across models in the representation space. As illustrated in Figure 1 (left), naïvely merging models trained on Task 1 (orange) and Task 2 (green) produces representations with clear directional inconsistencies when compared with those obtained from joint multi-class training. These inconsistencies highlight the need to explicitly align directional structures to achieve robust merging performance.

This perspective motivates us to ask: *to what extent does the failure of model merging stem from deviations from this ideal ETF geometry?* To answer this, we measure how closely the merged space align with the ETF structure and whether such deviations correlate with performance degradation. Specifically, we quantify this deviation as: $\Delta_{\text{etf}} = \Delta_{\text{Ours}} - \Delta_{\text{TSV}}, \Delta_{\text{Ours}} = \frac{1}{T} \sum_{t=1}^{T} \cos(W_{\text{Ours}}^t, W_{\text{ETF}}^t), \Delta_{\text{TSV}} = \frac{1}{T} \sum_{t=1}^{T} \cos(W_{\text{TSV}}^t, W_{\text{ETF}}^t), \cos(A, B) = \frac{\langle A,B \rangle_F}{\|A\|_F \|B\|_F}$, where $W_{\text{Ours}}^t$ represents the merged parameter space using our method, $W_{\text{TSV}}^t$ represents the merged parameter space using the TSV method (Gargiulo et al., 2025), $W_{\text{ETF}}^t$ denotes the corresponding standard ETF structure, $\langle A, B \rangle_F = \text{Tr}(A^\top B)$ is the Frobenius inner product, $\|A\|_F = \sqrt{\text{Tr}(A^\top A)}$ is the Frobenius norm, $\Delta_{\text{Ours}}$ denotes the cosine distance between the direction-aligned parameter space and the standard ETF, while $\Delta_{\text{TSV}}$ denotes the cosine distance between the pre-alignment parameter

space and the standard ETF, and $T$ denotes the total number of tasks. As shown in the right panels of Figure 1, the performance gap $\Delta_{\text{diff}}$ between our method and the baseline TSV approach (purple dashed line) exhibits a strong correlation with $\Delta_{\text{etf}}$ across 8 task configurations on ViT-B-32 and ViT-L-14 architectures. These empirical observations validate our hypothesis that directional alignment is crucial for effective model merging.

Motivated by the above analysis, we propose a novel *direction alignment* framework that can operate flexibly in both parameter and feature spaces. Concretely, in parameter space, we first apply low-rank decomposition to task-specific parameter vectors and reconstruct a shared parameter space. The reconstructed parameters are then aligned onto a pre-defined simplex ETF, ensuring that the merged model inherits the optimal geometric configuration. In feature space, we introduce a joint optimization framework that simultaneously learns fusion coefficients and task-specific rotation matrices. This process aligns the extracted features with the ETF structure, while balancing entropy minimization, alignment consistency, and rotation regularization. Together, these two components, parameter-space alignment and feature-space alignment, enable the merged model to narrow the gap to multi-class joint training and mitigating the performance degradation. Our contributions are threefold:

1. We highlight the critical role of directional alignment, which becomes particularly significant in model merging, encompassing both parameter and feature spaces.

2. We theoretically demonstrate the structural advantages introduced by directional alignment, showing that it promotes more coherent geometric organization in both parameter and feature spaces and yields provable generalization benefits.

3. We validate our approach through extensive experiments across diverse benchmarks and task configurations, varying in class counts, model scales, and data availability. Our results demonstrate the effectiveness of the proposed method, showing that it consistently improves performance across diverse tasks and datasets. The source code of our framework is accessible at `https://anonymous.4open.science/r/MDA-E188`.

## 2 RELATED WORK

**Data-free model merging.** Data-Free Model Merging methods operate solely on model parameters, offering significant practical advantages by eliminating the need for any training or validation data. Task Arithmetic (Ilharco et al., 2022) introduces the fundamental concept of task vectors, the difference between fine-tuned and pre-trained weights, and merges them through simple arithmetic averaging. TIES-Merging (Yadav et al., 2023) enhances this approach by trimming low-magnitude parameters, resolving sign conflicts, and performing disjoint merging of sparse vectors. Fisher Merging (Matena & Raffel, 2022) employs the Fisher Information matrix to prioritize important parameters during weighted averaging. More recent advances include Consensus Merging (Wang et al., 2024), which identifies and removes selfish parameters that benefit only specific tasks, and Adaptive Weight Disentanglement (AWD) (Xiong et al., 2024) that explicitly promotes orthogonality among task vectors to reduce interference. Task Singular Vectors (TSV) (Gargiulo et al., 2025) represents a geometric approach by applying singular value decomposition to per-layer task matrices, enabling compression and interference reduction through whitening transformations. Similarly, DOGE (Wei et al., 2025) formulates merging as a constrained optimization problem solved via adaptive projective gradient descent within a shared subspace. Most recently, Isotropic Merging (ISO) (Marczak et al., 2025) proposes to flatten the singular value spectrum of task matrices to enhance subspace alignment between task-specific and merged models, achieving state-of-the-art performance through both common and task-specific subspace integration. Although previous data-free methods have achieved notable performance improvements, most of them rely on decomposing the parameter space into different subspaces. Such partitioning breaks the global geometric coherence of the parameter space among tasks. In contrast, our approach introduces a new paradigm that performs **global parameter-space alignment** from a unified geometric perspective rather than separating the space into different components.

**Data-based model merging.** Data-Based Model Merging methods leverage additional data, typically unlabeled examples from the test distribution, to guide the merging process and achieve improved performance. AdaMerging (Yang et al., 2023a) treats merging coefficients as learnable parameters optimized by minimizing the prediction entropy in test data. Representation Surgery (Yang et al., 2024) extends this approach by explicitly aligning the intermediate representations of the merged

model with those of individual task-specific models, using test data to calibrate feature-level biases. These methods generally achieve higher accuracy by adapting to the target data distribution, but this advantage comes with significant costs: the requirement for data collection introduces practical constraints and potential privacy concerns, the additional computation during merging increases resource requirements, and the performance gains are often sensitive to the quality and representativeness of the provided data. Moreover, these approaches typically overlook the geometric implications of neural collapse in feature space, failing to preserve the optimal directional relationships between class prototypes that emerge in well-trained models. To address this, our framework extends the same directional alignment principle into the feature space, where we not only consider optimizing the fusion coefficients but also explicitly enforcing the directional coherence among task representations.

## 3 Preliminaries

### 3.1 Task Vector

We draw inspiration from prior work (Ilharco et al., 2022) that introduces the concept of task vectors and leverages task arithmetic to combine task-specific vectors from multiple tasks, thereby enabling the merged representation to inherit multi-task capabilities and achieve better generalization. For a given task $k$, its corresponding task vector $\boldsymbol{\tau}_k$ is computed as the vector by subtracting the pre-trained parameters $\theta_0$ from the fine-tuned parameters $\theta_k$: $\boldsymbol{\tau}_k = \theta_k - \theta_0$.

Furthermore, multiple task vectors $\{\boldsymbol{\tau}_k\}_{k=1}^K$ are aggregated and combined with the pre-trained model through a weighted summation, yielding the multi-task model: $\theta_{\text{MTL}} = \theta_0 + \lambda \sum_{k=1}^K \boldsymbol{\tau}_k$, where the merging coefficient $\lambda$ denotes the task-wise importance coefficient for adaptive vector fusion.

To achieve finer-grained control over the fusion process, our method, inspired by Yang et al. (2023a), performs layer-wise adaptation of task vectors $\boldsymbol{\tau}_k^{(l)} = \theta_k^{(l)} - \theta_0^{(l)}$. Thus, the formulation can be generalized as follows: $\theta^{(l)} = \theta_0^{(l)} + \lambda^{(l)} \sum_{k=1}^K \boldsymbol{\tau}_k^{(l)}$, where the merging coefficient $\lambda^{(l)}$ denotes the layer-wise importance coefficient for adaptive vector fusion.

### 3.2 Neural Collapse

Neural collapse (Papyan et al., 2020) refers to a highly structured phenomenon that emerges in the terminal phase of training on balanced datasets (i.e., after achieving a near-zero training error). Specifically, the last-layer features and classifier weights converge towards a simplex ETF geometry, formally defined as follows:

**Definition 1** (Simplex Equiangular Tight Frame). *A simplex ETF is a matrix* $W_{ETF} = [w_1, w_2, \ldots, w_C] \in \mathbb{R}^{d \times C}$ *composed of $C$ vectors and* $w_i \in \mathbb{R}^d$ *with* $d \geq C - 1$, *satisfying:*

$$W_{ETF} = \sqrt{\frac{C}{C-1}} \mathbf{U} \left( \mathbf{I}_C - \frac{1}{C} \mathbf{1}_C \mathbf{1}_C^\top \right), \tag{1}$$

*where* $\mathbf{U} \in \mathbb{R}^{d \times C}$ *satisfies* $\mathbf{U}^\top \mathbf{U} = \mathbf{I}_C$. *For any* $w_i, w_j \in W_{ETF}$, *we have*

$$w_i^\top w_j = \frac{C}{C-1} \delta_{i,j} - \frac{1}{C-1}, \quad \forall i, j \in [1, C], \tag{2}$$

*with* $\delta_{i,j} = 1$ *if* $i = j$ *and* $0$ *otherwise.*

Building on this definition, neural collapse can be characterized by four convergent properties:

**NC1** (Variability collapse). Intra-class feature variability vanishes, and features from the same class collapse to their class mean: $\Sigma_W^c = \frac{1}{n_c} \sum_{i=1}^{n_c} (\mathbf{h}_{c,i} - \mathbf{h}_c)(\mathbf{h}_{c,i} - \mathbf{h}_c)^T \to 0$, where $n_c$ is the number of samples in class $c$, $\mathbf{h}_{c,i}$ is the feature of the $i$-th sample, and $\mathbf{h}_c$ denotes the class mean of class $c$.

**NC2** (Convergence to simplex ETF). Class means become maximally separated, forming a centered simplex ETF: $\hat{\mathbf{h}}_c = \frac{\mathbf{h}_c - \mathbf{h}_G}{\|\mathbf{h}_c - \mathbf{h}_G\|}$, $\mathbf{h}_G = \frac{1}{\sum_k n_c} \sum_{k=1}^K \sum_{i=1}^{n_c} \mathbf{h}_{c,i}$.

**NC3** (Self-duality with classifier weights). The normalized class means align with the corresponding classifier weights: $\hat{\mathbf{h}}_c = \frac{w_c}{\|w_c\|}$, where $w_c$ denotes the classifier weight vector for class $c$.

**NC4** (Nearest class center decision rule). The classifier reduces to a nearest-class-mean classifier: $\arg\max_c \langle \mathbf{h}, w_c \rangle = \arg\min_c \|\mathbf{h} - \mathbf{h}_c\|$, where $\mathbf{h}$ is the feature of a test sample.

Although the standard Neural Collapse analysis primarily focuses on the final classifier layer, subsequent empirical and theoretical studies have shown that ETF-like organization progressively emerges throughout intermediate layers (Parker et al., 2023). This finding suggests that the representations at each layer tend to form locally collapsed, equiangular structures within their effective feature subspaces. In parallel, the Neural Feature Ansatz (NFA) (Beaglehole et al., 2024a) provides a complementary dynamical perspective, showing that during training process, the dominant singular directions of the weight matrices co-evolve with the feature covariance of the corresponding layer. This coupling implies that both the weights and features are jointly aligned toward directions that promote maximal inter-class separation, a property consistent with ETF geometry. Taken together, these two theoretical insights motivate our assumption that the column space of each layer's weight matrix can be approximated by a near-ETF subspace. This perspective captures the intrinsic co-alignment dynamics between weights and features observed across depth, providing a tractable geometric foundation for the layer-wise regularization design, and offering a theoretical explanation for the empirical correlation between the directional deviation $\Delta_{\mathrm{ETF}}$ and the performance gap $\Delta_{\mathrm{diff}}$ observed in Figure 1. We emphasize that the subsequent derivations are built upon this assumption, the validity of which is supported by both the theoretical insights and empirical evidence.

## 4 MERGING WITH DIRECTIONAL ALIGNMENT

Most prior works either partition the parameter space into different subspaces (Gargiulo et al., 2025; Wei et al., 2025; Marczak et al., 2025), or formulate model merging as an optimization problem over task-specific fusion coefficients (Yang et al., 2023a). In contrast, our method provides a unified geometric perspective that views model merging through the lens of directional alignment. In the parameter space, we align the shared subspace with the simplex ETF directions to ensure globally consistent task representations. In the feature space, we leverage the ETF structure naturally formed between the final-layer features and classifier weights, and jointly optimize the fusion coefficients and rotation matrices to achieve feature-level directional consistency. By aligning task updates with the simplex ETF geometry, we bridge parameter-space and feature-space alignment within a single framework, and further provide a theoretical analysis.

### 4.1 PARAMETER-SPACE DIRECTIONAL ALIGNMENT

We conduct parameter-space analysis under data-free conditions, focusing on optimizing the intrinsic geometric properties of multi-task parameters. Although the subsequent theoretical analysis could, in principle, apply to any $\tau_{\mathrm{share}}$ satisfying $\tau_{\mathrm{etf}} = \tau_{\mathrm{share}} P_{\mathrm{ETF}}$, we specifically adopt the SVD-based construction for consistency with our overall framework and in line with prior works (Gargiulo et al., 2025; Marczak et al., 2025; Wei et al., 2025). This choice not only ensures comparability with existing model-merging approaches but also provides a statistically optimal low-rank approximation that captures the dominant shared components across tasks.

#### 4.1.1 STRUCTURAL ADVANTAGE

For tasks $\{T_1, T_2, \ldots, T_T\}$ and layers $l = 1, \ldots, L$, we first obtain their SVD decompositions:

$$\tau_t^{(l)} = U_t^{(l)} \Sigma_t^{(l)} (V_t^{(l)})^\top.$$

We retain the top-$k$ principal components $U_t^{(l,k)} := U_t^{(l)}[:, 1:k]$ and construct a shared representation

$$\tau_{\mathrm{share}}^{(l)} := U_{\mathrm{share}}^{(l)}[:, 1:d_{\mathrm{out}}] \in \mathbb{R}^{d_{\mathrm{in}} \times d_{\mathrm{out}}},$$

where $U_{\mathrm{share}}^{(l)}$ is obtained from the SVD of the concatenated basis

$$U_{\mathrm{cat}}^{(l)} = \left[ U_1^{(l,k)}, U_2^{(l,k)}, \cdots, U_T^{(l,k)} \right] \in \mathbb{R}^{d_{\mathrm{in}} \times kT}, \qquad U_{\mathrm{cat}}^{(l)} = U_{\mathrm{share}}^{(l)} \Sigma_{\mathrm{share}}^{(l)} (V_{\mathrm{share}}^{(l)})^\top.$$

Here $d_{\mathrm{in}}$ is the parameter input dimension at layer $l$, and $d_{\mathrm{out}}$ is the parameter output dimension. We set $k = d_{\mathrm{out}}/T$ so that the concatenated dimension $kT$ exactly matches the output dimension $d_{\mathrm{out}}$,

**Algorithm 1** Parameter-Space Alignment

1: **Input:** Task vectors $\{\tau_t\}_{t=1}^T$, rank $k$
2: **Output:** ETF-guided task vectors $\tau_{\text{etf}} = \{\tau_t^{\text{etf}}\}_{t=1}^T$
3: **for** $t = 1$ to $T$ **do**
4:    SVD: $\tau_t^{(l)}$: $\tau_t^{(l)} = U_t^{(l)} \Sigma_t^{(l)} (V_t^{(l)})^\top$
5:    Keep top-$k$ components: $U_t^{(l)}[:, 1:k]$
6: **end for**
7: Concatenate $\{U_t^{(l,k)}\}_{t=1}^T$ to form $U_{\text{cat}}^{(l)}$
8: $U_{\text{cat}}^{(l)} = U_{\text{share}}^{(l)} \Sigma_{\text{share}}^{(l)} (V_{\text{share}}^{(l)})^\top$
9: Truncate $U_{\text{share}}^{(l)}$ to its first $d_{\text{out}}$ columns to obtain $\tau_{\text{share}}^{(l)}$
10: Build ETF matrix $W_{\text{ETF}}$ according to Appendix B.1:
$(W_{\text{ETF}} W_{\text{ETF}}^\top)_{ii} = ||w_i||^2 = 1$
$(W_{\text{ETF}} W_{\text{ETF}}^\top)_{ij} = w_i w_j^\top = -\frac{1}{C-1}, i \neq j$
11: $\tau_{\text{etf}} \leftarrow \tau_{\text{share}} W_{\text{ETF}}^\top W_{\text{ETF}}$

**Algorithm 2** Feature-Space Alignment

1: **Input:** ETF-guided task vectors $\tau_{\text{etf}} = \{\tau_t^{\text{etf}}\}_{t=1}^T$, the total number of layers in the model is $L$, datasets $\{\mathcal{D}_t\}_{t=1}^T$, epochs $E$
2: **Output:** Fusion coefficients $\boldsymbol{\lambda}$, rotation matrices $\{\mathbf{R}^t\}_{t=1}^T$
3: Initialize $\boldsymbol{\lambda_0} = \frac{1}{T}\mathbf{1}_T$, $\mathbf{R}^t = \mathbf{I}_d$; construct ETF $M^*$
  // Layer-wise Fusion
4: **for** $l = 1$ to $L$ **do**
5:    Fusion: $\theta^{(l)} = \theta_0^{(l)} + \lambda^{(l)} \sum_{t=1}^T \tau_t^{\text{etf},(l)}$
6: **end for**
7: **for** $e = 1$ to $E$ **do**
8:    Extract rotated features: $\tilde{\mathbf{h}}_{t,i} = \mathbf{R}^t \phi_{\boldsymbol{\theta}}(\mathbf{x}_{t,i})$
9:    Compute $\mathcal{L} = \mathcal{L}_{\text{entropy}} + \alpha\mathcal{L}_{\text{align}} + \beta\mathcal{L}_{\text{rotation}}$
10:    Update $\lambda^{(l)}$ via $\mathcal{L}_{\text{entropy}}$
11:    Update $\mathbf{R}^t$ via $\mathcal{L}_{\text{align}}$ and $\mathcal{L}_{\text{rotation}}$
12: **end for**

ensuring a balanced representation across all $T$ tasks. The proposed method is illustrated through Algorithm 1 and Figure 4 in Appendix F.

The ETF-aligned task parameter space at each layer is obtained through the following direction alignment:

$$\tau_{\text{etf}}^{(l)} = \tau_{\text{share}}^{(l)} W_{\text{ETF}}^\top W_{\text{ETF}} = \tau_{\text{share}}^{(l)} \mathcal{P}_{\text{ETF}} \tag{3}$$

where $W_{\text{ETF}} \in \mathbb{R}^{C \times d_{\text{out}}}$ denotes an ETF matrix, $C = \sum_t C_t$ classes, each task $t$ corresponds to a subset of $C_t$ classes, $\mathcal{P}_{\text{ETF}} = W_{\text{ETF}}^\top W_{\text{ETF}}$ is the direction alignment matrix, ETF-guided task vectors $\tau_{\text{etf}} = \{\tau_t^{\text{etf}}\}_{t=1}^T$ are obtained by projecting the shared parameter subspace $\tau_{\text{share}}$ onto the ETF basis.

**Theorem 1** (ETF Structural Coherence). *Let $\tau_{ideal}^{(l)} \in \mathbb{R}^{d_{in} \times d_{out}}$ represent the ideal jointly-trained multi-class parameters at layer $l$, and $\tau_{share}^{(l)}$ be the SVD-reconstructed parameters from task merging. When ETF is constructed, the ETF-aligned task vector $\tau_{etf}^{(l)} = \tau_{share}^{(l)} \mathcal{P}_{ETF}$ satisfies:*

***Approximation to Ideal Parameters:***

$$\|\tau_{ideal}^{(l)} - \tau_{etf}^{(l)}\|_F^2 \leq \|\tau_{ideal}^{(l)} - \tau_{share}^{(l)}\|_F^2 - g \|\tau_{share}^{(l)}(I - \tfrac{C-1}{C}\mathcal{P}_{ETF})\|_F^2 \tag{4}$$

*where $g \in (0, 1]$ quantifies the directional correction gain.*

Here, $C$ denotes the number of classes, and the factor $\frac{C-1}{C}$ originates from the geometry of the simplex ETF, where the pairwise cosine similarity between class directions is $-\frac{1}{C-1}$. As $C$ increases, the ETF structure becomes more balanced and representative, and the factor $\frac{C-1}{C}$ in the bound tends to amplify the effect of ETF projection. Meanwhile, the coefficient $g \in (0, 1]$ quantifies the effectiveness of the directional correction; a larger $g$ indicates that the ETF geometry more accurately captures the shared subspace. Assuming that the correction coefficient $g$ remains approximately constant under ideal conditions, the ETF-aligned representation $\tau_{\text{etf}}$ is expected to be closer to the ideal parameters $\tau_{\text{ideal}}$ than the shared subspace $\tau_{\text{share}}$. This suggests that, when the directional correction assumption holds well, ETF alignment can provide a tighter approximation to the ideal joint-training solution, and its effectiveness may increase as the number of classes $C$ grows and the ETF geometry becomes more representative. A complete derivation of Theorem 1 is provided in Appendix B.2.1.

### 4.1.2 GENERALIZATION ADVANTAGE

To assess whether the merged model exhibits improved generalization, we provide a theoretical analysis showing that our method enhances generalization performance. The detailed derivation is deferred to Appendix B.2.2.

**Theorem 2** (Generalization bound). *Let $\hat{\mathcal{R}}(\cdot)$ denote the empirical risk and $\mathcal{R}(\cdot)$ the population risk. Assume the loss $\ell(z, y)$ is $L$-Lipschitz in the logits and the feature satisfies $\|\phi(x)\|_2 \leq R$. Let $C \geq 2$ and let $W_{\mathrm{ETF}} \in \mathbb{R}^{C \times d_{out}}$ denote an ETF matrix. Define the ETF Gram operator $\mathcal{P}_{\mathrm{ETF}} = W_{\mathrm{ETF}}^\top W_{\mathrm{ETF}}$, and note that in the ideal simplex case*

$$\mathcal{P}_{\mathrm{ETF}} \;=\; \tfrac{C}{C-1} \Pi_{\mathcal{S}},$$

*where $\Pi_{\mathcal{S}}$ is the orthogonal projector onto the ETF subspace $\mathcal{S} = \mathrm{rowspan}(W_{\mathrm{ETF}}) \subset \mathbb{R}^{d_{out}}$, which has dimension $C - 1$. Assume for the sake of comparing generalization that both can be trained to achieve the same empirical risk, let $\tau_{\mathrm{share}}, \tau_{\mathrm{etf}} \in \mathbb{R}^{d_{in} \times d_{out}}$ satisfy $\tau_{\mathrm{etf}} = \tau_{\mathrm{share}} \mathcal{P}_{ETF}$ and $\hat{\mathcal{R}}(\tau_{\mathrm{share}}) = \hat{\mathcal{R}}(\tau_{\mathrm{etf}}) = \hat{\mathcal{R}}^*$. Then with probability at least $1 - \delta$ over the training set, we have*

$$\mathcal{R}(\tau_{share}) - \mathcal{R}(\tau_{etf}) \;\leq\; \left(\tfrac{C}{C-1}\right)^2 \frac{LR}{\sqrt{n}} \frac{\left\|\tau_{share}(I - \Pi_{\mathcal{S}})\right\|_F^2}{\left\|\tau_{share}\right\|_F} \;+\; c\sqrt{\tfrac{\log(1/\delta)}{n}}, \tag{5}$$

*where $L$ denotes the Lipschitz constant of the loss in the logits, $R$ is the upper bound on the norm of the feature, $n$ is the number of training samples, and $c > 0$ is an absolute constant.*

Theorem 2 characterizes the generalization advantage induced by directional alignment through a refined Rademacher complexity analysis. Specifically, the bound shows that the expected risk difference between the shared parameter solution $\tau_{\mathrm{share}}$ and the aligned solution $\tau_{\mathrm{etf}}$ scales with $\frac{LR}{\sqrt{n}} \frac{\left\|\tau_{\mathrm{share}}(I - \Pi_{\mathcal{S}})\right\|_F^2}{\left\|\tau_{\mathrm{share}}\right\|_F}$, where the factor $\frac{\left\|\tau_{\mathrm{share}}(I - \Pi_{\mathcal{S}})\right\|_F^2}{\left\|\tau_{\mathrm{share}}\right\|_F}$ directly reflects the directional benefits introduced by the alignment. Intuitively, aligning the parameter space onto the ETF basis reduces the effective hypothesis complexity, thereby tightening the generalization gap. This reduction highlights that our method not only preserves empirical risk but also improves generalization by lowering the capacity associated with the Rademacher complexity.

## 4.2 Feature-space Directional Alignment

We assign a trainable direction alignment matrix to the final layer of each task and jointly optimize the fusion coefficients and direction matrices using an ETF alignment loss, enhancing generalization and mitigating feature overlap and inter-task interference. In our framework, Algorithm 1 and Algorithm 2 are used sequentially. We first obtain the ETF-aligned task vectors $\tau_{\mathrm{etf}} = \{\tau_t^{\mathrm{etf},(l)}\}_{t=1}^T$ through Algorithm 1. Then, these aligned vectors are used to initialize the model parameters in Algorithm 2 as $\theta^{(l)} = \theta_0^{(l)} + \lambda^{(l)} \sum_{t=1}^T \tau_t^{\mathrm{etf},(l)}$. On this basis, Algorithm 2 further refines the model by jointly optimizing the fusion coefficients $\lambda^{(l)}$ and the rotation matrices $\{\mathbf{R}^t\}_{t=1}^T$ in the feature space. Based on the above analysis, we design the following loss function to jointly optimize both the fusion coefficients and the directional alignment matrix.

**Definition 2** (Task Vector Fusion with Feature Direction Alignment). *The fusion model combines task vectors in parameter space and applies direction alignment in feature space:*

$$\theta^{(l)} = \theta_0^{(l)} + \lambda^{(l)} \sum_{t=1}^T \boldsymbol{\tau}_t^{(l)}, \tag{6}$$

$$\tilde{\mathbf{h}}_{t,i} = \mathbf{R}^t \mathbf{h}_{t,i}, \quad \mathbf{R}^t \in SO(d), \tag{7}$$

*where $\lambda^{(l)}$ is layer-wise trainable fusion coefficient, $\{\mathbf{R}^t\}_{t=1}^T$ are task-specific rotation matrices, and $SO(d)$ denotes the group of all $d \times d$ pure rotation matrices. In the equation, constraining the transformation to $SO(d)$ ensures that the feature rotations preserve the vector norms and angles.*

Our approach formulates model merging as a joint optimization problem that simultaneously learns task vector fusion coefficients in parameter space and rotation matrices in feature space. The complete objective function combines three complementary loss terms:

$$\min_{\{\lambda^{(l)}\}_{l=1}^L, \{\mathbf{R}^t\}_{t=1}^T} \mathcal{L}(\{\lambda^{(l)}\}, \{\mathbf{R}^t\}) = \mathcal{L}_{\mathrm{entropy}} + \alpha\, \mathcal{L}_{\mathrm{align}} + \beta\, \mathcal{L}_{\mathrm{rotation}}. \tag{8}$$

The first item is the entropy minimization loss $\mathcal{L}_{\mathrm{entropy}}$, which aims to improve the calibration across the merged tasks (Yang et al., 2023a) as

$$\mathcal{L}_{\text{entropy}} = -\sum_{i=1}^{C} \sigma(\mathbf{x})_i \, \log \sigma(\mathbf{x})_i, \tag{9}$$

where $\sigma(\mathbf{x})$ denotes the Softmax output distribution for $\mathbf{x}$, and $\sigma(\mathbf{x})_i$ indicates the probability corresponding to the $i$-th class.

The second term is the neural collapse assignment loss $\mathcal{L}_{\text{align}}$, which enforces the structure of the ETF. We align normalized rotated features with their ETF targets as

$$\mathcal{L}_{\text{align}} = \sum_{t=1}^{T} \frac{1}{n_t} \sum_{i=1}^{n_t} \left\| \frac{\tilde{\mathbf{h}}_{t,i}}{\|\tilde{\mathbf{h}}_{t,i}\|_2} - m^*_{\phi_t(y_{t,i})} \right\|_2^2, \tag{10}$$

where $\tilde{\mathbf{h}}_{t,i} = \mathbf{R}^t \mathbf{h}_{t,i}$, $m^*_{\phi_t(y_{t,i})}$ is the ETF vector corresponding to class $y_{t,i}$ under the feature extraction layer of the neural network $\phi_t : \mathcal{Y}_t \to \{1, \ldots, C\}$.

The last item is the feature-space direction alignment regularization $\mathcal{L}_{\text{rotation}}$, which encourages each task-specific rotation matrix $\mathbf{R}^t$ to align with the optimal Procrustes rotation computed between the empirical class means and the ETF targets:

$$\mathcal{L}_{\text{rotation}} = \sum_{t=1}^{T} \left\| \mathbf{R}^t - \mathbf{R}^t_{\text{proc}} \right\|_F^2, \qquad \mathbf{R}^t_{\text{proc}} = \operatorname*{arg\,min}_{\mathbf{R} \in SO(d)} \left\| M^t \mathbf{R} - M^{t*} \right\|_F^2, \tag{11}$$

where $M^t \in \mathbb{R}^{C_t \times d}$ contains the empirical class means: $[M^t]_{c,:} = \frac{1}{|\{i:y_{t,i}=c\}|} \sum_{i:\, y_{t,i}=c} \frac{\mathbf{h}_{t,i}}{\|\mathbf{h}_{t,i}\|_2}$, and $M^{t*} \in \mathbb{R}^{C_t \times d}$ contains the corresponding ETF targets. Here, $\mathbf{R}^t \in SO(d)$ denotes the learnable rotation matrix for task $t$, which aligns its feature representations with the global feature space. In the Procrustes objective above, the optimization variable $\mathbf{R} \in SO(d)$ represents a candidate rotation matrix in the $d$-dimensional feature space ($\mathbf{R}^\top \mathbf{R} = \mathbf{I}$ and $\det(\mathbf{R}) = 1$), and the resulting $\mathbf{R}^t_{\text{proc}}$ is the optimal rotation that best aligns the empirical class means $M^t$ with the ETF targets $M^{t*}$ in the least-squares sense. The closed-form solution of $\mathbf{R}^t_{\text{proc}}$ follows from the orthogonal Procrustes formulation. $H^t = (M^t)^\top M^{t*} = U^t \Sigma^t (V^t)^\top$ is the singular value decomposition of $H^t$, the optimal rotation is given by $\mathbf{R}^t_{\text{proc}} = U^t V^{t\top}$.

## 5 EXPERIMENTS

To validate the effectiveness of our approach, we conduct experiments on both image classification and natural language understanding tasks. To ensure a fair comparison, we follow the settings of prior work (Gargiulo et al., 2025; Wei et al., 2025; Marczak et al., 2025) in terms of dataset and model selection. We provide more details in Appendix G.

### 5.1 SETTINGS

**Datasets and Models.** For vision tasks, we evaluate our approaches over three benchmark suites comprising 8, 14, and 20 tasks, respectively. More details are provided in Appendix E. To investigate the effect of model capacity, we evaluate our method using three CLIP (Radford et al., 2021) variants, each employing a different ViT (Dosovitskiy et al., 2020) visual encoder: ViT-B/32, ViT-B/16, and ViT-L/14. For Natural Language Process (NLP) tasks, we evaluate our method using Flan-T5-base (Chung et al., 2024) on eight representative datasets from the GLUE benchmark (Wang et al., 2019) in Table 13 in Appendix G.2.

**Baselines.** We conduct a comprehensive comparison with existing methods, covering both data-free and data-based optimization approaches. For data-free methods, we evaluate Task Arithmetic (TA) (Ilharco et al., 2022), Concrete TA (Tang et al., 2023), Ties-Merging (Yadav et al., 2023), Consensus Merging (Wang et al., 2024), AWD TA (Xiong et al., 2024), PCB-Merging (Du et al., 2024), TSV (Gargiulo et al., 2025), ISO (Marczak et al., 2025), and DOGE TA (Wei et al., 2025). For data-based optimization methods, we consider AdaMerging (Yang et al., 2023a), Concrete AM (Tang et al., 2023), Representation Surgery (Yang et al., 2024), AWD AM (Xiong et al., 2024), and DOGE AM (Wei et al., 2025). We denote the TA (Task Arithmetic) methods as data-free, since they perform model fusion solely in the parameter space without accessing any training data. In contrast, the AM (Adaptive Merging) methods are data-based, as they leverage available data to optimize the corresponding fusion coefficients during the merging process. MDA TA is derived from Algorithm 1, while MDA AM initializes with the ETF-aligned parameters obtained from Algorithm 1 and further optimizes them through Algorithm 2.

Table 1: Average accuracy (%) when merging models across a larger number of tasks.

| Method | ViT-B/32 | | | ViT-B/16 | | | ViT-L/14 | | |
|---|---|---|---|---|---|---|---|---|---|
| | 8 tasks | 14 tasks | 20 tasks | 8 tasks | 14 tasks | 20 tasks | 8 tasks | 14 tasks | 20 tasks |
| Pre-trained | 48.4 | 57.3 | 56.1 | 55.1 | 61.3 | 59.8 | 64.4 | 68.0 | 65.1 |
| Weight averaging | 66.5 | 64.4 | 61.1 | 72.1 | 69.3 | 93.1 | 79.4 | 76.6 | 71.5 |
| Task Arithmetic | 70.8 | 65.4 | 60.6 | 75.4 | 70.6 | 65.9 | 84.8 | 79.3 | 74.0 |
| Ties-Merging | 75.1 | 68.0 | 63.4 | 79.2 | 73.3 | 68.2 | 86.9 | 79.5 | 75.7 |
| Consensus TA | 75.0 | 70.4 | 65.4 | 79.0 | 74.5 | 69.7 | 86.2 | 82.2 | 78.9 |
| Consensus TIES | 74.8 | 67.7 | 63.2 | 78.2 | 75.3 | 67.1 | 86.9 | 81.5 | 76.8 |
| TSV TA | 85.2 | 79.8 | 76.4 | 89.0 | 84.5 | 80.5 | 91.3 | 88.2 | 87.1 |
| ISO TA | 82.9 | 78.9 | 73.1 | 89.1 | 84.6 | 79.5 | 91.1 | 88.0 | 87.0 |
| ISO-CLS TA | 80.8 | 79.7 | 76.6 | 88.5 | **85.8** | **82.8** | 90.7 | 89.0 | **88.9** |
| DOGE TA | 80.7 | 77.9 | 72.5 | 86.3 | 82.1 | 79.1 | 88.8 | 87.1 | 81.0 |
| MDA TA | **86.4** | **81.4** | **77.3** | **89.9** | 85.8 | 82.7 | **92.0** | **89.2** | 88.4 |

Table 2: Generalization results on two unseen tasks when merging ViT-B/32 models on six tasks.

| Method | Seen Tasks | | | | | | | Unseen Tasks | | |
|---|---|---|---|---|---|---|---|---|---|---|
| | SUN397 | Cars | RESISC45 | DTD | SVHN | GTSRB | Avg. | MNIST | EuroSAT | Avg. |
| Pre-trained | 63.2 | 59.9 | 60.6 | 43.9 | 23.5 | 30.4 | 46.9 | 47.6 | 45.6 | 46.6 |
| Task Arithmetic | 64.3 | 63.0 | 73.2 | 54.9 | 84.7 | 79.5 | 69.9 | 75.5 | 42.6 | 59.1 |
| Ties-Merging | 68.3 | 65.5 | 76.9 | 54.9 | 75.4 | 72.0 | 68.9 | 73.1 | 47.3 | 60.2 |
| AdaMerging | 68.4 | 71.9 | 87.9 | 69.1 | 92.2 | 93.8 | 80.5 | 77.7 | 47.3 | 62.5 |
| TSV TA | 68.2 | 71.2 | 90.6 | 90.0 | **95.8** | 96.7 | 85.4 | 85.3 | 42.9 | 64.1 |
| ISO TA | 72.4 | **74.2** | 89.8 | 87.1 | 83.7 | 90.8 | 83.0 | 79.9 | 51.2 | 64.1 |
| ISO-CLS TA | **72.6** | 74.1 | 88.5 | 85.3 | 80.0 | 87.5 | 81.3 | 78.2 | **52.3** | 65.3 |
| DOGE TA | 69.8 | 72.6 | 86.6 | 67.6 | 90.8 | 91.6 | 79.8 | 81.3 | 48.2 | 64.8 |
| MDA TA | 71.1 | 72.9 | **92.4** | 91.5 | 95.8 | 97.0 | 86.8 | 85.5 | 47.7 | **66.6** |
| MDA AM wo rotation | **71.5** | **73.2** | 93.1 | 92.9 | 95.8 | 96.9 | 87.2 | 85.5 | 48.2 | 66.9 |
| MDA AM | 71.2 | 72.9 | **93.3** | 94.3 | 96.1 | 97.3 | 87.5 | 86.1 | 48.8 | 67.5 |

## 5.2 DISCUSSION ON PARAMETER-SPACE ALIGNMENT

Table 1 provides compelling evidence that our parameter-space alignment approach delivers substantial directional benefits, which become increasingly pronounced as the number of tasks grows. The improvement in the accuracy of MDA TA over the TSV TA increases from approximately 0.9% at 8 tasks to 2.2% at 20 tasks in ViT-B/16 and from 0.7% to 1.3% in ViT-L/14, indicating that the structural benefits of alignment of the parameter space are amplified as the complexity of the task increases. By enforcing separation in the parameter space, our method ensures that individual task adaptations occupy orthogonal or minimally overlapping subspaces, preventing catastrophic interference. The resulting geometric separation not only preserves task-specific functionality, but also enables seamless multi-task integration.

## 5.3 DISCUSSION ON FEATURE-SPACE ALIGNMENT

As illustrated in Figure 2, incorporating feature direction alignment consistently leads to improved performance. These observations highlight that feature-space alignment acts as a critical structural prior, guiding the fusion process beyond naive parameter averaging or conventional optimization. The resulting improvements demonstrate that carefully orchestrated geometric arrangements in the feature space can substantially enhance multi-task model merging, particularly in scenarios with heterogeneous and high-dimensional tasks. We also conducted a detailed performance comparison in Table 7 and 10, which are provided in Appendix G.2.

## 5.4 DISCUSSION ON GENERALIZATION

Table 2 reports the generalization performance of various merging methods on two unseen tasks (MNIST and EuroSAT) after merging ViT-B/32 models trained on six seen tasks. Our parameter-space alignment approach, MDA TA, achieves the highest average accuracy of 66.6% on unseen tasks, surpassing all baselines, including TSV TA (64.1%) and DOGE TA (64.8%). In particular, our TA not only maintains strong performance on seen tasks (86.8% average), but also exhibits more balanced accuracy across both seen and unseen tasks. Similarly, we conduct generalization tests in the feature space and found that combining directional alignment with optimization of the fusion coefficient consistently improves the generalization capability of the model. From a theoretical perspective, this enhanced generalization can be attributed to the geometric properties induced by the directional alignment. The merged model could capture more task-invariant structures, enabling effective knowledge transfer to novel tasks that are not observed during training.

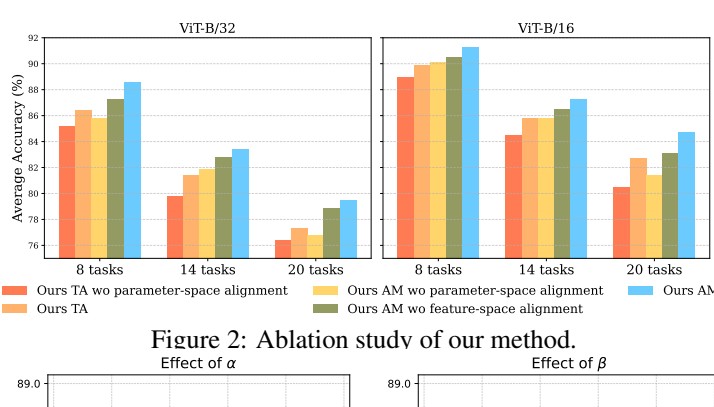

Figure 2: Ablation study of our method.

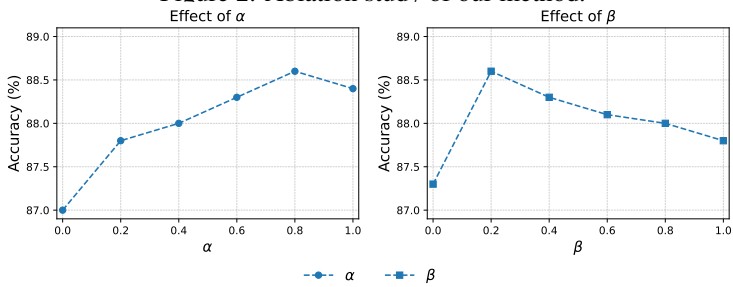

Figure 3: Ablation on the coefficients of different modules in the loss function.

## 5.5 ABLATION STUDY

**Direction Alignment Mechanisms.** We further conduct an ablation study to disentangle the contributions of parameter-space directional alignment and feature-space directional alignment in our framework. As shown in Figure 2, for ViT-B/32, removing parameter-space alignment leads to a noticeable drop, particularly on the 14-task and 20-task benchmarks, highlighting the importance of aligning parameter manifolds to reduce destructive task interference. A similar trend is observed in the AM setting: without parameter- or feature-space alignment , the accuracy decreases compared to the full model, especially as the number of tasks grows. These results confirm that parameter- and feature-space alignments are key to improving both scalability and generalization in model merging.

**Loss Components.** To better understand the role of different components in our objective, we conduct an ablation study on the coefficients $\alpha$ and $\beta$ in Eq. 8 based on ViT-B/32 across eight tasks. Specifically, $\alpha$ controls the neural collapse alignment loss $\mathcal{L}_{\text{align}}$, while $\beta$ balances the feature-space directional alignment loss $\mathcal{L}_{\text{rotation}}$. As shown in Figure 3 (left), increasing $\alpha$ improves performance up to $\alpha = 0.8$, where accuracy peaks at $88.6\%$. This highlights that feature alignment with the shared ETF space is crucial for mitigating task interference and enhancing generalization, though overly large $\alpha$ may over-constrain the model. For $\beta$ (Figure 3, right), moderate values (e.g., $\beta = 0.2$) significantly improve performance, but larger values gradually degrade accuracy, implying that excessive directional consistency hinders adaptation to diverse task directions.

**Rank Dimension $k$.** The ablation study on the rank dimension $k$ is provided in Appendix G.2.2.

## 6 CONCLUSION

In this work, we have established the critical role of directional alignment in model merging, both in parameter and feature spaces. We proposed a unified framework that aligns parameters and feature-space to preserve structural coherence and enhance generalization, and we further validate its effectiveness through both theoretical analysis and empirical experiments. Despite its strengths, our approach has certain limitations. First, due to limited resources, the current feature-space alignment introduces additional computational overhead due to the optimization of the rotation matrices, which may be non-trivial for extremely large-scale models. Second, extending this framework to merge models from different architectures or pre-training paradigms remains an open challenge. To further assess its practicality, future work will evaluate the proposed framework on medical and multimodal datasets, where reliable model merging can have significant real-world impact. Finally, while we focus on vision and NLP tasks, extending the principles of directional alignment to generative models is a promising future direction. In addition, exploring more lightweight alignment strategies to reduce computational cost, as well as studying the robustness of alignment under distribution shifts, may further broaden the applicability of our framework.

ETHICS STATEMENT

This work does not involve human subjects, sensitive personal data, or experiments that may directly cause harm to individuals, groups, or society. All datasets used in our study are publicly available, widely used in the machine learning community, and subject to their respective licenses. We have taken care to ensure fair use of these datasets and avoided introducing biases beyond what is inherent in the original data.

To promote reproducibility and transparency, we provide detailed experimental settings, algorithmic descriptions, and anonymized code. Our contributions are methodological in nature and aim to advance the understanding of the model merging field. We are committed to the principles outlined in the ICLR Code of Ethics and confirm that this work adheres to them in full.

REPRODUCIBILITY STATEMENT

To facilitate community development, we provide detailed descriptions of the datasets used in our experiments (Section 5.1), the compared baselines (Section D), the training configurations (Section 5.1), as well as our proposed methods (Algorithms 1, 2 and Figure 4). In addition, we release anonymized code to reduce the difficulty of reproduction.

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

## A NOTATION

We summarize the main symbols used in this paper as follows.

- $T$: the total number of tasks considered in model merging.
- $C$: the total number of classes across all tasks.
- $\theta_0$: parameters of the pre-trained model.
- $\theta_k$: parameters of the model fine-tuned on task $k$.
- $\tau_k = \theta_k - \theta_0$: task vector of task $k$, representing the update from pre-training to fine-tuning.
- $\tau_k^{(l)} = \theta_k^{(l)} - \theta_0^{(l)}$: layer-wise task vector at layer $l$.
- $\lambda$: merging coefficients; $\lambda^{(l)}$ denotes layer-wise coefficients.
- $W_{\text{ETF}} = [w_1, \ldots, w_C] \in \mathbb{R}^{d \times C}$: simplex equiangular tight frame (ETF) matrix with $C$ class directions.
- $\mathcal{S} = \text{rowspan}(W_{\text{ETF}}) \subset \mathbb{R}^d$: ETF subspace of dimension $C-1$.
- $\Pi_{\mathcal{S}}$: orthogonal projector onto subspace $\mathcal{S}$.
- $\mathcal{P}_{\text{ETF}} = W_{\text{ETF}}^{\top} W_{\text{ETF}} = \frac{C}{C-1} \Pi_{\mathcal{S}}$: ETF Gram operator.
- $\mathbf{h}_{c,i}$: feature vector of the $i$-th sample in class $c$.
- $\mathbf{h}_c = \frac{1}{n_c} \sum_i \mathbf{h}_{c,i}$: class mean feature of class $c$ with $n_c$ samples.
- $\mathbf{R}^t \in SO(d)$: task-specific rotation matrix for task $t$.
- $\tilde{\mathbf{h}}_{t,i} = \mathbf{R}^t \mathbf{h}_{t,i}$: rotated feature of sample $i$ from task $t$.
- $\Sigma_W^c$: within-class covariance matrix for class $c$.

- $\mathcal{L} = \mathcal{L}_{\text{entropy}} + \alpha\mathcal{L}_{\text{align}} + \beta\mathcal{L}_{\text{rotation}}$: overall training loss combining entropy minimization, ETF alignment, and rotation regularization.
- $\mathcal{L}_{\text{entropy}}$: entropy minimization loss to calibrate predictions.
- $\mathcal{L}_{\text{align}}$: ETF alignment loss to enforce simplex ETF structure in features.
- $\mathcal{L}_{\text{rotation}}$: rotation regularization loss encouraging $\mathbf{R}^t$ close to the optimal Procrustes solution.
- $\Delta_{\text{ETF}}$: deviation from the ETF geometry.
- $\Delta_{\text{diff}}$: performance gap metric between methods.

# B  THEORETICAL PROPERTIES

## B.1  ETF CONSTRUCTION METHODS

**SVD-based ETF Construction (when $d_{\text{out}} < C - 1$):**

1. Define centering alignment: $\mathbf{P} = \mathbf{I}_C - \frac{1}{C}\mathbf{1}_C\mathbf{1}_C^\top \in \mathbb{R}^{C \times C}$
2. Compute SVD: $\mathbf{P} = \mathbf{U}\mathbf{\Sigma}\mathbf{V}^\top$
3. Construct ETF: $\mathbf{W}_{\text{ETF}}^\top = \left(\sqrt{\frac{C}{C-1}}\mathbf{U}_{:,:d_{\text{out}}}\right)^\top \in \mathbb{R}^{d_{\text{out}} \times C}$

The centering alignment $\mathbf{P}$ naturally encodes the **ideal multi-class structure** where classes are maximally separated with equal pairwise correlations of $-\frac{1}{C-1}$.

**QR-based ETF Construction (when $d_{\text{out}} \geq C - 1$):**

1. Generate a random Gaussian matrix $\mathbf{X} \in \mathbb{R}^{d_{\text{out}} \times C}$ with entries $\mathbf{X}_{i,j} \sim \mathcal{N}(0,1)$.
2. Apply QR decomposition: $\mathbf{X} = \mathbf{Q}\mathbf{R}$, where $\mathbf{Q} \in \mathbb{R}^{d_{\text{out}} \times C}$ has orthonormal columns.
3. Center the columns: $\mathbf{Q}_c = \left(I - \frac{1}{C}\mathbf{1}\mathbf{1}^\top\right)\mathbf{Q}$, so that $\mathbf{Q}_c^\top\mathbf{1} = 0$.
4. Construct ETF by rescaling:

$$\mathbf{W}_{\text{ETF}}^\top = \sqrt{\frac{C}{C-1}}\,\mathbf{Q}_c(:,1:C) \;\in\; \mathbb{R}^{d_{\text{out}} \times C}.$$

## B.2  PARAMETER-SPACE DIRECTION ALIGNMENT

### B.2.1  ETF STRUCTURAL COHERENCE

*Proof.* **Setup and dimensions.** Let $W_{\text{ETF}} \in \mathbb{R}^{C \times d_{\text{out}}}$ be a row-wise unit-norm simplex ETF, so

$$(W_{\text{ETF}}W_{\text{ETF}}^\top)_{ii} = 1, \quad (W_{\text{ETF}}W_{\text{ETF}}^\top)_{ij} = -\tfrac{1}{C-1} \ (i \neq j).$$

Then the frame operator is

$$\mathcal{P}_{\text{ETF}} = W_{\text{ETF}}^\top W_{\text{ETF}} = \tfrac{C}{C-1}\,\Pi_{\mathcal{S}}, \qquad \mathcal{S} = \text{rowspan}(W_{\text{ETF}}) \subseteq \mathbb{R}^{d_{\text{out}}},$$

with eigenvalues $\frac{C}{C-1}$ (mult. $C - 1$) on $\mathcal{S}$ and 0 (mult. $d_{\text{out}} - (C - 1)$) on $\mathcal{S}^\perp$.

For $\tau_{\text{share}}^{(l)} \in \mathbb{R}^{d_{\text{in}} \times d_{\text{out}}}$, define the aligned parameter

$$\tau_{\text{etf}}^{(l)} = \tau_{\text{share}}^{(l)}\mathcal{P}_{\text{ETF}} = \tfrac{C}{C-1}\,\tau_{\text{share}}^{(l)}\Pi_{\mathcal{S}}.$$

**Inner-product identity.** By cyclicity of trace,

$$\langle\tau_{\text{share}}^{(l)}, \tau_{\text{etf}}^{(l)}\rangle_F = \text{Tr}\big((\tau_{\text{share}}^{(l)})^\top\tau_{\text{share}}^{(l)}\,\mathcal{P}_{\text{ETF}}\big) \tag{12}$$

$$= \tfrac{C}{C-1}\,\|\tau_{\text{share}}^{(l)}\Pi_{\mathcal{S}}\|_F^2. \tag{13}$$

**Implication for alignment bounds.** Since $\|W_{\text{ETF}}\|_F^2 = \text{Tr}(W_{\text{ETF}}W_{\text{ETF}}^\top) = C$, any inequality of the form

$$\|\tau_{\text{share}}^{(l)}W_{\text{ETF}}^\top\|_F^2 \;\geq\; \gamma^2\,\|\tau_{\text{share}}^{(l)}\|_F^2\,\|W_{\text{ETF}}\|_F^2$$

implies

$$\gamma^2 \leq \frac{1}{C-1} \cdot \frac{\|\tau_{\text{share}}^{(l)} \Pi_{\mathcal{S}}\|_F^2}{\|\tau_{\text{share}}^{(l)}\|_F^2}.$$

In the best-aligned case (when $\tau_{\text{share}}^{(l)}$ lies entirely in $\mathcal{S}$), the universal bound is $\gamma = 1/\sqrt{C-1}$. $\qquad \square$

**Approximation to Ideal Parameters.**

*Proof.* Consider the decomposition of the approximation error:

$$\|\tau_{\text{ideal}}^{(l)} - \tau_{\text{etf}}^{(l)}\|_F^2 = \|\tau_{\text{ideal}}^{(l)} - \tau_{\text{share}}^{(l)} \mathcal{P}_{\text{ETF}}\|_F^2 \qquad (14)$$

$$= \|\tau_{\text{ideal}}^{(l)} - \tau_{\text{share}}^{(l)} + \tau_{\text{share}}^{(l)}(I - \tfrac{C-1}{C}\mathcal{P}_{\text{ETF}})\|_F^2, \qquad (15)$$

where we used the identity $\frac{C-1}{C}\mathcal{P}_{\text{ETF}} = \Pi_{\mathcal{S}}$ with $\Pi_{\mathcal{S}}$ the orthogonal projector onto the ETF subspace.

Expanding the squared norm:

$$= \|\tau_{\text{ideal}}^{(l)} - \tau_{\text{share}}^{(l)}\|_F^2 + \|\tau_{\text{share}}^{(l)}(I - \tfrac{C-1}{C}\mathcal{P}_{\text{ETF}})\|_F^2 \qquad (16)$$

$$+ 2\langle \tau_{\text{ideal}}^{(l)} - \tau_{\text{share}}^{(l)}, \; \tau_{\text{share}}^{(l)}(I - \tfrac{C-1}{C}\mathcal{P}_{\text{ETF}})\rangle. \qquad (17)$$

Although the canonical Neural Collapse analysis is often centered on the final classifier layer, subsequent empirical and theoretical studies indicate that ETF-like organization progressively manifests throughout intermediate layers (Parker et al., 2023). This phenomenon implies that each layer can be locally approximated by a near-ETF configuration within its effective representation subspace. From a dynamical perspective, the Neural Feature Ansatz (NFA) (Beaglehole et al., 2024a) further suggests that stochastic gradient descent implicitly aligns the singular vectors of the weight matrix with the evolving feature covariance. Such alignment enforces an emergent orthogonality and angular separation that mirrors ETF geometry. Therefore, assuming that the column space of each layer lies approximately within an ETF subspace is a reasonable and empirically supported simplification: it captures the intrinsic alignment dynamics observed across depth while providing a tractable analytic framework for our layerwise regularization design.

Under the assumption that $\tau_{\text{ideal}}^{(l)}$ exhibits near-ETF structure, its column space lies essentially in the ETF subspace. This implies

$$\langle \tau_{\text{ideal}}^{(l)}, \; \tau_{\text{share}}^{(l)}(I - \tfrac{C-1}{C}\mathcal{P}_{\text{ETF}})\rangle \approx 0. \qquad (18)$$

Therefore,

$$\langle \tau_{\text{ideal}}^{(l)} - \tau_{\text{share}}^{(l)}, \; \tau_{\text{share}}^{(l)}(I - \tfrac{C-1}{C}\mathcal{P}_{\text{ETF}})\rangle \approx -\|\tau_{\text{share}}^{(l)}(I - \tfrac{C-1}{C}\mathcal{P}_{\text{ETF}})\|_F^2. \qquad (19)$$

Substituting back yields

$$\|\tau_{\text{ideal}}^{(l)} - \tau_{\text{etf}}^{(l)}\|_F^2 \leq \|\tau_{\text{ideal}}^{(l)} - \tau_{\text{share}}^{(l)}\|_F^2 - g\,\|\tau_{\text{share}}^{(l)}(I - \tfrac{C-1}{C}\mathcal{P}_{\text{ETF}})\|_F^2, \qquad (20)$$

where $g \in (0, 1]$ quantifies the approximation quality of the directional correction assumption. $\qquad \square$

**Enhanced Direction Error Bound.**

*Proof.* We analyze the alignment residual $\tau_{\text{share}}^{(l)}(I - \mathcal{P}_{\text{ETF}})$. Using the alignment property:

$$\|\tau_{\text{share}}^{(l)}(I - \mathcal{P}_{\text{ETF}})\|_F^2 = \|\tau_{\text{share}}^{(l)}\|_F^2 - \|\tau_{\text{share}}^{(l)}\mathcal{P}_{\text{ETF}}\|_F^2 \qquad (21)$$

The alignment component satisfies:

$$\|\tau_{\text{share}}^{(l)}\mathcal{P}_{\text{ETF}}\|_F^2 = \text{Tr}((\tau_{\text{share}}^{(l)})^\top \tau_{\text{share}}^{(l)} \mathcal{P}_{\text{ETF}}) \qquad (22)$$

Since $\mathcal{P}_{\text{ETF}}$ has dimension at most $m = \min(d_{\text{in}}, d_{\text{out}} - 1)$, and using the spectral properties:

$$\|\tau_{\text{share}}^{(l)}\mathcal{P}_{\text{ETF}}\|_F^2 \leq m\|\tau_{\text{share}}^{(l)}\|_2^2 \qquad (23)$$

However, due to spectral alignment (from Part 1), we also have:

$$\|\tau_{\text{share}}^{(l)} \mathcal{P}_{\text{ETF}}\|_F^2 \geq \gamma \|\tau_{\text{share}}^{(l)}\|_F^2 \tag{24}$$

Combining these bounds with the stable rank $r_s = \|\tau_{\text{share}}^{(l)}\|_F^2 / \|\tau_{\text{share}}^{(l)}\|_2^2$:

$$\|\tau_{\text{share}}^{(l)}(I - \mathcal{P}_{\text{ETF}})\|_F^2 \geq \left(1 - \frac{m}{r_s}\right)(1 - \gamma)\|\tau_{\text{share}}^{(l)}\|_F^2 \tag{25}$$

$\square$

### B.2.2 GENERALIZATION ADVANTAGE

*Proof.* We rigorously establish the generalization advantage of ETF alignment through a geometric decomposition of the parameter space and Rademacher complexity analysis.

**ETF Structure and Orthogonal Decomposition.**

Let $\Pi_{\mathcal{S}}$ denote the orthogonal projector onto the ETF subspace $\mathcal{S} = \text{rowspan}(W_{\text{ETF}}) \subset \mathbb{R}^{d_{\text{out}}}$, which has dimension $C - 1$. The key property of the simplex ETF is that its Gram matrix satisfies:

$$\mathcal{P}_{\text{ETF}} = W_{\text{ETF}}^\top W_{\text{ETF}} = \frac{C}{C-1}\Pi_{\mathcal{S}}.$$

Define the orthogonally aligned parameter $\tau_\Pi = \tau_{\text{share}}\Pi_{\mathcal{S}}$. The ETF-aligned parameter can then be expressed as:

$$\tau_{\text{etf}} = \tau_{\text{share}}\mathcal{P}_{\text{ETF}} = \frac{C}{C-1}\tau_\Pi.$$

**Frobenius Norm Reduction.**

Since $\Pi_{\mathcal{S}}$ is an orthogonal projection, we have the Pythagorean decomposition:

$$\|\tau_{\text{share}}\|_F^2 = \|\tau_{\text{share}}\Pi_{\mathcal{S}}\|_F^2 + \|\tau_{\text{share}}(I - \Pi_{\mathcal{S}})\|_F^2 \tag{26}$$
$$= \|\tau_\Pi\|_F^2 + \|\tau_{\text{share}}(I - \Pi_{\mathcal{S}})\|_F^2. \tag{27}$$

This immediately implies $\|\tau_\Pi\|_F \leq \|\tau_{\text{share}}\|_F$. Moreover, the norm difference can be bounded as:

$$\|\tau_{\text{share}}\|_F - \|\tau_\Pi\|_F = \frac{\|\tau_{\text{share}}\|_F^2 - \|\tau_\Pi\|_F^2}{\|\tau_{\text{share}}\|_F + \|\tau_\Pi\|_F} \tag{28}$$
$$\geq \frac{\|\tau_{\text{share}}(I - \Pi_{\mathcal{S}})\|_F^2}{2\|\tau_{\text{share}}\|_F}. \tag{29}$$

**Spectral Characterization of the Residual.**

Let $\{v_1, \ldots, v_{C-1}, v_C\}$ be an orthonormal eigenbasis of $\Pi_{\mathcal{S}}$, where $v_1, \ldots, v_{C-1}$ span $\mathcal{S}$ (eigenvalue 1) and $v_C$ spans the orthogonal complement (eigenvalue 0). For the standard simplex ETF, $\mathcal{S}_{\text{ETF}} = \mathbf{1}^\perp$, and $v_C = \mathbf{1}/\sqrt{C}$. The residual norm becomes:

$$\|\tau_{\text{share}}(I - \Pi_{\mathcal{S}})\|_F^2 = \sum_{i=m+1}^{d_{\text{out}}} \|\tau_{\text{share}}v_i\|_2^2 = \|\tau_{\text{share}}v_C\|_2^2 = \left\|\tau_{\text{share}}\frac{\mathbf{1}\mathbf{1}^\top}{C}\right\|_F^2.$$

This quantifies the energy of $\tau_{\text{share}}$ outside the ETF subspace.

**Generalization Gap via Rademacher Complexity.**

We now invoke the following Rademacher complexity bound:

**Theorem 3** (Rademacher Bound for Gram-Aligned Models). *For a Lipschitz loss function with constant $L$ and bounded data with $\|x\|_2 \leq R$, with probability at least $1 - \delta$ over the training set of size $n$, the generalization gap satisfies:*

$$\mathcal{R}(\tau) - \widehat{\mathcal{R}}(\tau) \leq \frac{LR}{\sqrt{n}}\|\tau\|_F + c\sqrt{\frac{\log(1/\delta)}{n}}.$$

Applying this to both alignment schemes and noting that $\widehat{\mathcal{R}}(\tau_{\text{share}}) \approx \widehat{\mathcal{R}}(\tau_{\text{etf}})$ due to equivalent expressive power on training data, we obtain:

$$\mathcal{R}(\tau_{\text{share}}) - \mathcal{R}(\tau_{\text{etf}}) \lesssim \frac{LR}{\sqrt{n}} \left( \|\tau_{\text{etf}}\|_F - \|\tau_{\text{share}}\|_F \right) \tag{30}$$

$$= \frac{LR}{\sqrt{n}} \left( \frac{C}{C-1} \|\tau_{\Pi}\|_F - \|\tau_{\text{share}}\|_F \right). \tag{31}$$

Using the norm reduction and the scaling factor, we derive the main bound:

$$\mathcal{R}(\tau_{\text{share}}) - \mathcal{R}(\tau_{\text{etf}}) \leq \left( \frac{C}{C-1} \right)^2 \frac{LR}{\sqrt{n}} \frac{\|\tau_{\text{share}}(I - \Pi_{\mathcal{S}})\|_F^2}{\|\tau_{\text{share}}\|_F} + c\sqrt{\frac{\log(1/\delta)}{n}}. \tag{32}$$

To provide interpretable bounds, we observe that:

$$\|\tau_{\text{share}}(I - \Pi_{\mathcal{S}})\|_F^2 \leq \min \left\{ \|\tau_{\text{share}}\|_F^2, \frac{1}{r_s} \|\tau_{\text{share}}\|_F^2 \right\},$$

where $r_s = \|\tau_{\text{share}}\|_F^2 / \|\tau_{\text{share}}\|_2^2$ is the stable rank. This yields the practical bound:

$$\mathcal{R}(\tau_{\text{share}}) - \mathcal{R}(\tau_{\text{etf}}) \leq \left( \frac{C}{C-1} \right)^2 \frac{LR}{\sqrt{n}} \min \left\{ 1, \frac{1}{r_s} \right\} \|\tau_{\text{share}}\|_F + c\sqrt{\frac{\log(1/\delta)}{n}}.$$

When $\tau_{\text{share}}$ is class-centered ($\tau_{\text{share}}\mathbf{1} = 0$), we have $\tau_{\text{share}}(I - \Pi_{\mathcal{S}}) = 0$, and the generalization gap reduces to the confidence term, demonstrating the optimality of ETF alignment in this scenario.

$\square$

### B.3 FEATURE-SPACE DIRECTION ALIGNMENT

**Low-dimensional regime** (d $< C - 1$): In this case, the feature space lacks sufficient degrees of freedom to support class-level separation, leading to highly entangled representations. Optimizing fusion coefficients using standard cross-entropy loss yields suboptimal results compared to joint multi-task learning. However, by introducing an ETF-based alignment loss that enforces directional consistency with an equiangular tight frame (ETF), we observe substantial performance gains. This highlights the importance of directionality under low-dimensional constraints.

**High-dimensional regime** (d $\geq C - 1$): Here, the feature space is sufficiently expressive to allow class separation. Optimizing only the fusion coefficients already achieves competitive performance, and additional directional alignment provides marginal improvements. This suggests that direction constraints are less crucial in high-dimensional settings.

### B.3.1 GENERALIZATION ANALYSIS

**Theorem 4** (Rademacher Complexity Bound). *Let $\mathcal{H}_{rot}$ be the hypothesis space of rotation-merged models. For any $\delta > 0$, with probability at least $1 - \delta$:*

$$\mathcal{R}(\hat{h}) \leq \hat{\mathcal{R}}(\hat{h}) + 2\mathcal{R}_n(\mathcal{H}_{rot}) + \sqrt{\frac{\log(2/\delta)}{2n}} \tag{33}$$

*where the Rademacher complexity satisfies:*

$$\mathcal{R}_n(\mathcal{H}_{rot}) \leq \mathcal{O}\left( \sqrt{\frac{d^2 T \log(nT)}{n}} \right). \tag{34}$$

Theorem 4 establishes a generalization bound for direction-merged models through Rademacher complexity. Importantly, feature alignment reduces the effective hypothesis space by constraining representations onto coherent geometric structures, thereby lowering the capacity term $\mathcal{R}_n(\mathcal{H}_{rot})$. Compared with naive parameter aggregation, alignment eliminates redundant degrees of freedom across tasks, which tightens the bound from $\mathcal{O}\left( \sqrt{\frac{d^2 T \log(nT)}{n}} \right)$ toward a smaller effective complexity.

This indicates that feature alignment not only preserves empirical performance but also systematically improves generalization by reducing overfitting risks across tasks.

*Proof of Theorem 4.* **Hypothesis Space Decomposition.** Define:

$$\mathcal{H}_{\text{rot}} = \Big\{ \mathbf{x} \mapsto g\Big( \sum_{t=1}^{T} \theta_t^{\top} \mathbf{R}^t \phi(\mathbf{x}) \Big), \ \mathbf{R}^t \in SO(d) \Big\}.$$

**Covering Number Analysis.** The covering number of $SO(d)$ satisfies

$$\mathcal{N}(SO(d), \epsilon) \leq (C/\epsilon)^{d(d-1)/2}$$

for some universal constant $C$. The simplex constraint on task weights introduces additional logarithmic factors.

**Composition Bounds.** Using Lipschitz composition properties and a union bound over $T$ tasks:

$$\mathcal{R}_n(\mathcal{H}_{\text{rot}}) \leq \sum_{t=1}^{T} \mathcal{R}_n\big(\{\mathbf{x} \mapsto \theta_t^{\top} \mathbf{R}^t \phi(\mathbf{x})\}\big)$$

$$\leq T \cdot \mathcal{O}\Big( \sqrt{\tfrac{d^2}{n}} \Big) \cdot \sqrt{\log(nT)}$$

$$= \mathcal{O}\Big( \sqrt{\tfrac{d^2 T \log(nT)}{n}} \Big).$$

The logarithmic factor arises from the union bound and discretization of continuous parameter spaces. $\square$

## B.4 COMPUTATIONAL COMPLEXITY

### B.4.1 NOTATION

Table 3 summarizes the main symbols used in this section.

Table 3: Notation summary.

| Symbol | Meaning |
|---|---|
| $T$ | Number of tasks (models to be merged). |
| $L$ | Number of network layers in each model. |
| $n$ | Matrix dimension of each layer weight ($n \times n$ for simplicity). |
| $m$ | Smaller dimension for non-square layers ($m = \min(p, q)$ for a $p \times q$ matrix). |
| $\tau_t^{(\ell)}$ | Parameter matrix of task $t$ at layer $\ell$. |
| $U_t^{(\ell)}, \Sigma_t^{(\ell)}, V_t^{(\ell)}$ | Singular value decomposition (SVD) components of $\tau_t^{(\ell)}$. |
| $U_{\text{cat}}^{(\ell)}, V_{\text{cat}}^{(\ell)}$ | Concatenated matrices of top-$k$ components across tasks (for shared subspace). |
| $W_{\text{ETF}}$ | Equiangular Tight Frame (ETF) basis matrix. |
| $C$ | Number of classes (ETF dimension). |
| $\mathcal{O}(\cdot)$ | Big-O notation for asymptotic computational complexity. |

### B.4.2 COMPLEXITY OF ALGORITHM 1

For each layer $\ell = 1, \ldots, L$, Algorithm 1 performs three main computational steps.

**Per-task SVDs.** For each task $t = 1, \ldots, T$, the SVD

$$\tau_t^{(\ell)} = U_t^{(\ell)} \Sigma_t^{(\ell)} V_t^{(\ell)^{\top}}$$

requires $\mathcal{O}(n^3)$ time. The total cost across all layers and tasks is $\mathcal{O}(TLn^3)$.

**Concatenated SVDs.** After concatenating the top-$k$ components of all tasks into $U_{\text{cat}}^{(\ell)}$ and $V_{\text{cat}}^{(\ell)}$, two additional SVDs are performed. Each SVD costs $\mathcal{O}(n^3)$, yielding an additional $\mathcal{O}(2Ln^3)$.

**ETF projection.** The ETF projection,

$$\tau_{\text{etf}}^{(\ell)} = \tau_{\text{share}}^{(\ell)} W_{\text{ETF}}^{\top} W_{\text{ETF}},$$

involves two matrix multiplications of size $n \times C$ and $C \times n$, with per-layer cost $\mathcal{O}(n^2 C)$, or $\mathcal{O}(n^2 LC)$ in total.

**Overall complexity.** Combining all terms gives:

$$\boxed{\mathcal{O}\big((T+2)\,L\,n^3 + n^2 LC\big)}. \tag{35}$$

Since $C \ll n$ in most practical cases, the ETF term is negligible relative to the cubic SVD cost.

### B.4.3   COMPLEXITY COMPARISON WITH BASELINES

Table 4 compares our method with existing baselines in terms of their main SVD operations and asymptotic complexity per layer. Complexity expressions follow the standard cubic-time assumption for dense SVD.

Table 4: Computational complexity comparison per layer across methods. All methods involve $L$ layers with weight dimension $n \times n$.

| Method | Main SVD operations per layer | Total Complexity |
|---|---|---|
| ISO-C (Marczak et al., 2025) | One SVD on $\Delta_{\text{TA}}$ | $\mathcal{O}(Ln^3)$ |
| ISO-CTS (Marczak et al., 2025) | One SVD on $\Delta_{\text{TA}} + T$ on $\Delta_t$ + two orthogonalization SVDs | $\mathcal{O}((T+3)Ln^3)$ |
| TSV-M (Gargiulo et al., 2025) | $T$ task SVDs + two for reconstruction | $\mathcal{O}((T+2)Ln^3)$ |
| **Ours (Alg. 1)** | $T$ per-task SVDs + two concatenated SVDs + ETF projection | $\mathcal{O}((T+2)Ln^3 + n^2 LC)$ |

When $C \ll n$, our method scales similarly to TSV-M and Iso-CTS in asymptotic order, while introducing only a lightweight ETF projection step.

### B.4.4   DATA-BASED ALIGNMENT: EMPIRICAL OVERHEAD

When optimizing rotation matrices and fusion coefficients with data, the additional wall-clock cost per epoch is marginal (about $1-3\%$). The dominant extra FLOPs come from the Procrustes step (solving for the optimal rotation via SVD).

Table 5: Per-epoch timing breakdown (averaged across architectures). The Procrustes step accounts for a small fraction ($\sim 1$–$3\%$) of total training time.

| Model | Forward (s) | Backward (s) | Procrustes (s) | Epoch total (s) | Procrustes ratio |
|---|---|---|---|---|---|
| ViT-L/14 | 10.82 | 3.33 | 0.24 | $\approx 14.47$ | 1.7% |
| ViT-B/32 | 4.18 | 0.87 | 0.16 | $\approx 5.27$ | 3.0% |
| ViT-B/16 | 6.12 | 1.12 | 0.16 | $\approx 7.48$ | 2.2% |

**FLOPs analysis.** Table 6 reports the floating-point operation counts. The counters measure computational cost rather than loss components.

**Interpretation.** The alignment step, comprising rotation application and Procrustes optimization, adds negligible wall-clock time and a manageable FLOPs increase dominated by the SVD-based Procrustes computation.

Table 6: FLOPs of rotation and alignment components per epoch. The Procrustes step dominates the additional FLOPs cost.

| Model | rot_forward | rot_total | procrustes |
|---|---|---|---|
| ViT-L/14 | $3.78 \times 10^7$ | $1.13 \times 10^8$ | $9.06 \times 10^8$ |
| ViT-B/32 | $1.68 \times 10^7$ | $5.03 \times 10^7$ | $2.68 \times 10^8$ |
| ViT-B/16 | $1.68 \times 10^7$ | $5.03 \times 10^7$ | $2.68 \times 10^8$ |

## C  RELATED WORK

**Neural Collapse.** Recently, several studies (Ji et al., 2021; Zhu et al., 2021; Tirer & Bruna, 2022; Zhu et al., 2023; Li et al., 2023; Xie et al., 2023; Yang et al., 2023b; Beaglehole et al., 2024b; Fisher et al., 2024; Guo et al., 2024; Kothapalli et al., 2024; Súkeník et al., 2024; Chen et al., 2024) have utilized this phenomenon to guide the training process in imbalanced data sets. Among them, Yang et al. (2023b) introduce a continual learning framework that pre-allocates a fixed number of classes within a simplex ETF, thereby guiding the representation learning of minority classes in subsequent incremental steps. This design enforces intra-class feature convergence to predetermined positions while maximizing and uniformly separating inter-class features. To address class imbalance, Li et al. (2023) leverage its geometric structure under distributed conditions to align the representation directions across clients, while preserving client-specific characteristics through fine-tuning. The aforementioned methods only consider collapsing the same class onto a single point in a fixed simplex ETF without accounting for the impact of intra-class spurious correlations. Kothapalli (2022) theoretically analyze the generalization benefits induced by neural collapse and propose a generalization bound based on the Class Distance Normalized Variance (CDNV), which demonstrates that training-induced collapse can facilitate generalization, albeit requiring substantial data support. To the best of our knowledge, we are the first to revisit the importance of directional alignment through the lens of neural collapse, conducting the analysis from both the parameter space and the feature space.

## D  BASELINES

**Data-Free Methods**

Task Arithmetic (TA) (Ilharco et al., 2022) introduces a simple yet effective approach by treating task-specific knowledge as arithmetic operations in the parameter space. It computes task vectors as the difference between fine-tuned and pre-trained parameters, enabling direct manipulation through addition and negation operations to compose or forget specific capabilities.

Concrete Task Arithmetic (Tang et al., 2023) proposes a multi-task model merging method based on continuous relaxation of discrete (Concrete) subspace learning. The key idea is to identify common low-dimensional subspaces in the parameter space and exploit the shared information therein to effectively mitigate task interference in multi-task fusion, while preserving overall performance as much as possible. We formulate the problem as a bi-level optimization and introduce a meta-learning framework, where gradient-based techniques are employed to learn shared Concrete masks that guide model merging within the subspace. Experiments across multiple tasks in both vision and language domains validate the effectiveness of the proposed approach, showing substantial improvements over existing Task Arithmetic and AdaMerging methods.

Ties-Merging (Yadav et al., 2023), a novel method for merging multiple task-specific models into a single multitask model without additional training. Existing merging techniques suffer from performance degradation due to parameter interference caused by redundant values and sign conflicts across models. Ties-Merging addresses these issues through three key steps: trimming low-magnitude parameters, electing dominant signs, and disjointly merging only aligned values. Extensive experiments across NLP and vision domains, various model architectures, and fine-tuning settings demonstrate that Ties-Merging consistently outperforms prior methods, achieving significant gains in both in-domain and out-of-domain generalization. Our work highlights the critical role of resolving sign interference and provides a robust, hyperparameter-efficient solution for model merging.

Consensus Merging (Wang et al., 2024) consistently improves upon previous model merging techniques such as Task Arithmetic and TIES, achieving state-of-the-art performance. Additionally, our compression scheme using TALL-masks reduces storage requirements by over 85% while retaining 99.7% of the original performance, demonstrating a highly effective trade-off between model efficiency and multi-task capability.

AWD Task Arithmetic (Xiong et al., 2024) uses the following techniques by explicitly promoting orthogonality among task vectors. By decomposing task vectors into a shared redundant component and disentangled orthogonal components, AWD effectively reduces inter-task interference while preserving task-specific performance. When integrated with Task Arithmetic and AdaMerging, AWD achieves state-of-the-art results on multi-task benchmarks, demonstrating strong generalization and robustness across varying task numbers and scaling coefficients.

PCB-Merging (Du et al., 2024) proposes a novel, training-free model merging method, which effectively balances parameter-level competition across tasks via intra- and inter-task balancing mechanisms, leading to significant performance gains in cross-task, cross-domain, and out-of-domain generalization settings without requiring additional data or retraining.

Task Singular Vectors (TSV) (Gargiulo et al., 2025) proposes a novel model merging framework, which leverages singular value decomposition at the layer level to compress task-specific updates and reduce interference through a geometrically-informed whitening transformation, achieving state-of-the-art performance without additional training or validation data.

Isotropic Model Merging (ISO) (Marczak et al., 2025) proposes a novel model merging framework based on the analysis of the correlation between subspace alignment and model merging performance, which enhances alignment between task-specific and merged model subspaces by flattening the singular value spectrum of the merged task matrix and incorporating both common and task-specific directions.

DOGE Task Arithmetic (Wei et al., 2025) proposes a novel adaptive projective gradient descent framework for multi-task model merging, which formulates the problem as a constrained optimization objective to minimize performance gaps with individual task-specific models while preserving shared knowledge through subspace-constrained gradient updates and training-free task-aware merging coefficients.

**Data-Based Optimization Methods**

AdaMerging (Yang et al., 2023a) adaptively learns merging coefficients through gradient-based optimization on a small validation set. It introduces both task-wise and layer-wise coefficient learning strategies, allowing fine-grained control over the merging process based on actual task performance.

Concrete AdaMerging (Tang et al., 2023) extends AdaMerging with concrete masks, combining the benefits of adaptive coefficient learning with parameter-level masking. This dual optimization approach achieves superior performance by simultaneously learning what and how much to merge.

Representation Surgery (Yang et al., 2024) operates at the representation level rather than parameter space. It aligns and merges intermediate representations across models using surgical precision, employing optimal transport and feature matching techniques to ensure semantic consistency.

AWD AdaMerging (Xiong et al., 2024) integrates Adaptive Weight Disentanglement with AdaMerging's optimization framework. This combination leverages both the structural decomposition of AWD and the data-driven optimization of AdaMerging for enhanced performance.

DOGE AdaMerging (Wei et al., 2025) extends the DOGE framework to the data-based setting, utilizing validation data to guide the discrete optimization process. It employs advanced search strategies and ensemble techniques to discover optimal merging configurations that maximize multi-task performance.

These baselines represent the current state-of-the-art in model merging, spanning from simple arithmetic operations to sophisticated optimization frameworks, providing a comprehensive benchmark for evaluation.

# E  DATASETS

For vision tasks, we evaluate our approaches over three benchmark suites comprising 8, 14, and 20 tasks, respectively. The first suite, introduced in (Radford et al., 2021), consists of eight datasets: Stanford Cars (Krause et al., 2013), DTD (Cimpoi et al., 2014), EuroSAT (Helber et al., 2019), GTSRB (Stallkamp et al., 2011), MNIST (LeCun et al., 2002), RESISC45 (Cheng et al., 2017), SUN397 (Xiao et al., 2016), and SVHN (Netzer et al., 2011).

The 14-task benchmark extends the above by incorporating six additional datasets: CIFAR-100 (Krizhevsky et al., 2009), STL10 (Coates et al., 2011), Flowers102 (Nilsback & Zisserman, 2008), Oxford-IIIT Pet (Parkhi et al., 2012), PCAM (Veeling et al., 2018), and FER2013 (Goodfellow et al., 2013).

Finally, the 20-task benchmark includes all the previous 14 tasks plus six more: EMNIST (Cohen et al., 2017), CIFAR-10 (Krizhevsky et al., 2009), Food101 (Bossard et al., 2014), Fashion-MNIST (Xiao et al., 2017), Rendered-SST2 (Socher et al., 2013), and KMNIST (Clanuwat et al., 2018). To investigate the effect of model capacity, we evaluate our method using three CLIP (Radford et al., 2021) variants, each employing a different ViT (Dosovitskiy et al., 2020) visual encoder: ViT-B/32, ViT-B/16, and ViT-L/14.

SUN397: A large-scale scene recognition dataset containing 397 categories with over 100K images. The dataset covers a diverse range of indoor and outdoor scenes, such as churches, bedrooms, highways, and landscapes, making it one of the most comprehensive scene benchmarks. It is widely used to evaluate the generalization ability of visual recognition models, especially in transfer learning and domain generalization studies.

Stanford Cars: A fine-grained object classification dataset with 196 car models and 16K images. Each image is annotated with the make, model, and year of the car, requiring models to distinguish between visually similar subcategories. It is a standard benchmark for fine-grained recognition and is often used to evaluate representation learning, attention mechanisms, and metric learning methods.

RESISC45: A remote sensing image scene classification dataset comprising 31.5K images across 45 categories. Images are collected from Google Earth, covering diverse scenes such as residential areas, airports, rivers, and forests. The dataset is designed to evaluate robustness under varying imaging conditions (viewpoints, illumination, resolutions), and is widely used in remote sensing and geospatial vision research.

EuroSAT: A remote sensing dataset based on Sentinel-2 satellite imagery, containing 27K images across 10 land-use and land-cover categories, such as industrial areas, forests, and highways. With multispectral bands available, it provides a valuable benchmark for both RGB and multispectral classification tasks. EuroSAT is widely used for studying domain adaptation, few-shot learning, and environmental monitoring applications.

SVHN: The Street View House Numbers dataset, with 73K training images and 10 digit categories, extracted from Google Street View. Unlike MNIST, SVHN features digits in natural scene images, often with cluttered backgrounds, varying scales, and illumination conditions. It is considered a more challenging real-world digit recognition benchmark and is widely used to test the robustness of deep learning models.

GTSRB: The German Traffic Sign Recognition Benchmark, containing 50K images across 43 traffic sign categories. Images vary significantly in lighting, perspective, and occlusion, making it highly representative of real-world autonomous driving conditions. GTSRB is a standard benchmark for evaluating models in traffic sign recognition, robustness under distribution shifts, and safety-critical perception tasks.

MNIST: A canonical digit classification dataset with 70K grayscale images of handwritten digits (0–9). As one of the earliest benchmarks in computer vision, MNIST remains a widely used baseline for algorithm prototyping and teaching, despite being relatively easy for modern deep learning models. It has inspired many extended datasets (e.g., EMNIST, Fashion-MNIST, KMNIST).

DTD: The Describable Textures Dataset, containing 5.6K images across 47 categories, each corresponding to a human-describable texture attribute (e.g., "bumpy," "striped," "zigzagged"). Unlike

object or scene classification datasets, DTD emphasizes fine-grained texture perception, making it useful for evaluating mid-level representations and generalization across visual domains.

Flowers102: A fine-grained flower recognition dataset with 102 categories and 8K images. Each category corresponds to a specific flower species, with large intra-class variation due to differences in viewpoint, lighting, and scale. Flowers102 is frequently used for transfer learning and fine-grained classification, particularly in evaluating few-shot and zero-shot learning algorithms.

PCAM: The PatchCamelyon dataset for histopathology image classification, containing 327K image patches with binary labels (tumor vs. normal). Derived from the CAMELYON16 challenge, it is widely used in medical imaging benchmarks to evaluate weakly supervised learning, multiple-instance learning, and robustness in clinical decision support systems.

FER2013: A facial expression recognition dataset with 35.8K grayscale images across 7 emotion categories (e.g., happy, sad, angry, surprised). Images were collected in the wild from the ICML 2013 Kaggle competition, making the dataset challenging due to variations in pose, illumination, and occlusion. It is widely used for affective computing and human-computer interaction research.

Oxford-IIIT Pet: A fine-grained object dataset containing 37 pet breeds with 7K images. Each image includes both class labels and pixel-level segmentation masks, enabling tasks in both classification and semantic segmentation. The dataset is commonly used for fine-grained recognition and multi-task learning in vision.

STL-10: A dataset similar to CIFAR but with higher-resolution images (96×96), containing 10 categories and 13K labeled images. In addition to the labeled set, it includes a large set of 100K unlabeled images, making it well-suited for semi-supervised and unsupervised representation learning. STL-10 is often used to evaluate scalable learning methods.

CIFAR-100: A widely used benchmark dataset with 100 fine-grained categories and 60K images. Each category has 500 training and 100 testing images, requiring models to handle limited data per class. CIFAR-100 is considered more challenging than CIFAR-10 and is a standard benchmark for evaluating the scalability of deep learning models.

CIFAR-10: A subset of CIFAR with 10 categories and 60K images. It remains one of the most widely used datasets for small-scale object classification, serving as a testbed for model architectures, optimization strategies, and regularization techniques.

Food-101: A large-scale food recognition dataset with 101 categories and 101K images. Each category represents a distinct food dish, with significant variation in presentation and context. It is a benchmark for fine-grained recognition, domain adaptation, and applications in food computing and lifestyle analysis.

Fashion-MNIST: A drop-in replacement for MNIST, containing 70K grayscale images from 10 clothing categories such as T-shirts, trousers, and coats. Compared to MNIST, Fashion-MNIST provides a more challenging benchmark while maintaining the same data format, making it a popular choice for evaluating lightweight models.

EMNIST: The Extended MNIST dataset, comprising 800K images across up to 62 categories (digits and uppercase/lowercase letters). It provides multiple splits (balanced, letters, digits, etc.), making it a versatile benchmark for handwritten character recognition and multi-class classification tasks.

KMNIST: A dataset of 70K grayscale images from 10 categories of handwritten Japanese characters (Kuzushiji). As a more challenging alternative to MNIST, KMNIST captures complex character structures and is used to evaluate transfer learning, few-shot learning, and robustness across different scripts.

Rendered SST-2: A rendered version of the Stanford Sentiment Treebank (SST-2), where binary sentiment labels (positive vs. negative) are transformed into visual classification tasks through rendered images. It bridges NLP and computer vision by enabling cross-modal evaluation, particularly in vision-language pretraining and multimodal learning research.

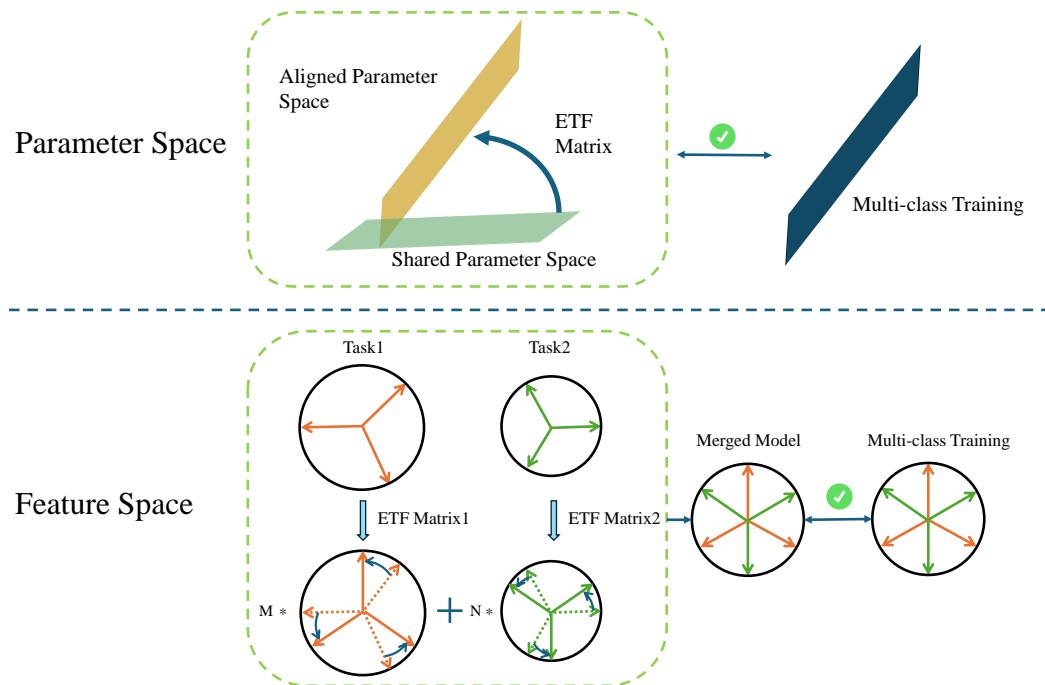

Figure 4: **Top Figure:** Description of our approach for directional alignment in the parameter space. **Bottom Figure:** Description of our approach for directional alignment in the feature space.

## F METHOD

As illustrated in Figure 4, We provide a schematic illustration of our proposed method. In the parameter space, we leverage an Equiangular Tight Frame (ETF) to enforce directional alignment across the subspaces associated with different classes, thereby establishing a consistent geometric structure among task-specific parameters. In the feature space, each task is assigned an initialized alignment matrix that guides the orientation of its representations. During training, we jointly optimize both the alignment matrices and the fusion coefficients, which allows the model to adaptively refine the directions and magnitudes of the merged representations. This joint optimization yields a more coherent alignment across tasks and ultimately leads to improved performance and generalization.

## G PERFORMANCE

### G.1 IMPLEMENTATION DETAILS

To assess the role of directional structure in the parameter space, we remove residual scale effects and align the merged parameters with a simplex equiangular tight frame (ETF), thereby enforcing global directional consistency. Meanwhile, to evaluate the importance of feature-space alignment, we draw upon the theory of neural collapse. Specifically, we constrain the penultimate-layer features to align with a simplex ETF, while simultaneously aligning the classifier prototypes to the same simplex structure. This joint constraint, optimized with a learning rate of $1 \times 10^{-3}$, updates both the feature rotation matrix and the fusion coefficients, thereby enforcing a coherent geometric configuration that not only validates the importance of direction but also facilitates faster convergence and improved generalization.

Following standard practice in model merging, we adopt the experimental setups used in TSV (Gargiulo et al., 2025) and AdaMerging (Yang et al., 2023a) for a fair comparison under both data-free and data-based regimes.

Data-free setting (Algorithm 1). In the data-free regime, our parameter-space alignment experiments employ exactly the same data splits as those used by TSV. Each task corresponds to an independently

Table 7: Multi-task performance when merging ViT-B/32 models on 8-task vision benchmark.

| Method | SUN397 | Cars | RESISC45 | EuroSAT | SVHN | GTSRB | MNIST | DTD | Avg. |
|---|---|---|---|---|---|---|---|---|---|
| **Non-merging Methods** | | | | | | | | | |
| Pre-trained | 62.3 | 59.7 | 60.7 | 45.5 | 31.4 | 32.6 | 48.5 | 43.8 | 48.0 |
| Individual | 79.2 | 77.7 | 96.1 | 99.7 | 97.5 | 98.7 | 99.7 | 79.4 | 90.8 |
| Traditional MTL | 73.9 | 74.4 | 93.9 | 98.2 | 95.8 | 98.9 | 99.5 | 77.9 | 88.9 |
| **Data-free Methods** | | | | | | | | | |
| Task Arithmetic | 55.2 | 54.9 | 66.7 | 78.9 | 80.2 | 69.7 | 97.3 | 50.4 | 69.1 |
| Ties-Merging | 59.8 | 58.6 | 70.7 | 79.7 | 86.2 | 72.1 | 98.3 | 54.2 | 72.4 |
| Consensus Merging | 65.7 | 63.6 | 76.5 | 77.2 | 81.7 | 70.3 | 97.0 | 57.1 | 73.6 |
| AWD TA | 63.5 | 61.9 | 72.6 | 84.9 | 85.1 | 79.1 | 98.1 | 56.7 | 75.2 |
| PCB-Merging | 66.7 | 65.5 | 78.5 | 79.3 | 86.4 | 77.1 | 98.2 | 59.1 | 76.3 |
| Concrete TA | 62.5 | 61.1 | 76.0 | 95.7 | 91.0 | 81.9 | 98.5 | 51.9 | 77.3 |
| TSV TA | 67.7 | 70.8 | 87.3 | 95.9 | 93.6 | 94.9 | 84.7 | 86.9 | 85.2 |
| ISO TA | 72.1 | **73.8** | 88.5 | 91.3 | 81.4 | 90.0 | 80.1 | 86.0 | 82.9 |
| ISO-CLS TA | **72.2** | 73.7 | 87.1 | 88.5 | 75.8 | 86.3 | 78.9 | 83.9 | 80.8 |
| DOGE TA | 67.7 | 70.1 | 82.0 | 90.3 | 86.3 | 86.8 | 98.3 | 64.0 | 80.7 |
| MDA TA | 70.4 | 71.9 | **88.9** | 96.7 | 94.3 | 95.4 | 85.3 | 88.6 | 86.4 |
| **Data-based Optimization Methods** | | | | | | | | | |
| AdaMerging | 64.5 | 68.1 | 79.2 | 93.8 | 87.0 | 91.9 | 97.5 | 59.1 | 80.1 |
| AdaMerging++ | 66.6 | 68.3 | 82.2 | 94.2 | 89.6 | 89.0 | 98.3 | 60.6 | 81.1 |
| Representation Surgery | 63.8 | 59.9 | 83.3 | **97.9** | 87.0 | 87.0 | 98.6 | 69.4 | 80.9 |
| AWD AM | 68.1 | 71.4 | 83.4 | 94.8 | 87.7 | 93.6 | 97.9 | 66.1 | 82.9 |
| Concrete AM | 67.8 | 70.0 | 87.5 | 96.0 | 91.6 | **96.7** | 98.7 | 63.8 | 84.0 |
| TSV AM | 68.1 | 72.1 | 87.7 | 96.4 | 92.3 | 93.3 | **99.4** | 88.9 | 87.3 |
| DOGE AM | **70.5** | **74.8** | 88.7 | 94.1 | 91.6 | 95.7 | 98.8 | 72.5 | 85.9 |
| MDA AM | 70.0 | 72.0 | **89.1** | 97.6 | **93.7** | 95.1 | **99.4** | **91.8** | **88.6** |

Table 8: Multi-task performance when merging ViT-B/32 models on 8-task vision benchmark.

| Method | SUN397 | Cars | RESISC45 | EuroSAT | SVHN | GTSRB | MNIST | DTD | Avg. |
|---|---|---|---|---|---|---|---|---|---|
| TSV TA | 67.2 | 70.3 | 85.7 | 94.5 | 91.6 | 92.2 | 99.3 | 84.8 | 85.7 |
| ISO TA | 71.8 | 73.8 | 87.7 | 90.9 | 82.3 | 88.1 | 98.2 | 86.0 | 84.9 |
| ISO-CLS TA | 72.4 | 74.0 | 87.9 | 90.6 | 76.9 | 85.3 | 97.5 | 85.5 | 83.8 |
| MDA TA | 70.3 | 72.0 | 87.9 | 96.0 | 92.5 | 93.3 | 99.3 | 87.5 | 87.4 |

fine-tuned CLIP model on one of the datasets from the 8-, 14-, and 20-task. No additional data are used during merging; only the fine-tuned model weights are required.

Data-based setting (Algorithm 2). For the adaptive (feature-space) alignment, we follow the same protocol as AdaMerging and utilize the unlabeled validation/test splits provided for each dataset as the unsupervised set to optimize the rotation matrices and fusion coefficients. This set corresponds to the same unlabeled data employed by AdaMerging for entropy-based coefficient adaptation.

All experiments therefore share identical dataset partitions across baselines, ensuring direct comparability. Specifically, the 8-, 14-, and 20-task benchmarks include datasets such as Cars, DTD, EuroSAT, RESISC45, SVHN, GTSRB, MNIST, SUN397, and progressively expanded subsets up to 20 tasks, matching the configurations reported in TSV. We emphasize that no labeled data are used in the data-free setting, while the data-based setup leverages only the same unlabeled validation data as AdaMerging for feature-space optimization.

## G.2 RESULTS

### G.2.1 MULTI-TASK PERFORMANCE WHEN MERGING VIT-B/32 MODELS ON 8-TASK VISION BENCHMARK

We conducted experiments on eight tasks using the ViT-B-32 architecture, based on the checkpoints provided by (Marczak et al., 2025; Gargiulo et al., 2025). The results are summarized in Table 8.

Table 7 presents a comprehensive evaluation of multi-task performance when merging ViT-B/32 models across eight vision datasets. The results clearly demonstrate that incorporating feature-space alignment to optimize task-wise fusion coefficients yields consistent performance improvements

across a variety of tasks. Our method, Ours AM, achieves the highest average accuracy of 88.6%, surpassing all baseline merging methods, including TSV AM (85.8%) and DOGE AM (85.9%).

Table 9: Ablation study on multi-task performance when merging ViT-B/32 models on 8-task vision benchmark.

| Method | SUN397 | Cars | RESISC45 | EuroSAT | SVHN | GTSRB | MNIST | DTD | Avg. |
|---|---|---|---|---|---|---|---|---|---|
| Ours TA | **70.4** | 71.9 | 88.9 | 96.7 | **94.3** | **95.4** | 85.3 | 88.6 | 86.4 |
| Ours w/o rotation AM | 68.1 | **72.1** | 87.7 | 96.4 | 92.3 | 93.3 | **99.4** | 88.9 | 87.3 |
| Ours AM | 70.0 | 72.0 | **89.1** | **97.6** | 93.7 | 95.1 | **99.4** | **91.8** | **88.6** |

Table 9 reports the ablation results on ViT-B/32 across eight vision benchmarks. Our rotation-aware alignment method (Ours AM) achieves the highest average accuracy of 88.6%, outperforming both the variant without rotation alignment (Ours w/o rotation AM, 87.3%) and the task-arithmetic baseline (Ours TA, 86.4%). The improvement is particularly pronounced on challenging datasets such as DTD (91.8% vs. 88.9%) and RESISC45 (89.1% vs. 87.7%), where feature distributions are highly heterogeneous. This result highlights that incorporating rotation alignment is crucial for preserving feature geometry across tasks.

### G.2.2 Multi-task performance when merging ViT-L/14 models on 8-task vision benchmark

Table 10: Multi-task performance when merging ViT-L/14 models on 8-task vision benchmark.

| Method | SUN397 | Cars | RESISC45 | EuroSAT | SVHN | GTSRB | MNIST | DTD | Avg. |
|---|---|---|---|---|---|---|---|---|---|
| **Non-merging Methods** | | | | | | | | | |
| Pre-trained | 66.8 | 77.7 | 71.0 | 59.9 | 58.4 | 50.5 | 76.3 | 55.3 | 64.5 |
| Individual | 82.3 | 92.4 | 97.4 | 100 | 98.1 | 99.2 | 99.7 | 84.1 | 94.2 |
| Traditional MTL | 80.8 | 90.6 | 96.3 | 96.3 | 97.6 | 99.1 | 99.6 | 84.4 | 93.5 |
| **Data-free Methods** | | | | | | | | | |
| Task Arithmetic | 73.9 | 82.1 | 86.6 | 94.1 | 87.9 | 86.7 | 98.9 | 65.6 | 84.5 |
| Ties-Merging | 76.5 | 85.0 | 89.3 | 95.7 | 90.3 | 83.3 | 99.0 | 68.8 | 86.0 |
| Consensus Merging | 75.0 | 84.3 | 89.4 | 95.6 | 88.3 | 82.4 | 98.9 | 68.0 | 85.2 |
| AWD TA | 76.2 | 85.4 | 88.7 | 96.1 | 92.4 | 92.3 | 99.3 | 69.4 | 87.5 |
| PCB-Merging | 76.8 | 86.2 | 89.4 | 96.5 | 88.3 | 91.0 | 98.6 | 73.6 | 87.5 |
| Concrete TA | **86.2** | 66.9 | **96.7** | 93.4 | **99.1** | 89.0 | 74.6 | **93.6** | 87.4 |
| TSV TA | 78.3 | 90.0 | 94.1 | 98.5 | 95.2 | 96.1 | 99.5 | 78.9 | 91.3 |
| ISO AM | 79.9 | 90.9 | 94.8 | 98.3 | 91.4 | 95.5 | 99.2 | 79.2 | 91.1 |
| DOGE TA | 76.7 | 87.7 | 91.6 | 96.2 | 94.4 | 93.4 | 98.9 | 71.6 | 88.8 |
| MDA TA | 79.1 | **90.6** | 94.7 | **98.7** | 95.6 | **97.0** | 99.6 | 80.4 | **92.0** |
| **Data-based Optimization Methods** | | | | | | | | | |
| AdaMerging | 79.0 | 90.3 | 90.8 | 96.2 | 93.4 | 98.0 | 99.0 | 79.9 | 90.8 |
| AdaMerging++ | 79.4 | 90.3 | 91.6 | 97.4 | 93.4 | 97.5 | 99.0 | 79.2 | 91.0 |
| Representation Surgery | 75.7 | 84.4 | 93.1 | 98.8 | 91.3 | 93.4 | 99.1 | 76.1 | 89.0 |
| AWD AM | **79.8** | 90.6 | 91.8 | 97.0 | 93.9 | **98.4** | 99.2 | 81.1 | 91.5 |
| Concrete AM | 77.8 | **91.2** | 92.1 | 97.0 | 94.4 | 97.9 | 99.0 | 79.5 | 91.1 |
| MDA AM | 79.1 | 91.0 | **94.7** | **98.9** | **95.9** | 97.9 | **99.7** | **84.0** | **92.7** |

Table 10 presents results for ViT-L/14. In the data-free regime, our method (Ours TA) achieves an average accuracy of 92.0%, outperforming strong baselines such as TSV (91.3%) and AWD TA (87.5%). In the optimization-based regime, Ours AM further improves performance to 92.7%, surpassing AdaMerging (90.8%) and AWD AM (91.5%). Notably, on EuroSAT, Ours AM achieves 98.9%, which is higher than TSV (98.5%) and AdaMerging (96.2%). These results indicate that our alignment mechanism generalizes well to large-scale backbones and heterogeneous tasks.

Table 12 further confirms the benefit of rotation alignment on ViT-L/14. Ours AM achieves the best average performance of 92.7%, compared to Ours w/o rotation AM (92.3%) and Ours TA (92.0%). Although the gains appear smaller than in ViT-B/32, the improvements are consistent across all tasks (e.g., DTD: 84.0% vs. 83.1%, EuroSAT: 98.9% vs. 98.7%), showing that rotation alignment stabilizes feature fusion and ensures reliable improvements even at larger scales.

We conducted experiments on eight tasks using the ViT-L-14 architecture, based on the checkpoints provided by (Marczak et al., 2025; Gargiulo et al., 2025). The results are summarized in Table 11.

Table 11: Multi-task performance when merging ViT-B/14 models on 8-task vision benchmark.

| Method | SUN397 | Cars | RESISC45 | EuroSAT | SVHN | GTSRB | MNIST | DTD | Avg. |
|---|---|---|---|---|---|---|---|---|---|
| TSV TA | 77.8 | 89.8 | 93.5 | 98.7 | 94.7 | 96.1 | 99.5 | 93.2 | 92.9 |
| ISO TA | 79.9 | 91.4 | 94.8 | 99.0 | 90.5 | 95.5 | 99.3 | 96.3 | 93.3 |
| ISO-CLS TA | 79.6 | 91.7 | 94.6 | 98.8 | 88.8 | 95.4 | 99.2 | 95.6 | 92.9 |
| MDA TA | 78.8 | 90.7 | 94.3 | 99.2 | 95.2 | 97.0 | 99.6 | 94.5 | 93.6 |

Table 12: Ablation study on multi-task performance when merging ViT-L/14 models on 8-task vision benchmark.

| Method | SUN397 | Cars | RESISC45 | EuroSAT | SVHN | GTSRB | MNIST | DTD | Avg. |
|---|---|---|---|---|---|---|---|---|---|
| MDA TA | 79.1 | 90.6 | **94.7** | 98.7 | 95.6 | 97.0 | 99.6 | 80.4 | 92.0 |
| MDA w/o rotation AM | 78.5 | 90.8 | 94.0 | 98.7 | 95.9 | 97.7 | 99.6 | 83.1 | 92.3 |
| MDA AM | **79.1** | **91.0** | **94.7** | **98.9** | **95.9** | **97.9** | **99.7** | **84.0** | **92.7** |

### G.2.3 ABLATION ON THE RANK DIMENSION $k$.

When $k = d_{\text{out}}/T$, the concatenated dimension $kT$ naturally equals $d_{\text{out}}$, so the shared representation $\tau_{\text{share}}^{(l)}$ directly matches the layer's output dimension without requiring any truncation. More generally, when $k$ varies across tasks or $kT \neq d_{\text{out}}$, the formulation still holds:

1. If $kT > d_{\text{out}}$, we simply truncate $U_{\text{share}}^{(l)}$ to its top-$d_{\text{out}}$ singular components, preserving the directions that explain the highest shared variance.

2. If $kT < d_{\text{out}}$, the missing dimensions can be completed either by zero-padding or implicitly through the subsequent ETF projection.

We investigate the impact of the subspace dimension $k$ on model merging performance. Recall that $k$ controls the number of principal components retained per task in the shared subspace construction. By default, we set $k = d_{\text{out}}/T$, where $d_{\text{out}}$ is the output dimension of each layer and $T$ is the number of tasks being merged. This choice ensures that the concatenated dimension $kT$ naturally matches $d_{\text{out}}$, leading to a balanced contribution from each task without requiring additional normalization or scaling.

Table 14 reports the results of varying $k$ across different ViT architectures and task counts. We observe that the performance remains stable near $k = d_{\text{out}}/T$, with slight fluctuations depending on the total number of tasks. Specifically, Smaller $k$ values (e.g., $k = 0.01\,d_{\text{out}}$) underrepresent task-specific subspaces and lead to noticeable drops in accuracy, while excessively large $k$ values (e.g., $k = 0.5\,d_{\text{out}}$ or $0.9\,d_{\text{out}}$) introduce redundancy and degrade generalization. Interestingly, when $k$ is chosen close to $d_{\text{out}}/T$, the model achieves a favorable trade-off between shared and task-specific components, and in some cases even surpasses the performance of the setting with $k = d_{\text{out}}/T$. In most configurations, the setting $k = d_{\text{out}}/T$ achieves a favorable trade-off between efficiency and accuracy, and is therefore adopted as the default in all main results, consistent with prior work (Gargiulo et al., 2025). MDA TA 0.01 means that we assign the top $1\%$ proportion of feature dimensions $d_{\text{out}}$ as the principal components for each task.

### G.2.4 MULTI-TASK PERFORMANCE WHEN MERGING FLAN-T5-BASE (LORA FINE-TUNED) MODELS ON ALL EIGHT TASKS.

Table 13 reports multi-task performance on Flan-T5-base (LoRA fine-tuned) across eight NLP benchmarks. Our method achieves the highest average score of 82.1% among data-free methods, outperforming TSV (81.8%) and DoGE (79.9%). For instance, on STSB, Ours TA obtains 84.7%, substantially higher than TSV (82.5%) and Ties-Merging (71.2%). Compared to optimization-based methods, Ours TA (82.1%) also surpasses AdaMerging++ (78.3%) and Concrete AM (78.5%) *without access to training data*, demonstrating the effectiveness of our approach in data-free language model merging.

Across both vision and NLP benchmarks, our results demonstrate that *directional alignment* is a key factor in successful model merging. Traditional methods often suffer from feature misalignment: when task-specific representations are merged without considering their geometric orientation, the

Table 13: Multi-task performance when merging Flan-T5-base (LoRA fine-tuned) models on all eight tasks.

| Method | CoLA | MNLI | MRPC | QNLI | QQP | RTE | SST2 | STSB | Avg. |
|---|---|---|---|---|---|---|---|---|---|
| **Non-merging Methods** | | | | | | | | | |
| Individual | 69.1 | 82.7 | 85.5 | 90.9 | 84.0 | 84.4 | 92.9 | 87.4 | 84.6 |
| **Data-free Methods** | | | | | | | | | |
| Weight Averaging | **69.7** | 59.7 | 78.9 | 90.1 | 83.8 | 80.5 | 91.2 | 72.0 | 78.2 |
| Task Arithmetic | 68.8 | 55.2 | 78.7 | 89.8 | 83.7 | 79.1 | 91.5 | 72.4 | 77.4 |
| Ties-Merging | 68.3 | 56.3 | 79.2 | 89.8 | 83.7 | 79.4 | 91.6 | 71.2 | 77.5 |
| Concrete TA | 69.1 | 58.1 | 78.4 | 89.9 | 83.5 | 79.4 | 91.6 | 73.4 | 78.0 |
| TSV TA | 69.3 | **77.1** | 80.4 | 90.0 | 83.6 | 79.1 | 92.5 | 82.5 | 81.8 |
| ISO TA | 69.1 | 57.4 | 76.7 | 88.6 | 82.7 | 80.1 | 91.3 | 63.3 | 76.2 |
| DoGE TA | 69.1 | 71.9 | **80.9** | **90.3** | 83.5 | 79.8 | **92.5** | 71.1 | 79.9 |
| MDA TA | 69.5 | 76.9 | 76.9 | 89.6 | **83.9** | **82.4** | 92.5 | **84.7** | **82.1** |
| **Data-based Optimization Methods** | | | | | | | | | |
| AdaMerging++ | 69.1 | 60.3 | 78.4 | 90.0 | 83.6 | 79.1 | 91.6 | 74.1 | 78.3 |
| Concrete AM | 69.0 | 59.4 | 80.1 | 89.9 | 82.9 | 79.1 | 91.7 | 75.4 | 78.5 |

Table 14: Ablation study on the subspace dimension $k$ for different ViT architectures and numbers of tasks.

| Method | ViT-B/32 | | | ViT-B/16 | | | ViT-L/14 | | |
|---|---|---|---|---|---|---|---|---|---|
| | 8 tasks | 14 tasks | 20 tasks | 8 tasks | 14 tasks | 20 tasks | 8 tasks | 14 tasks | 20 tasks |
| **MDA TA** $(1/T)$ | 86.4 | 81.4 | **77.3** | 89.9 | 85.8 | **84.7** | 92.0 | **89.4** | **88.4** |
| MDA TA 0.01 | 77.9 | 76.3 | 72.6 | 82.0 | 80.5 | 78.2 | 87.8 | 86.6 | 85.4 |
| MDA TA 0.06 | 86.1 | **81.9** | 77.1 | 90.1 | **86.1** | 82.5 | 91.9 | 89.3 | 88.2 |
| MDA TA 0.1 | **86.7** | 81.2 | 75.9 | **90.6** | 85.5 | 81.1 | **92.1** | 89.3 | 87.1 |
| MDA TA 0.5 | 77.8 | 70.6 | 66.7 | 83.4 | 76.3 | 73.0 | 87.9 | 82.9 | 79.5 |
| MDA TA 0.9 | 70.5 | 66.8 | 64.3 | 78.3 | 72.9 | 70.5 | 84.2 | 79.8 | 76.9 |

resulting model exhibits degraded generalization, especially on heterogeneous datasets. Our method aligns task vectors via rotation, which brings two major benefits: (1) it preserves the relative geometry of task-specific features, leading to more coherent shared representations; (2) it reduces destructive interference between tasks, as misaligned directions in parameter space are corrected before fusion. This is empirically validated by the consistent improvements across backbones (ViT-B/32 and ViT-L/14) and modalities (vision and NLP). These findings suggest that *geometric alignment is not merely a technical detail but a fundamental principle* for effective model merging.

### G.2.5 EFFECT OF CLASS-TO-DIMENSION RATIO: STRONGER GAINS IN VISION THAN IN LANGUAGE TASKS

**Difference in class granularity and dimensional ratios.** In vision experiments, each benchmark suite contains **8 (758 classes)**, **14 (1016 classes)**, or **20 (1306 classes)** tasks, with the corresponding number of categories ranging from approximately **758 to over 1,300 classes in total**. For models such as **ViT-B/32 ($d_{\mathbf{out}}$ is typically equal to 768)**, **ViT-B/16 ($d_{\mathbf{out}}$ is typically equal to 768)**, and **ViT-L/14 ($d_{\mathbf{out}}$ is typically equal to 1024)**, the total number of classes *exceeds* the feature dimensionality. In this regime ($C > d_{\mathbf{out}}$), enforcing balanced angular separation between classes becomes geometrically nontrivial, and **directional alignment via ETF** plays a critical role in achieving near-uniform separation across class directions. Therefore, ETF alignment provides clear structural and generalization benefits for vision models, where the class-to-dimension ratio is high.

**Smaller class space in NLP tasks.** In contrast, the NLP setting (based on **eight GLUE tasks**) involves far fewer labels—mostly binary or three-way classification problems (e.g., SST-2, MRPC, QQP, RTE). Even across all tasks combined, the total number of categories remains below **20**, which is *much smaller* than the output feature dimension of **Flan-T5-base ($d_{\mathbf{out}}$ is typically equal to 2048)**. In

this high-dimensional, low-class regime ($C \ll d_{\text{out}}$), class vectors are already well separated without explicit geometric regularization. Thus, ETF alignment provides **limited additional improvement**, as the geometry is already near-orthogonal.

### G.2.6 WHY ETF RATHER THAN ORTHOGONAL OR LOW-RANK STRUCTURES?

**Theoretical motivation.** When the number of classes $C$ exceeds the feature dimension $d_{\text{out}}$, assigning perfectly orthogonal class directions becomes mathematically impossible. Enforcing strict orthogonality in such settings not only distorts meaningful inter-class relations but also wastes representational capacity. In contrast, the Equiangular Tight Frame (ETF) structure provides an *optimal and milder compromise* between angular separation and dimensional constraints:

- ETF is the unique configuration on the unit sphere that maintains *equal pairwise angles* and *balanced norms*;
- When $C > d_{\text{out}}$, orthogonality fails, yet ETF preserves constant pairwise inner products of $-1/(C-1)$;
- This property directly aligns with Neural Collapse theory, in which class means converge to a centered simplex ETF at the end of training.

Hence, ETF is not strictly orthogonal but represents a relaxation achieving maximal angular separation under dimensional constraints.

**Empirical comparison.** To further validate this design choice, we compared ETF-based alignment against **Low-Rank Alignment** and **Orthogonal Alignment** baselines, following the reviewer's suggestion. The rank in low-rank alignment was fixed to $d_{\text{out}}/4$. Although low-rank alignment preserved variance, it failed to maintain balanced angular separation across tasks, resulting in weaker class discrimination. Orthogonal alignment enforced stronger independence but sacrificed geometric uniformity when $C > d_{\text{out}}$. As shown in Table 15, both baselines perform consistently worse than our ETF-based method, confirming that enforcing the full ETF geometry yields superior directional consistency and inter-task separability.

Table 15: Comparison between ETF-based, low-rank, and orthogonal alignment strategies across different ViT architectures and task counts.

| Method | ViT-B/32 | | | ViT-B/16 | | | ViT-L/14 | | |
|---|---|---|---|---|---|---|---|---|---|
| | 8 tasks | 14 tasks | 20 tasks | 8 tasks | 14 tasks | 20 tasks | 8 tasks | 14 tasks | 20 tasks |
| **MDA TA (ETF)** | **86.4** | **81.4** | **77.3** | **89.9** | **85.8** | **84.7** | **92.0** | **89.4** | **88.4** |
| MDA TA (Low-Rank) | 79.7 | 74.0 | 68.1 | 85.6 | 79.8 | 75.6 | 90.6 | 87.1 | 85.8 |
| MDA TA (Orthogonal) | 84.7 | 79.7 | 75.0 | 88.6 | 84.2 | 80.5 | 91.2 | 88.2 | 87.0 |

**Conclusion.** Both theoretical reasoning and empirical evidence demonstrate that the ETF geometry provides the most balanced and expressive alignment target among feasible configurations. Unlike purely orthogonal or low-rank constraints, ETF regularization maintains uniform angular relationships across classes, preserving both discriminative power and representational efficiency in merged models.

## H USE OF LARGE LANGUAGE MODELS (LLMS)

In preparing this paper, we used large language models (LLMs) solely as an assistive tool for language polishing and minor writing improvements. The models were not involved in research ideation, experimental design, data analysis, or drawing scientific conclusions. All conceptual and technical contributions are the work of the authors. The authors take full responsibility for the contents of this paper.

## I DEVICES

In the experiments, we conduct all methods on a local Linux server equipped with one AMD EPYC 7742 64-Core Processor (128 logical threads). All methods are implemented using the PyTorch framework, and all models are trained on NVIDIA A800-SXM4-80G GPUs (80GB HBM2e memory).

