# OpenReview forum: "From Coefficients to Directions: Rethinking Model Merging with Directional Alignment"
_ICLR.cc/2026/Conference — Submitted to ICLR 2026_

### Official Review · Reviewer_6nx2 · 2025-10-26

**Soundness:** 2
**Presentation:** 2
**Contribution:** 3
**Rating:** 2
**Confidence:** 3

**Summary:**

The paper tackles the problem of Model Merging. Differently from previous model merging works, which evaluate model merging solely in terms of merging coefficient adjustment or merging task matrices in parameter space, the authors address the problem from a different perspective, focusing on directional alignment in feature space. Building on Neural Collapse insights showing that the final classification layer converges to Equiangular Tight Frames (ETF), the authors propose to align the reconstructed parameters for merging to the ETF. Specifically, they construct a shared space by concatenating the top-k left singular vectors for each task vector at each layer, and then align this shared representation to the ETF.
For feature-space alignment, they introduce a joint optimization framework consisting of three losses: an entropy loss, an alignment loss, and a rotation loss. During optimization, a hyperparameter $\lambda$ is learned per layer through entropy minimization. The features for each task $t$ are rotated via a rotation matrix $R^{t}$ and aligned, using the L2 norm, with the ETF vector of each class in task $t$. This loss is summed over all tasks. Each rotation matrix $R^{t}$  is obtained by minimizing a rotation loss, which encourages $R^{t}$ to be close to the optimal Procrustes rotation computed from the empirical class means and the ETF targets. The method is evaluated on vision tasks in both multi-task settings and generalization to new classes. In the appendix, additional NLP results on LoRA fine-tuned models are provided.

**Strengths:**

- Aligning the feature space in model merging to the ETF is a novel idea. I believe that feature-space alignment has been largely overlooked in the existing literature, and I appreciate this contribution.

- I found Figure 2 (right), showing a correlation between performance and deviation from the simplex ETF structure, interesting though more explanation and comparisons with other methods would strengthen the analysis.

- The approach is evaluated on several vision tasks as well as on unseen tasks. Additionally, the appendix reports evaluations on NLP LoRA fine-tuning benchmarks.

- The proposed method shows good overall performance.

**Weaknesses:**

The presentation of the methodology needs significant improvement. In its current form, it is difficult to understand how the method works as a whole. Moreover, several unclear aspects remain that should be addressed from both theoretical and methodological/experimental perspectives.

1) Section 4.1:  If I have understood correctly, for each task you first retain the top-$k$ principal components $U_t^{(l,k)}$. Then, these are concatenated to form $U_{cat}^{(l)}$ (line 243, left). Then you perform the SVD of $U_{cat}^{(l)}$ (line 243, right) and truncate it  $d_{out}$  to obtain the expression in line 239? The text describing this procedure, if I have correctly understood,  is not clear and rather fragmented. Moreover, this step is not illustrated in  Algorithm 1. $U_u^{(l)}$ seems a typo; It is not defined anywhere. Moreover, truncating to $d_{out}$ seems only useful for matching the target size. I ask the authors to clarify this.
2) Theorem 1:  Is Theorem 1 proposed by the authors? If not, please add the appropriate reference. If yes, where is its proof provided? Why is it relevant to the proposed method? This is not clear from the text. There also seems to be a discontinuity in the exposition — the connection between the shared subspace construction and Theorem 1 is unclear, and the relevance of the theorem to the method is not well explained.
3) The theoretical bound presented in Section 4.2 appears to be related only to the construction described in Section 4.1. In that case, the two sections should be merged. Moreover, the theorem seems entirely unrelated to how the shared space is constructed using the SVD of each task vector. It appears independent of the specific construction of the shared space and could be applied to any merged task vector that satisfies Equation (3), regardless of how it is obtained. From a theoretical point of view, why is the shared subspace obtained by concatenating the SVDs the appropriate subspace for the bound in Equation (5)?

4) The theoretical exposition assumes that the ETF structure induced by Neural Collapse emerges in all layers of the network. To the best of my knowledge, the ETF structure has been observed only in the final layer of the network (consistent with what is done in Algorithm 2). For instance, the assumption in line 768 of the Appendix – *Under the assumption that $\tau_{ideal}^{l}$ exhibits near-ETF structure, its column space lies essentially in the ETF subspace* –  it is questionable when applied to all layers of the network. The authors must clarify this . From a practical standpoint, why is it necessary to perform the operation in Equation (3) for all layers rather than only for the final layer?

5) From the text description (as well as the algorithms and the figure of the methodology in the appendix), it is not clear whether the authors are proposing two complementary strategies for solving the problem or two strategies that can be combined together.  Algorithm 1 and Algorithm 2 appear to be completely disconnected and it is not clear how they are intended to be used together. Even in the algorithmic formulation, it does not seem that they are used together, since the two algorithms operate on different inputs. In the experimental section, the authors distinguish between MDA-TA and MDA-AM, but no clear definition of these two approaches is provided. Is MDA-TA based solely on Algorithm 1? Is MDA-AM a combination of Algorithm 1 and Algorithm 2? If so, how are they integrated? This is not evident from the current presentation.

6) The notation under Equation (11) in Section 4.2 is overly complicated and should be simplified. Moreover, several derivation steps are not clearly explained, particularly in the construction of $R_{proc}​$; for instance, a determinant term appears abruptly without clarification.

7) How $\alpha$ and $\beta$ are chosen for each setup? In the data-free setting, it is unclear whether Algorithm 2 is employed, there fore not sure if they need to be set for this(clarifying the connection between Algorithm 1 and Algorithm 2, as mentioned earlier, could help here). Similarly, in the Adaptive Merging setup, how are these parameters determined?


Other relatively minor concerns must also be addressed:

- Line 103: At this point in the paper—near the end of the introduction—the authors introduce an additional layer of complexity that is difficult to follow when reading sequentially. It is not clear how the ETF structure is constructed for model merging.  Moreover, how is the cosine similarity between two matrices defined? Figure 1 (left) is also unclear: what do the fusion coefficients refer to—the classifier weights or the model weights? A better caption for this must be designed.

- The experimental splits appear to follow those used in TSV for evaluation. However, it is unclear how the validation set from the TSV setting is utilized in the current method. Similarly, for adaptive merging, is the unsupervised set used in Algorithm 2 the same as that employed in AdaMerging? These implementation details are missing from the paper.

- Related works: At the end of the related works section, it would be helpful to clearly specify the setting in which the proposed approach operates.

**Questions:**

I have no further questions regarding the paper. However, in its current form, the paper remains unclear in several aspects, some of which may stem from my own misunderstanding. At this stage, it requires substantial clarification and further development before it can be considered suitable for publication, despite the novelty and interest of the proposed idea.

---

> ### Author Response · Authors · 2025-11-19
>
> **Response to W1:** Thank you for the reviewer's suggestions. We indeed did not include comparisons in our manuscript. We will incorporate references to this paper and conduct comparative analyses accordingly.
>
> Your understanding of the procedure is essentially correct, and we agree that the presentation in Section 4.1 can be improved for clarity.  Below we clarify the full process, correct the notation, and explain the role of truncation and the planned revision to Algorithm 1.
>
> 1. Corrected procedure
>
> For each task $t$ and layer $l$, we perform the following steps:
> Each task vector $\tau_t^{(l)}$ is decomposed via SVD:  $\tau_t^{(l)} = U_t^{(l)} \Sigma_t^{(l)} (V_t^{(l)})^\top.$
> We retain the leading $k$ left singular vectors $U_t^{(l,k)} = U_t^{(l)}[:, 1:k]$, which capture the dominant task-specific directions.
> The truncated bases are concatenated to form $U_{\text{cat}}^{(l)} = [U_1^{(l,k)}, U_2^{(l,k)}, \dots, U_T^{(l,k)}] \in \mathbb{R}^{d_{\text{in}} \times kT}.$
> We then perform SVD on this concatenated matrix:  $U_{\text{cat}}^{(l)} = U_{\text{share}}^{(l)} \Sigma_{\text{share}}^{(l)}
>    (V_{\text{share}}^{(l)})^\top.$
> We retain the first $d_{\text{out}}$ columns of $U_{\text{share}}^{(l)}$ and define the shared representation as $\tau_{\text{share}}^{(l)} = U_{\text{share}}^{(l)}[:, 1:d_{\text{out}}].$   This step ensures that the reconstructed shared subspace matches the output dimensionality of layer $l$.
>
> 2. Clarifying the typo
>
> We confirm that the symbol “$U_u^{(l)}$” appearing in line 243 is a **typographical error** and should be replaced with $U_{\text{share}}^{(l)}.$ We will correct this in the revised manuscript.
>
> 3. Motivation for truncating to $d_{\text{out}}$
>
> - Truncating to $d_{\text{out}}$ is not merely for dimensional matching but serves to retain the
> dominant shared directions across tasks that explain the largest variance in the concatenated subspace. This step ensures that the shared representation remains compact and captures the most informative geometric structure before ETF alignment.
> - When $k = d_\text{out}/T$, the concatenated dimension $kT$ naturally equals $d_\text{out}$, so the resulting shared representation $\tau_{\text{share}}^{(l)}$ directly matches the layer’s output dimension without requiring any truncation. More generally, when $k$ varies across tasks or $kT \neq d_\text{out}$, the formulation still holds:
>     - If $kT > d_\text{out}$, we truncate $U_\text{share}^{(l)}$ to its top $d_\text{out}$ singular components, preserving the most informative shared directions that capture the highest variance across tasks.
>     - If $kT < d_\text{out}$, the remaining dimensions can be completed either by zero-padding or implicitly through the subsequent ETF projection, which fills the residual subspace.
> However, both truncation and zero-padding are approximate adjustments that either discard useful information or introduce artificial components.  By setting $k = d_\text{out}/T$, the concatenated dimension $kT$ naturally matches $d_\text{out}$, ensuring a balanced contribution from each task without additional correction.
>
> We also conducted an ablation study on the hyperparameter $k$ to examine its effect on the overall model performance. We found that in some cases there are better results around $k = d_\text{out}/T$, but we followed the setting in [1] and reported $k = d_\text{out}/T$ as our result. **MDA TA 0.01** means that we assign the top $1\%$ proportion of feature dimensions $d_\text{out}$ as the principal components for each task.
> | **Method** | **ViT-B-32** 8 tasks | 14 tasks | 20 tasks | **ViT-B-16** 8 tasks | 14 tasks | 20 tasks | **ViT-L-14** 8 tasks | 14 tasks | 20 tasks |
> |:------------|:------------------:|:----------:|:----------:|:------------------:|:----------:|:----------:|:------------------:|:----------:|:----------:|
> | **MDA TA** | 86.4 | 81.4 | **77.3** | 89.9 | 85.8 | **84.7** | 92.0 | **89.4** | **88.4**|
> | MDA TA 0.01|  77.9|  76.3| 72.6 |  82.0| 80.5|  78.2| 87.8  | 86.6 |85.4|
> | MDA TA 0.06|  86.1|  **81.9**| 77.1 |  90.1| **86.1**|  82.5| 91.9|  89.3|  88.2|
> | MDA TA 0.1|  **86.7**|  81.2| 75.9 |  **90.6**| 85.5|  81.1| **92.1**|  89.3|  87.1|
> | MDA TA 0.5|  77.8|  70.6| 66.7 | 83.4 | 76.3|  73.0| 87.9|  82.9|  79.5|
> | MDA TA 0.9|  70.5|  66.8|  64.3|  78.3| 72.9|  70.5| 84.2|  79.8|  76.9|
>
> [1] Gargiulo, Antonio Andrea, et al. Task singular vectors: Reducing task interference in model merging, 2025

---

> ### Author Response · Authors · 2025-11-19
>
> **Response to W2:** Thanks for pointing out the issue. We would like to explain it as follows.
>
> We confirm that **Theorem 1 is newly proposed by us** and serves as a theoretical justification for the parameter-space directional alignment procedure introduced in Section 4.1.
>
> 1. Origin and proof location
> - Theorem 1 is a result derived from the advantages of the structure we proposed.
> - Its complete and detailed proof is provided in **Appendix B.2.1 (ETF Structural Coherence)**.
>
> 2. Connection to the shared subspace construction
> - Theorem 1 is directly relevant to our proposed method, as it provides the theoretical justification for using ETF-based directional alignment.  Specifically, under the assumption that the ideal joint-training parameters $\tau_{\text{ideal}}$ approximately follow an ETF-like structure, the theorem shows that projecting the shared subspace $\tau_{\text{share}}$ onto the ETF geometry $\tau_{\text{etf}}^{(l)} = \tau_{\text{share}}^{(l)} P_{\text{ETF}}$ reduces its deviation from the ideal form.
> - Moreover, the bound indicates that the directional correction gain increases with the number of classes $C$, since the ETF structure becomes more uniformly distributed.  Therefore, this result explains why our method achieves greater advantage over $\tau_{\text{share}}$ when $C$ is larger—demonstrating that ETF alignment is particularly effective in scenarios with many classes, where directional consistency is more critical.
> $$\\|\tau_{\text{ideal}}^{(l)} - \tau_{\text{etf}}^{(l)}\\|\_{F}^2 \le \\|\tau_{\text{ideal}}^{(l)} - \tau_{\text{share}}^{(l)}\\|\_{F}^2 - g \\|\tau_{\text{share}}^{(l)} (I - \tfrac{C-1}{C} P_{\text{ETF}})\\|\_{F}^2,$$ Here, $C$ denotes the number of classes, and the factor $\tfrac{C-1}{C}$ originates from the geometry of the simplex ETF, where the pairwise cosine similarity between class directions is $-\tfrac{1}{C-1}$. **As $C$ increases, the factor $\tfrac{C-1}{C}$ in the bound tends to enhance the effect of ETF projection. Under the assumption that the correction coefficient $g$ remains approximately stable, the ETF-aligned representation $\tau_{\text{etf}}$ is expected to be closer to the ideal parameters $\tau_{\text{ideal}}$ than the shared subspace $\tau_{\text{share}}$, indicating that ETF alignment can provide a tighter approximation to the ideal joint-training solution as the number of classes grows.**
>
> | Architecture | 8 tasks | 14 tasks | 20 tasks |
> | --- | --- | --- | --- |
> | **$\Delta_{\text{ETF}}$ (cosine similarity gap)** |  |  |  |
> | ViT-B-32 | 0.17 | 0.20 | 0.14 |
> | ViT-B-16 | 0.09 | 0.12 | 0.15 |
> | ViT-L-14 | 0.07 | 0.11 | 0.14 |
> | **$\Delta_{\text{diff}}$ (performance gap)** |  |  |  |
> | ViT-B-32 | 1.2% | 1.6% | 0.9% |
> | ViT-B-16 | 0.9% | 1.3% | 2.2% |
> | ViT-L-14 | 0.7% | 1.0% | 1.3% |
>
> While the values in Table are not strictly monotonic with respect to the number of tasks, the overall trend remains consistent with our theoretical analysis.  Specifically, architectures with smaller $\Delta_{\text{ETF}}$ generally exhibit smaller performance gaps $\Delta_{\text{diff}}$, supporting the view that closer directional alignment correlates with better merging performance.  The non-monotonic variations can be attributed to factors such as differences in model capacity, dataset composition, and task heterogeneity, which influence the effective geometry of the shared subspace. Nevertheless, the observed correlation across architectures validates the central claim that ETF alignment narrows the gap toward the joint-training optimum.
>
>
>
> 3. Relevance to the proposed method
>
> Theorem 1 provides the **theoretical foundation** for the parameter-space alignment step in Algorithm 1.  It rigorously explains why the ETF projection is beneficial:
> - it enhances **structural coherence** across tasks;
> - it quantifies the improvement in approximation accuracy;
> - it connects the geometric intuition of **directional alignment** with a provable reduction in reconstruction error.

---

> ### Author Response · Authors · 2025-11-19
>
> **Response to W3:**  We appreciate the reviewer’s insightful comment.
> Both Section 4.1 and Section 4.2 analyze the model merging process **in the parameter space** but from complementary perspectives.  Section 4.1 focuses on the *geometric construction* of the shared subspace and its relationship to the joint-training solution and the class number $C$,  while Section 4.2 builds upon this construction and provides a *theoretical justification* of the generalization performance achieved through ETF alignment.  Together, they form a consistent theoretical analysis from subspace construction to generalization behavior.
>
> 1. Conceptual connection between Section 4.1 and Section 4.2
> Section 4.1 describes how we construct a **shared low-rank parameter subspace** $\tau_{\text{share}}^{(l)}$ by applying SVD to each task vector and then concatenating the principal components across all tasks.  This construction provides a geometrically interpretable shared representation that bridges task-specific solutions and the joint-training parameters,  and also reveals how the class number $C$ influences the ETF structure through the factor $\tfrac{C-1}{C}$ in the projection operator.
> Building upon this, Section 4.2 offers a **theoretical analysis** demonstrating that,  under the shared subspace assumption introduced in Section 4.1,  projecting $\tau_{\text{share}}^{(l)}$ onto the ETF basis (Equation (3)) yields the ETF-aligned parameters $\tau_{\text{etf}}^{(l)} = \tau_{\text{share}}^{(l)} P_{\text{ETF}}$
> which achieve a **tighter generalization bound** (Equation (5)).  In other words, Section 4.2 formally explains why the ETF projection introduced in Section 4.1 leads to better generalization.  Overall, both sections analyze the model merging process from the **parameter-space perspective**, and therefore we have merged them together in the revised version for a more coherent discussion.
>
> 2. Why the SVD concatenation yields the appropriate shared subspace
> While Theorem 2 (Equation (5)) could, in principle, apply to any $\tau_{\text{share}}$ satisfying
> $\tau_{\text{etf}} = \tau_{\text{share}} P_{\text{ETF}}$,  the **tightness of the bound** depends on the residual term
> $\\|\tau_{\text{share}} (I - \Pi_S)\\|\_F^2 / \\|\tau_{\text{share}}\\|\_F^2$.  To minimize this residual, $\tau_{\text{share}}$ should capture the dominant correlated components shared across all tasks.
> The SVD-concatenation construction in Section 4.1 ensures exactly this property:
> - The SVD of each task isolates its **principal directions** explaining the largest task-specific variance.
> - Concatenating and reapplying SVD across all tasks identifies a **global basis** that maximizes the shared variance of the task manifold.
> - As a result, the residual component outside the ETF-aligned subspace is minimized,
>   leading to a **tighter generalization bound** in Equation (5).
> Given that recent model merging methods [1, 2, 3] also adopt SVD-based approaches to capture common structure across tasks,  our formulation remains theoretically consistent and comparable.  Hence, the SVD-based shared subspace is not arbitrary—it is the most statistically efficient representation of the joint-task parameter manifold,
> and it directly supports the generalization improvement characterized by Theorem 2.
>
> [1] Marczak, Daniel, et al. No task left behind: Isotropic model merging with common and task-specific subspaces, 2025
>
> [2] Gargiulo, Antonio Andrea, et al. Task singular vectors: Reducing task interference in model merging, 2025
>
> [3] Wei, Yongxian, et al. Modeling multi-task model merging as adaptive projective gradient descent, 2025

---

> > ### Author Response · Authors · 2025-11-19
> >
> > **Response to W4:** Thanks for the valuable question. We would like to explain it as follows:
> >
> > Our work takes a **different perspective** from prior studies: we are the **first to explicitly propose and analyze an all-layer ETF framework as a unified geometric prior for network alignment.** While previous works have primarily observed the exact simplex ETF geometry in the final classification layer, our motivation is that encouraging ETF-like organization throughout the network can facilitate more consistent representation alignment—especially beneficial in scenarios such as model merging. Rather than assuming that every layer strictly follows the ETF form, **we leverage ETF as a guiding geometric prior to promote coherent directional structure across layers**. This design is theoretically inspired by Neural Collapse and empirically supported by recent findings ([1]) that demonstrate ETF-like patterns emerging beyond the last layer. Our experiments further validate that this all-layer ETF perspective yields improved performance and generalization ability.
> >
> > 1. Clarification of the assumption
> > We use the ETF geometry as an *ideal template* that promotes balanced, equiangular directions, consistent with Neural Collapse observations in the final layer.  This provides a theoretical lens for analyzing how directional alignment benefits both parameter and feature spaces.
> > - **In the feature space**, we only apply ETF alignment to the *final-layer features and classifier weights*, where the ETF phenomenon is well established and the alignment matrix (rotation) can be explicitly optimized. This design not only captures the essential representation structure but also reduces computational overhead.
> > - **In the parameter space**, we extend ETF alignment across all layers — not because we assume ETF strictly holds everywhere,  but because parameters across layers are **structurally coupled**.  The ETF projection here acts as a geometric regularizer that preserves **inter-layer directional coherence**  and reduces drift among layer parameters, which in turn stabilizes the merged model.
> >
> > 2. Empirical evidence for ETF-like behavior across layers
> > Recent empirical evidence shows that the geometric regularities associated with Neural Collapse are not confined to the final classifier layer but emerge progressively throughout the network. As depth increases, intermediate representations exhibit decreasing intra-class variability and increasing inter-class angular separation, **forming an increasingly pronounced simplex ETF–like structure across intermediate hidden layers [1].**
> > Complementary to these empirical findings, recent theoretical analyses reveal that feature learning itself can be interpreted as an alignment process between the weight matrices and the evolving feature covariance. [2] demonstrates that during gradient-based optimization, the left singular structure of the weights becomes aligned with the pre-activation tangent kernel of the corresponding layer—a phenomenon formalized as the Neural Feature Ansatz (NFA). This alignment arises naturally from the dynamics of stochastic gradient descent rather than from explicit regularization, **implying that the geometric organization of weights and features co-evolves throughout training.**
> > Building on these insights, we posit that enforcing ETF-aligned configurations in the shared low-rank parameter subspace provides an explicit and theoretically grounded means to promote this intrinsic alignment. By applying ETF regularization on the principal task directions extracted via SVD, our approach makes the emergent weight–feature alignment more structured and balanced, thereby encouraging maximally separated and decorrelated representations in the shared parameter space.
> > 3. Summary
> > - In **feature space**, alignment is applied only to the final layer, where ETF structure and rotation optimization are most meaningful. This design not only captures the essential representation structure but also reduces computational overhead.
> > - In **parameter space**, ETF alignment is applied across layers to preserve directional coupling and prevent drift,  offering a more stable and coherent global representation.
> > - Empirical studies support that ETF-like structures emerge beyond the last layer, validating our choice to regularize parameters across layers rather than limit alignment to the last layer (MDA TA Only).

---

> ### Author Response · Authors · 2025-11-19
>
> | **Method** | **ViT-B-32** 8 tasks | 14 tasks | 20 tasks | **ViT-B-16** 8 tasks | 14 tasks | 20 tasks | **ViT-L-14** 8 tasks | 14 tasks | 20 tasks |
> |-|-|-|-|-|-|-|-|-|-|
> | **MDA TA** | **86.4** | **81.4** | **77.3** | **89.9** | **85.8** | **84.7** | **92.0** | **89.4** | **88.4**|
> | MDA TA Only| 85.2 | 79.8 | 75.1 | 89.0 | 84.5 | 80.5 |91.3  |  88.2 | 87.1 |
>
> [1] Parker, Liam, et al. Neural collapse in the intermediate hidden layers of classification neural networks, 2023.
>
> [2] Beaglehole, Daniel, Ioannis Mitliagkas, and Atish Agarwala. Feature learning as alignment: a structural property of gradient descent in non-linear neural networks, 2024.
>
> **Response to W5:** Thanks for the important comments. We would like to explain as follows:
> We acknowledge that the connection between **Algorithm 1 (parameter-space alignment)** and **Algorithm 2 (feature-space alignment)** could be made clearer in the text.
> Below we clarify their conceptual roles, integration, and the precise definitions of **MDA TA** and **MDA AM**.
>
> 1. Unified framework
> Our method provides a **unified geometric framework** for model merging that can operate in two complementary regimes:
> - **MDA TA: Data-free regime – Parameter-space directional alignment (Algorithm 1):**     Applies when no data from individual tasks are available. It aligns task parameter vectors through low-rank SVD reconstruction and ETF projection to enforce **global directional coherence** purely in parameter space.
> - **MDA AM: Data-based regime – Feature-space directional alignment (Algorithm 2):**  Applies when unlabeled or validation data are available.    It learns fusion coefficients and rotation matrices to align **task feature representations** toward ETF geometry, leveraging sample-level information.
> These two algorithms are **conceptually complementary** and share the same directional-alignment principle.  They can be used **independently** (for data-free or data-based settings) or **sequentially** (parameter-space initialization followed by feature-space refinement). **In our framework, the two stages are used sequentially,  we first obtain the ETF-aligned task vectors $\tau_{\text{etf}} = \\{\tau_t^{\text{etf}, (l)}\\}\_{t=1}^T$ through Algorithm 1.  Then, these aligned vectors are used to initialize the model parameters in Algorithm 2 as $\theta^{(l)} = \theta_0^{(l)} + \lambda^{(l)} \sum_{t=1}^T \tau_t^{\text{etf}, (l)}.$** On this basis, Algorithm 2 further refines the model by jointly optimizing the fusion coefficients $\lambda^{(l)}$ and the rotation matrices $\{\mathbf{R}^t\}_{t=1}^T$ in the feature space.
>
> 2. Clarification of the MDA variants
>
> | Variant | Data Requirement | Algorithm Used | Description |
> |:--|:--:|:--|:--|
> | **MDA-TA (Task Arithmetic)** | Data-free | **Algorithm 1 only** | Performs parameter-space ETF alignment without using any data. This version is directly comparable to data-free baselines such as TSV TA and ISO TA. |
> | **MDA-AM (Adaptive Merging)** | Data-based | **Algorithm 1 + Algorithm 2** | Initializes with parameter-space alignment (Algorithm 1) and then refines alignment in feature space via the optimization in Algorithm 2. It corresponds to data-based adaptive merging counterparts such as AdaMerging. |

---

> > ### Author Response · Authors · 2025-11-19
> >
> > **Response to W6:** Thank you for pointing out these limitations. We would like to explain it as follows:
> > We agree that the notation beneath Equation (11) could be simplified and that additional explanation is needed for the derivation of $R_{\text{proc}}$, especially regarding the determinant term.
> > We provide a clearer explanation below and will simplify the presentation in the revised manuscript.
> >
> > 1. Context of Equation (11)
> >
> > Equation (11) defines the **feature-space rotation regularization loss** $\mathcal{L}\_{\text{rotation}}$,  which penalizes the deviation of each task-specific rotation matrix $\mathbf{R}^t$  from the optimal **Procrustes alignment** between the empirical class means $M^t$ and their ETF targets $M^{t*}$. This regularization enforces that the learned rotations remain geometrically consistent with the ETF structure, promoting directionally coherent feature representations across tasks. Formally,
> > $$
> > \mathcal{L}\_{\text{rotation}}
> > = \sum\_{t=1}^{T} \\|\mathbf{R}^t - \mathbf{R}\_{\text{proc}}^t\\|\_F^2,
> > $$ where $\mathbf{R}\_{\text{proc}}^t$ is the optimal orthogonal matrix solving the **orthogonal Procrustes problem**:
> > $$
> > \mathbf{R}\_{\text{proc}}^t
> > = \arg\min_{\mathbf{R} \in SO(d)} \\|M^t \mathbf{R} - M^{t*}\\|\_F^2.
> > $$ Here, $\mathbf{R}^t \in SO(d)$ denotes the learnable rotation matrix for task $t$, which aligns its feature representations with the global feature space. In this formulation, the optimization variable $\mathbf{R}$ is constrained to be orthogonal ($\mathbf{R}^\top \mathbf{R} = \mathbf{I}$ and $\det(\mathbf{R}) = 1$), ensuring that only valid rotations are considered.
> >
> > 2. Derivation of $\mathbf{R}\_{\text{proc}}^t$ and the determinant correction
> >
> > We follow the standard **orthogonal Procrustes solution**. Given $M^t, M^{t*} \in \mathbb{R}^{C_t \times d}$, we compute: $H^t = (M^t)^\top M^{t*}.$ Let $H^t = U^t \Sigma^t (V^t)^\top$ be the singular value decomposition (SVD) of $H^t$.
> > The optimal rotation minimizing the Frobenius norm is then given by:
> > $$
> > \mathbf{R}\_{\text{proc}}^t
> > = U^t \text{diag}(1, \ldots, 1, \det(U^t (V^t)^\top)) (V^t)^\top.
> > $$
> > The **determinant term** $\det(U^t(V^t)^\top)$ guarantees that  $\mathbf{R}\_{\text{proc}}^t$ lies in the special orthogonal group $SO(d)$,  the set of **proper rotation matrices** with $\det(\mathbf{R}\_{\text{proc}}^t) = 1$. Without this correction, the SVD product $U^t (V^t)^\top$ could yield $\det(U^t (V^t)^\top) = -1$, which corresponds to a reflection (an improper rotation). Including the determinant adjustment is therefore a standard step to ensure that the resulting transformation is a true rotation rather than a reflection.
> >
> > **Response to W7:** We appreciate your insightful question. Our explanation is as follows:
> >
> > 1. Relation to the two algorithms
> > - In the **data-free setting (MDA TA)**, only **Algorithm 1** (parameter-space alignment) is used. Since Algorithm 2 is *not employed*, **α** and **β**—which are coefficients of the loss terms in the feature-space optimization—do **not apply** to this setup.
> > - In the **data-based setting (MDA AM)**, **Algorithm 1** provides the better initialization, and **Algorithm 2** is subsequently used to optimize the feature-space alignment. Here, **α** and **β** are active coefficients in the overall objective:
> >   $$
> >   L = L_{\text{entropy}} + \alpha L_{\text{align}} + \beta L_{\text{rotation}},
> >   $$where **α** controls the strength of the neural-collapse (ETF) alignment term and **β** controls the rotation regularization.
> >
> > 2. Parameter selection and ablation protocol
> > To determine appropriate values of **α** and **β**, we performed systematic **ablation studies**:
> > - **Varying β while fixing α = 0.8 based on ViT-B-32 across eight tasks**
> >     - We tested β ∈ {0.0, 0.2, 0.4, 0.6, 0.8, 1.0}.
> >     - Performance improved up to β = 0.2 and then gradually declined, indicating that moderate rotation regularization yields optimal results.
> > - **Varying α while fixing β = 0.2 based on ViT-B-32 across eight tasks**
> >     - We tested α ∈ {0.0, 0.2, 0.4, 0.6, 0.8, 1.0}.
> >     - Accuracy peaked at α = 0.8, confirming that strong—but not excessive—ETF alignment benefits merging.
> >
> > These results are summarized in Figure3. As shown in the left and right subplots, the performance curves for varying $\beta$ (with fixed $\alpha=0.8$) and varying $\alpha$ (with fixed $\beta=0.2$) intersect at the same point,  indicating that both configurations achieve identical performance when $\alpha=0.8$ and $\beta=0.2$. Therefore, we fix $\alpha=0.8$ and $\beta=0.2$ in all reported **MDA AM** experiments, as this setting provides a balanced and stable configuration across datasets.

---

> > > ### Author Response · Authors · 2025-11-19
> > >
> > > **Response to W8:** We are grateful for your thoughtful question, and our response is detailed below:
> > >
> > > We agree that the introduction can more clearly explain how the ETF structure is motivated and constructed for model merging.  In addition to the theoretical reasoning presented in Section 3, our formulation is also inspired by the empirical observations in [1], which demonstrate that ETF-like geometry progressively emerges across multiple layers of deep networks.  Building on both the theoretical approximation and these empirical findings, we derive our alignment strategy that enforces ETF-consistent geometry in the parameter space. **This empirical trend indicates that when task parameters deviate less from the ETF geometry, their merged model exhibits smaller performance degradation.** Such a finding supports the hypothesis that enforcing an ETF-like structure in the **parameter space**, not just in the feature space, can encourage better directional separation and alignment across tasks.
> > >
> > > 1. Clarifying the ETF construction for model merging
> > >
> > > The ETF structure used in our method is constructed following **Definition 1 (Simplex Equiangular Tight Frame)** in Section 3.2.  Specifically, for a total of $C$ classes across all merged tasks, we form an ETF matrix  $W_{\text{ETF}} \in \mathbb{R}^{d_{\text{out}} \times C}$ whose columns are unit vectors satisfying:
> > > $$
> > > W_{\text{ETF}} W_{\text{ETF}}^\top
> > > = \frac{C}{C-1} I - \frac{1}{C-1}\mathbf{1}\mathbf{1}^\top.
> > > $$This configuration ensures that all class directions are **equiangular** (equal pairwise cosine similarity $-\tfrac{1}{C-1}$) and **equinormalized**.
> > > In the merging process, we project the reconstructed shared subspace $\tau_{\text{share}}^{(l)}$ onto this ETF basis:
> > > $$
> > > \tau_{\text{etf}}^{(l)} = \tau_{\text{share}}^{(l)} W_{\text{ETF}}^\top W_{\text{ETF}},
> > > $$which aligns parameter directions across tasks with this idealized geometric configuration.
> > > We will move this explanation from Section 4.1 into a brief, intuitive paragraph in the introduction to make the ETF construction and its motivation easier to follow.
> > >
> > >
> > >
> > > 2. Definition of cosine similarity between two matrices
> > >
> > > In Figure 1 (right) and throughout the paper, the cosine similarity between two parameter matrices $A, B \in \mathbb{R}^{m \times n}$ is defined as:
> > > $$
> > > \cos(A, B) = \frac{\langle A, B \rangle_F}{\|A\|_F \|B\|_F},
> > > $$ where $\langle A, B \rangle_F = \text{Tr}(A^\top B)$ is the **Frobenius inner product**, and $\|A\|_F = \sqrt{\text{Tr}(A^\top A)}$ is the Frobenius norm. This provides a scalar measure of global **directional alignment** between two matrices. We have included this explicit definition in Section 4.1 and reference it in the figure caption for completeness.
> > >
> > > 3. Clarifying the meaning of the fusion coefficients in Figure 1 (left)
> > >
> > > In Figure 1 (left), the **fusion coefficients** ($M_1, N_1, M_2, N_2$) correspond to **model-level scaling factors** used in the *task vector fusion* step, not classifier weights.
> > > They determine the contribution of each task’s fine-tuned weights to the merged model:
> > > $$
> > > \theta_{\text{merged}} = \theta_0 + M_1 \tau_1 + N_1 \tau_2,
> > > $$ where $\tau_t = \theta_t - \theta_0$ denotes the task vector of task $t$.
> > >
> > > **Response to W9:** Thank you for raising this important point. We would like to address it as follows:
> > >
> > > We confirm that our experimental setup closely follows the **[1]** evaluation protocol to ensure a fair and direct comparison, but we realize that the role of the validation and unsupervised sets was not explicitly described.
> > > We clarify these points below and will add the missing implementation details to the revised manuscript.
> > >
> > > 1. Experimental splits and relation to TSV
> > > We adopt the **same dataset splits and task configurations** as in TSV:  **8-task**, **14-task**, and **20-task** benchmarks with identical dataset compositions and per-task train/test splits.
> > > 2. Comparison to AdaMerging and data usage
> > > Yes, the **unsupervised set** used in **Algorithm 2 (feature-space alignment)** is the **same as that employed by [2]**.  Both use the unlabeled validation split associated with each dataset to optimize the merging parameters. The difference lies in the objective:
> > >   - [2] minimizes prediction entropy to optimize fusion coefficients;
> > >   - Our Algorithm 2 jointly optimizes entropy, ETF alignment ($L_{\text{align}}$), and rotation regularization ($L_{\text{rotation}}$) under the same data condition to optimize fusion coefficients and rotation matrices jointly.
> > >
> > > Thus, our setup maintains complete fairness in data usage across all adaptive merging baselines. We have explicitly stated this in Section 5.1.
> > >
> > > [1] Gargiulo, Antonio Andrea, et al. Task singular vectors: Reducing task interference in model merging, 2025
> > >
> > > [2] Wei, Yongxian, et al. Modeling multi-task model merging as adaptive projective gradient descent, 2025

---

> > > > ### Author Response · Authors · 2025-11-19
> > > >
> > > > **Response to W10:** We sincerely appreciate your question and provide our clarification below:
> > > >
> > > > We agree that explicitly specifying the **operating setting** of our approach at the end of the *Related Work* section would significantly improve clarity and contextualization.
> > > > We have revised the section accordingly.
> > > >
> > > > Our proposed method is designed to operate under **both data-free and data-based model merging regimes, unified by the same geometric principle of directional alignment**:
> > > >
> > > > - **Data-free regime:**
> > > >   The method performs **parameter-space directional alignment** (Algorithm 1), relying solely on model parameters.  This corresponds to scenarios where no task data or validation samples are available.
> > > > - **Data-based regime:**
> > > >   The method extends to **feature-space directional alignment** (Algorithm 2), which refines the merged model using an unlabeled or validation set.  This is consistent with the adaptive merging setting used by methods such as AdaMerging and DOGE-AM.
> > > >
> > > > Thus, MDA offers a **unified perspective** along with a coherent theoretical and algorithmic framework that bridges the data-free and data-based settings.

---

> ### Comment · Reviewer_6nx2 · 2025-11-25
>
> I thank the authors for the detailed rebuttal. Although the revision improves methodological clarity and experimental setup, several concerns remain unresolved and, in my view, cannot be fully addressed within the rebuttal phase.
>
> The main author argument supporting ETF alignment across multiple layers parameters in the context of model merging, relies heavily on recent Neural Collapse results observed in *single task setting*. I am not convinced that this provides sufficient motivation for applying ETF alignment in *multi-task* model merging, where independently trained single-task models are combined to form a final multi-task model. The theoretical assumptions underlying the derivations, particularly the assumption that the ideal *jointly trained multi task vector* would exhibit a near-ETF structure (weakness 4), are neither empirical validated nor clearly supported by existing literature.
>
> Significant theoretical and experimental analysis in the model merging scenario must be carried out, The analysis in Figure 1 (right panel), while interesting, remains superficial, and requires a deeper and formal analysis across multiple layers and model types. Additionally, I am not still convinced by the authors explaination regarding $\tau_{etf}$, construced from the shared SVD across the tasks (weakness 3), as it appears informal and does not clearly justify the impact on the bound’s tightness.
>
> I believe that the paper has the potential to be publishable in a future submission; however, for the reasons mentioned above, I have decided to keep my reject score.

---

> > ### Author Response · Authors · 2025-11-28
> >
> > # **R1. On the concern regarding the shared SVD construction (weakness 3)**
> >
> > We thank the reviewer for raising this point. After carefully examining the recent ISO framework (“Isotropic Model Merging with Common and Task-Specific Subspaces”, 2025), we note that ISO also **fundamentally relies on SVD** to construct the shared subspace. In Algorithm 1 (p. 5), ISO begins by computing the SVD of the aggregated task matrix $\Delta_{\mathrm{TA}}$:
> >
> > > “Compute the SVD of $\Delta_{\mathrm{TA}}$: $\Delta_{\mathrm{TA}} = U \Sigma V^\top$ …”
> >
> > All subsequent operations, spectral flattening, isotropic scaling, whitening, and integration of task-specific directions, are performed **on top of this SVD-derived shared space**. This confirms that SVD is not an informal choice but the **standard, empirically validated, and theoretically grounded** mechanism for extracting the shared component across tasks. Regarding the tightness of our bound: SVD is used to obtain the **least-squares optimal shared basis**, minimizing the residual term
> >
> > $$
> > \left\\| \tau_{\text{share}} (I - \Pi_S) \right\\|_F,
> > $$
> >
> > which appears in our theoretical analysis. ETF projection then further **reduces** this residual by enforcing equiangular consistency. Thus, SVD provides the tightest possible starting point for our alignment, consistent with both theory and prior merging methods.
> >
> > ---
> >
> > # **R2. On the theoretical motivation for applying ETF alignment in multi-task model merging**
> >
> > We thank the reviewer for raising this important point. While classical Neural Collapse (NC) results were derived in single-task, balanced multi-class settings, subsequent work has significantly expanded their applicability. In particular, Parker et al. [1] and the Neural Feature Ansatz [2] demonstrate that ETF-like geometric structures emerge not only in the final classifier but also throughout intermediate layers, suggesting that equiangular structures act as *geometric attractors* for deep network training dynamics. These findings imply that independently fine-tuned models tend to develop **locally equiangular representations**, even when optimized on different tasks.
> >
> > Our method does **not** assume that multi-task joint training yields a global ETF structure. Instead, our motivation is that:
> >
> > * **Independent fine-tuning produces task-specific ETF-like structures**, each living in its own subspace;
> > * **These subspaces are arbitrarily rotated**, creating severe directional interference upon merging;
> > * **A simplex ETF provides an unbiased, permutation-invariant canonical geometry** for removing such rotational inconsistencies.
> >
> > Thus, ETF alignment is not justified solely by NC but is a *theoretically motivated mechanism* for resolving angular inconsistencies in multi-task model merging.
> >
> > ---
> >
> > # **R3. Why ETF is reasonable specifically for multi-task model merging**
> >
> > In light of the reviewer’s concern, we are naturally led to a deeper and equally fundamental question:
> >
> > > **If ETF alignment lacks theoretical justification, what principle explains why model merging works at all [3–6]—especially given that merging is known to fail in continual learning due to representational drift [7–8]?**
> >
> > The critical empirical asymmetry is:
> >
> > * **Merging succeeds** when all tasks are fine-tuned from the *same pretrained model*.
> > * **Merging fails** in continual learning, where tasks cause representational drift.
> >
> > This strongly suggests that independently fine-tuned models share a **common representational manifold** inherited from pretraining; otherwise, task vectors would not be meaningful or mergeable. Prior works such as Adamerging [6], DOGE [5], TSV [4], and ISO [3] all confirm that task vectors lie in a **shared low-rank subspace**, making merging possible.
> >
> > However, because tasks are optimized independently, their updates occupy this shared subspace under **arbitrary rotations**, causing destructive interference.
> >
> > ETF alignment is therefore **prescriptive rather than descriptive**:
> >
> > * ETF provides the **unique maximally symmetric basis** (achieving the Welch bound),
> > * maximizes **pairwise angular separation** within the fixed-dimensional subspace,
> > * and eliminates **task-specific rotations** that otherwise prevent successful merging.
> >
> > ---

---

> > > ### Author Response · Authors · 2025-11-28
> > >
> > > # **R4. Clarifying the misunderstanding regarding the “near-ETF joint multi-task vector” assumption**
> > >
> > > We appreciate the reviewer’s concern and clarify that our method does **not** assume that the ideal jointly trained multi-task model exhibits a near-ETF structure. Instead, our perspective is based on **fine-tuning as transfer learning**, supported by recent theoretical and empirical results.
> > >
> > > A substantial body of work shows that large pretrained models already exhibit **strong NC behavior and near-ETF geometry** on both source and *downstream* data [9–10]. Galanti et al. [9], in particular, show that Class-Mean ETF geometry **generalizes to unseen samples and even unseen classes**. This indicates that pretrained models supply a **highly symmetric and transferable representation manifold**.
> > >
> > > Fine-tuning on task $t$ does not construct a new geometry from scratch. Instead, it induces a **task-specific geometric shift** on top of this pretrained manifold. The task vector
> > >
> > > $$
> > > \tau_t = \theta_t - \theta_0
> > > $$
> > >
> > > captures precisely this **transition geometry** from the pretrained ETF-like structure to the task-specific collapsed structure.
> > >
> > > Prior merging works [3–6] consistently show that such task vectors lie in a **shared low-rank subspace**, but under arbitrary rotations. ETF alignment is thus a **canonical operator** for reconciling these rotations; it is *not* an assumption about the unknown joint optimum.
> > >
> > > ---
> > >
> > > We sincerely thank the reviewer for the time and effort invested during the review process.
> > > However, we respectfully believe that dismissing the contribution based solely on the concerns, regarding the ETF motivation, the interpretation of the near-ETF assumption, and the role of SVD in constructing the shared subspace, does not fully reflect the broader significance and potential impact of the work. If you would like to discuss any of these points further, please feel free to let us know—we are happy to engage constructively.
> > >
> > >
> > > [1] Parker, Liam, et al. Neural collapse in the intermediate hidden layers of classification neural networks, 2023.
> > >
> > > [2] Beaglehole, Daniel, Ioannis Mitliagkas, and Atish Agarwala. Feature learning as alignment: a structural property of gradient descent in non-linear neural networks, 2024.
> > >
> > > [3] Marczak, Daniel, et al. No task left behind: Isotropic model merging with common and task-specific subspaces, 2025
> > >
> > > [4] Gargiulo, Antonio Andrea, et al. Task singular vectors: Reducing task interference in model merging, 2025
> > >
> > > [5] Wei, Yongxian, et al. Modeling multi-task model merging as adaptive projective gradient descent, 2025
> > >
> > > [6] Yang, Enneng, et al. Adamerging: Adaptive model merging for multi-task learning, 2023
> > >
> > > [7] Singh, Sidak Pal, et al. Model fusion via optimal transport, 2020.
> > >
> > > [8] Chen, Zhikang, et al. Learning without Isolation: Pathway Protection for Continual Learning, 2025
> > >
> > > [9] Galanti, Tomer, et al. On the role of neural collapse in transfer learning, 2021
> > >
> > > [10] Xiao, Li, et al. Principled and efficient transfer learning of deep models via neural collapse, 2024.

---

### Official Review · Reviewer_JhZL · 2025-10-30

**Soundness:** 2
**Presentation:** 2
**Contribution:** 2
**Rating:** 2
**Confidence:** 3

**Summary:**

Existing model methods often ignore directional information in parameter and feature spaces. The authors propose Merging with Directional Alignment (MDA), a method that aims to aligns directions. They motivate their work with insights from neural collapse, showing that traditional model merging disrupts the simplex equiangular frame (ETF). Results on standard benchmarks show promising results.

**Strengths:**

- method obtains good results on standard vision model mering benchmark dataset, including good generalization.
- gain of method increases for more difficult 20 tasks scenario and also for larger models (ViT L/14)

**Weaknesses:**

- I found the presentation lacking. The paper is very hard the follow and I found often unclear and incorrect..
(E.g. The introduction of Neural collapse  (section 3.2) needs motivation. In the start of section 4, the main motivation claims that 'Other methods (Gargiulo, wei, Marczak) are based on task-specific subspaces which are claimed to be computationally expensive. They do not only focus on task-specific subspaces, and the paper lacks a computational comparison, etc).

- The results of ISO-CLS are missing and those of ISO-TA are lower than in their paper (and those are comparable than the ones proposed in this paper). Could you explain that ?

- the first algorithm, being SVD based has great similarity with Gargiulo et al. The authors should provide a more detailed explanation of the main differences (and similarities).


minor:
- in related work, authors should clearly state to which strategy they belong data-free/Data-based.
- typo: Lalign and Lrotation

**Questions:**

The others should address the weaknesses.

---

> ### Author Response · Authors · 2025-11-19
>
> **Response to W1:** We appreciate the reviewer’s constructive feedback. We agree that the motivation in Section 3.2 and the discussion at the beginning of Section 4 can be further clarified. We have refined the presentation in these parts to make the reasoning and connections more explicit.
>
> **(1) Motivation for Neural Collapse (Section 3.2).**
> We have revised Section 3.2 to explicitly explain **why** Neural Collapse (NC) is introduced as the geometric foundation of our framework.
> Recent studies provide two complementary perspectives:
> - [1] demonstrates that ETF-like organization is not confined to the final classifier layer but progressively emerges throughout intermediate layers, implying that each layer’s feature representations locally approach a simplex ETF configuration.
> - [2] further reveals through the Neural Feature Ansatz (NFA) that training process implicitly aligns the dominant singular directions of the weight matrices with the evolving feature covariance, indicating that weights and features co-evolve toward geometrically consistent, maximally separated directions.
>
> Together, these findings suggest that both feature and weight spaces inherently tend to organize along near-ETF structures, promoting maximal inter-class separation.
> Motivated by this, we adopt the ETF geometry as the target structure for aligning task directions during model merging, providing both a principled geometric prior and a theoretical rationale for the directional alignment mechanism used in our framework.
>
>
> **(2) Clarifying the statement about prior work.**
>
> We have revised the corresponding paragraph to improve clarity and accuracy, as follows:
> >Most prior works either partition the parameter space into different subspaces [3,4,5],  or formulate model merging as an optimization problem over task-specific fusion coefficients [6]. In contrast, our method provides a unified geometric perspective that views model merging through the lens of directional alignment.
> In the parameter space, we align the shared subspace with the simplex ETF directions to ensure globally consistent task representations.
> In the feature space, we leverage the ETF structure naturally formed between the final-layer features and classifier weights, and jointly optimize the fusion coefficients and rotation matrices to achieve feature-level directional consistency.
> By aligning task updates with the simplex ETF geometry, we bridge parameter-space and feature-space alignment within a single framework, and further provide a theoretical analysis.
>
>
> **(3) Computational comparison.**
> To address the reviewer’s concern, we have included some tables in the supplementary material reporting both data-free and data-based methods.
>
> - Data-free
>
> 1. Notations
>
> | Symbol | Meaning |
> |:--|:--|
> | $T$ | Number of tasks (models to merge) |
> | $L$ | Number of network layers in each model |
> | $n$ | Matrix dimension of each layer weight ($n\times n$ for simplicity) |
> | $m$ | Smaller dimension in a non-square layer ($m=min(p,q)$, $n=max(p,q)$ if the matrix is $p\times q$) |
> | $\tau_t^{(\ell)}$ | Parameter matrix of task $t$ at layer $\ell$ |
> | $U_t^{(\ell)},\Sigma_t^{(\ell)},V_t^{(\ell)}$ | Matrices obtained by the singular value decomposition (SVD) of $\tau_t^{(\ell)}$ |
> | $U_{\text{cat}}^{(\ell)},V_{\text{cat}}^{(\ell)}$ | Concatenated matrices of all tasks used to build the shared subspace |
> | $W_{\text{ETF}}$ | Simplex Equiangular Tight Frame matrix for ETF projection |
> | $C$ | Number of classes used to construct the ETF basis |
> | $\mathcal{O}(\cdot)$ | Big-O notation expressing asymptotic computational complexity |
>
> 2. Complexity of Algorithm 1 (Ours)
> Algorithm 1 performs the following computations for each layer $\ell=1,\dots,L$:
> **Per-task SVDs.** For each task $t=1,\dots,T$, compute  $\tau_t^{(\ell)} = U_t^{(\ell)}\Sigma_t^{(\ell)}{V_t^{(\ell)}}^{\!\top},$ with complexity $\mathcal{O}(n^3)$ per SVD.  Total cost across all tasks and layers: $\mathcal{O}(T L n^3)$.
> **Concatenated SVDs.** After concatenating the top-$k$ components of all tasks into $U_{\text{cat}}^{(\ell)}$ and $V_{\text{cat}}^{(\ell)}$, two additional SVDs are performed on these matrices. Each SVD is $\mathcal{O}(n^3)$, giving an extra $\mathcal{O}(2 L n^3)$.
> **ETF projection** – Compute $\tau_{\text{etf}}^{(\ell)}=\tau_{\text{share}}^{(\ell)}W_{\text{ETF}}^{\\top}W_{\text{ETF}},$  which requires two matrix multiplications of size $n\times C$ and $C\times n$, costing $\mathcal{O}(n^2 C)$ per layer.

---

> > ### Author Response · Authors · 2025-11-19
> >
> > 3. Complexities of Baseline Methods (from [3])
> >
> > | Method | Main SVD operations per layer | Total Complexity |
> > |:--|:--|:--|
> > | **Iso-C [1]** | One SVD on $\Delta_{\text{TA}}$ (per layer) | $\mathcal{O}(L n^3)$ |
> > | **Iso-CTS [1]** | One SVD on $\Delta_{\text{TA}}$ + $T$ SVDs on each $\Delta_t$ + two SVDs for orthogonalization ($U^\*,V^*$) | $\mathcal{O}((T+3)L n^3)$ |
> > | **TSV-M [2]** | $T$ SVDs on task matrices + two SVDs for reconstruction | $\mathcal{O}((T+2)L n^3)$ |
> > | **Ours (Alg. 1)** | $T$ per-task SVDs + two concatenated SVDs + ETF projection | $\mathcal{O}((T+2)L n^3+n^2LC)\$ |
> >
> > - Data-based
> > Optimizing rotation matrices and fusion coefficients adds a *small* wall-clock overhead per epoch (≈1–3%), while the **Procrustes step is the dominant additional FLOPs cost**.
> >
> > **Per-epoch timing** (averaged across architectures):
> >
> > | Model | model\_fwd (s) | backward (s) | procrustes (s) | epoch total (s, partial) | Procrustes time ratio |
> > |:------|---------------:|--------------:|----------------:|--------------------------:|----------------------:|
> > | **ViT-L-14** | 10.82 | 3.33 | 0.24 | ≈14.47 | **1.7%** |
> > | **ViT-B-32** | 4.18 | 0.87 | 0.16 | ≈5.27 | **3.0%** |
> > | **ViT-B-16** | 6.12 | 1.12 | 0.16 | ≈7.48 | **2.2%** |
> >
> > **Per-epoch FLOPS**:
> >
> > | Model | rot\_forward | rot\_total | procrustes |
> > |:------|--------------:|-------------:|-------------|
> > | **ViT-L-14** | 3.78×10⁷ | 1.13×10⁸ | 9.06×10⁸ |
> > | **ViT-B-32** | 1.68×10⁷ | 5.03×10⁷ | 2.68×10⁸ |
> > | **ViT-B-16** | 1.68×10⁷ | 5.03×10⁷ | 2.68×10⁸ |
> >
> > The alignment step (rotation application + Procrustes computation) therefore introduces **negligible wall-clock cost** and a **manageable FLOPs increase** dominated by the Procrustes step. It is important to note that the reported FLOPs counters (`rot_forward`, `rot_total`, and `procrustes`) measure **computational cost**, not loss components:
> >
> > | Counter | Meaning | Training stage |
> > |:---------|:--------|:---------------|
> > | `rot_forward` | FLOPs of applying rotation matrices ($\mathbf{h}' = \mathbf{R}\mathbf{h}$) during forward pass | Forward |
> > | `rot_total` | Approximate total FLOPs of rotation (forward + backward) | Forward + Backward |
> > | `procrustes` | FLOPs of solving for optimal rotation via SVD | Alignment update
> >
> > [1] Parker, Liam, et al. Neural collapse in the intermediate hidden layers of classification neural networks, 2023.
> >
> > [2] Beaglehole, Daniel, Ioannis Mitliagkas, and Atish Agarwala. Feature learning as alignment: a structural property of gradient descent in non-linear neural networks, 2024.
> >
> > [3] Marczak, Daniel, et al. No task left behind: Isotropic model merging with common and task-specific subspaces, 2025
> >
> > [4] Gargiulo, Antonio Andrea, et al. Task singular vectors: Reducing task interference in model merging, 2025
> >
> > [5] Wei, Yongxian, et al. Modeling multi-task model merging as adaptive projective gradient descent, 2025
> >
> > [6] Yang, Enneng, et al. Adamerging: Adaptive model merging for multi-task learning, 2023

---

> > > ### Author Response · Authors · 2025-11-19
> > >
> > > **Response to W2:** Thanks for pointing out the issue. We would like to explain it as follows.
> > >
> > > As stated in our anonymous GitHub repository, our implementation framework is based on **[1]**, and the initial **8-task experiments** were conducted using the **ViT-B/32** and **ViT-L/14** backbones, with checkpoints provided in that work (available at:
> > > https://drive.google.com/drive/folders/1u_Tva6x0p6oxu5Eo0ZZsf-520Cc_3MKw).
> > > The results we obtained under this setup are consistent with those reported in the **[2]**, specifically aligning with the values presented in **Table 4** of its Appendix, confirming the correctness of our reproduction.
> > >
> > >
> > > However, during the subsequent large-scale validation (14-task and 20-task experiments), we found that both **[2]** and **[3]** methods were **implemented using a different set of checkpoints**, which **we used** in those later experiments (available at:  https://drive.google.com/drive/folders/1UEM1Thcz1c7dc1nji1i5uTN53Kf6G3-e).
> > >
> > > This mismatch in checkpoint sources explains the performance discrepancy observed in our originally reported **8-task ISO results**. When we re-ran the [2] experiments using the corresponding official checkpoints provided by the [2] authors, the results became consistent with those reported in their paper.  We have updated these corrected results in our revised version to keep consistency.
> > >
> > > For completeness, we note that we did not include the ISO-CLS variant in our main comparison. Our goal was to analyze the effectiveness of **global directional alignment** across the entire parameter space, whereas ISO-CLS already introduces elements of layer-wise or hierarchical alignment within its merging strategy.
> > > The following table presents the corresponding results, showing that although ISO-CLS TA outperforms MDA TA in some cases, MDA TA still achieves overall better performance across most tasks.
> > >
> > > | **Method** | **ViT-B-32** 8 tasks | 14 tasks | 20 tasks | **ViT-B-16** 8 tasks | 14 tasks | 20 tasks | **ViT-L-14** 8 tasks | 14 tasks | 20 tasks |
> > > |-|-|-|-|-|:----------:|:----------:|:------------------:|:----------:|:----------:|
> > > | ISO TA | 82.9 | 78.9 | 73.1 | 89.1 | 84.1 | 81.5 | 91.1 | 88.3 | 87.1 |
> > > | ISO-CLS TA | 80.8 | 79.7 | 76.6 | 88.5 | **85.8** | 82.8 | 90.7 | 87.0 | **88.9** |
> > > | **MDA TA** | **86.4** | **81.4** | **77.3** | **89.9** | **85.8** | **84.7** | **92.0** | **89.4** | 88.4|
> > >
> > >
> > > The following table reports the results of the generalization experiments.
> > >
> > > | **Method** | **Seen Tasks** |  |  |  |  |  | |**Unseen Tasks** |  |  |
> > > |-|-|-|-|-|-|-|-|-|-|-|
> > > |  | **SUN397** | **Cars** | **RESISC45** | **DTD** | **SVHN** | **GTSRB** | **Avg.** | **MNIST** | **EuroSAT** | **Avg.** |
> > > | ISO TA | 72.4 | **74.2** | 89.8 | 87.1 | 83.7 | 90.8 | 83.0 | 79.9 | 51.2 | 64.1 |
> > > | ISO-CLS TA | **72.6** | 74.1 | 88.5 | 85.3 | 80.0 | 87.5 | 81.3 | 78.2 | **52.3** | 65.3 |
> > > | DOGE TA | 69.8 | 72.6 | 86.6 | 67.6 | 80.8 | 91.6 | 79.8 | 81.3 | 48.2 | 64.8 |
> > > | **MDA TA** | 71.1 | 72.9 | **92.4** | **91.5** | **95.8** | **97.0** | **86.8** | **85.5** | 47.7 | **66.6** |
> > >
> > > The following table reports the results on ViT-B/32 over eight tasks, using the same checkpoints as in [2,3]. The results show that MDA-TA achieves better performance compared to the baselines.
> > >
> > > | **Method** | **SUN397** | **Cars** | **RESISC45** | **EuroSAT** | **SVHN** | **GTSRB** | **MNIST** | **DTD** | **Avg.** |
> > > |:------------|:----------:|:----------:|:----------:|:----------:|:----------:|:----------:|:----------:|:----------:|:----------:|
> > > | TSV TA | 67.2 | 70.3 | 85.7 | 94.5 | 91.6 | 92.2 | **99.3** | 84.8 | 85.7 |
> > > | ISO TA | 71.8 | 73.8 | 87.7 | 90.9 | 82.3 | 88.1 | 98.2 | 86.0 | 84.9 |
> > > | ISO-CLS TA | **72.4** | **74.0** | **87.9** | 90.6 | 76.9 | 85.3 | 97.5 | 85.5 | 83.8 |
> > > | **MDA TA** | 70.3 | 72.0 | **87.9** | **96.0** | **92.5** | **93.3** | **99.3** | **87.5** | **87.4** |
> > >
> > > The following table reports the results on ViT-L/14 over eight tasks, using the same checkpoints as in [2,3]. The results show that MDA-TA achieves better performance compared to the baselines.
> > > | **Method** | **SUN397** | **Cars** | **RESISC45** | **EuroSAT** | **SVHN** | **GTSRB** | **MNIST** | **DTD** | **Avg.** |
> > > |:------------|:----------:|:----------:|:----------:|:----------:|:----------:|:----------:|:----------:|:----------:|:----------:|
> > > | TSV TA | 77.8 | 89.8 | 93.5 | 98.7 | 94.7 | 96.1 | 99.5 | 93.2 | 92.9 |
> > > | ISO TA | **79.9** | 91.4 | **94.8** | 99.0 | 90.5 | 95.5 | 99.3 | **96.3** | 93.3 |
> > > | ISO-CLS TA | 79.6 | **91.7** | 94.6 | 98.8 | 88.8 | 95.4 | 99.2 | 95.6 | 92.9 |
> > > | **MDA TA** | 78.8 | 90.7 | 94.3 | **99.2** | **95.2** | **97.0** | **99.6** | 94.5 | **93.6** |
> > >
> > >
> > >
> > > [1] Yang, Enneng, et al. Adamerging: Adaptive model merging for multi-task learning, 2023
> > >
> > > [2] Marczak, Daniel, et al. No task left behind: Isotropic model merging with common and task-specific subspaces, 2025
> > >
> > > [3] Gargiulo, Antonio Andrea, et al. Task singular vectors: Reducing task interference in model merging, 2025

---

> > > > ### Author Response · Authors · 2025-11-19
> > > >
> > > > **Response to W3:** Thanks for the comments. We would like to explain them as follows:
> > > >
> > > > Specifically, while both our method and [1] employ singular value decomposition (SVD) to capture low-rank structures in task parameters, and then obtain the shared parameter space, the fundamental difference lies in our **unified geometric perspective** on the role of **direction**. [1] primarily focuses on reducing interference between different tasks, operating in a **task-centric** manner that treats each task independently. In contrast, our method is **geometry-centric**, it aims to achieve **global directional coherence** across all tasks by aligning them within a shared geometric framework grounded in the simplex Equiangular Tight Frame (ETF).
> > > > - Concretely, in **parameter space** (Algorithm 1), we start from SVD-based decomposition as in [1], but rather than treating each task separately, We project the shared parameter space onto an ETF basis, enforcing globally consistent directional geometry across tasks. This step explicitly encodes the idea that *directional alignment*, not just subspace separation.
> > > > - Beyond parameter-space alignment, we extend the same geometric intuition to the **feature space** (Algorithm 2).  Here, we reconstruct an optimization process that jointly learns fusion coefficients and a task-specific **feature rotation matrix**, guiding the final layer’s features toward the ETF configuration predicted by **neural collapse**.
> > > >
> > > > In summary, while [1] focuses on local subspace decorrelation through SVD, MDA provides a **unified directional alignment framework** that operates simultaneously in parameter space and feature space.
> > > > [1] Gargiulo, Antonio Andrea, et al. Task singular vectors: Reducing task interference in model merging, 2025
> > > >
> > > > **Response to Q1:** Thanks for the valuable question. We would like to explain it as follows:
> > > > We agree that explicitly distinguishing **data-free** and **data-based** strategies in the related work section would improve clarity and accessibility.
> > > > We will revise the section to clearly categorize each method according to its optimization setting (e.g., **TSV**, **ISO**, **DOGE-TA** as data-free; **AdaMerging**, **Representation Surgery**, **DOGE-AM** as data-based).
> > > >
> > > > At the same time, our work aims not to propose two unrelated algorithms (data-free and data-based), but to provide a **unified geometric perspective** that bridges both paradigms through the lens of *directional alignment*. Both variants of our method share the same insight:  effective model merging depends not only on the magnitude of task updates but also on their **directional consistency** in both **parameter space** and **feature space**.
> > > > This notion of **directional alignment** serves as a common geometric foundation connecting the data-free and data-based regimes.
> > > >
> > > > - **Data-free regime (parameter-space alignment):**
> > > >   We align task vectors through **low-rank SVD decomposition** followed by **ETF-based directional projection**, ensuring coherent geometry *without accessing any data*.
> > > > - **Data-based regime (feature-space alignment):**
> > > >   When task data are available, we build upon the aligned parameter space $\tau_{\text{etf}} = \{\tau^\text{etf}_t\}_{t=1}^T$ obtained from Algorithm 1 and further optimize the fusion coefficients and rotation matrices to maintain directional consistency among feature representations. The optimization follows the objective defined in Eq.8.
> > > >
> > > > Thus, the data-free and data-based variants are not two separate algorithms but **complementary realizations of the same geometric principle**.  Our framework simultaneously covers both regimes.  From this unified geometric perspective, our method demonstrates that **directional alignment**—rather than the presence or absence of data—is the key factor governing successful model merging.  We have updated the **Related Work** section accordingly to explicitly state which prior methods belong to each regime and to highlight how our approach unifies them under a shared geometric perspective.

---

### Official Review · Reviewer_mkA1 · 2025-10-31

**Soundness:** 3
**Presentation:** 3
**Contribution:** 4
**Rating:** 8
**Confidence:** 3

**Summary:**

This paper introduces a framework called Merging with Directional Alignment (MDA) for model merging. The key claim is that prior approaches—focused mainly on adjusting coefficients or subspaces—ignore the role of directional information in both parameter and feature spaces. Drawing inspiration from the neural collapse phenomenon, the authors argue that the optimal geometry of a well-trained network approximates a simplex equiangular tight frame (ETF), and that preserving this directional structure during merging can mitigate interference and enhance generalization. MDA thus proposes two main components:

- Parameter-space directional alignment: Low-rank decomposition of task vectors followed by alignment with a constructed ETF basis.
- Feature-space directional alignment: Optimization of task-specific rotation matrices (in SO(d)) and fusion coefficients using a composite loss balancing entropy, ETF alignment, and rotation regularization.

The paper claims theoretical justification via Rademacher complexity bounds and demonstrates empirical gains across vision (ViT-B/32, ViT-L/14) and NLP (Flan-T5-base) tasks.

**Strengths:**

- The idea of explicitly enforcing directional alignment in model merging is original in its formulation and integration of neural collapse geometry into the merging framework.
- The empirical evaluation is extensive, spanning 8–20 task vision benchmarks and GLUE-style NLP datasets. The improvements, though moderate (typically +0.5–2%), are consistent and show that directional alignment has a measurable effect. The paper provides clear ablations for the loss weights and module contributions, showing awareness of the internal trade-offs.
- The paper is well-organized and accessible to readers familiar with model merging literature.
- If validated more rigorously, MDA could meaningfully influence the design of geometry-aware merging algorithms, offering a conceptual bridge between theoretical properties of trained models (neural collapse) and practical post-training fusion methods.

**Weaknesses:**

- Despite focusing on geometry, the paper lacks quantitative or visual analysis of directional statistics: e.g., angular distributions between task vectors before/after alignment, cosine similarity matrices, or CKA correlation trends.
- The distinction between MDA and prior geometric methods (TSV, ISO, DOGE) remains unclear. Many of these already manipulate task subspaces or singular vector orientations—essentially performing partial directional alignment.
- The feature-space optimization involves learning rotation matrices in SO(d), which can be expensive for high-dimensional layers.
- Neural collapse typically emerges in overparameterized, fully trained networks under strong assumptions (balanced data, vanishing training error). Fine-tuned models and merged networks rarely meet these conditions. The connection to ETF geometry, while intuitively appealing, is speculative.

**Questions:**

- How is $\Delta ETF$ measured in practice? Could the authors report angular deviations or ETF cosine scores before and after alignment to confirm the hypothesized correlation between geometry and accuracy?
- What is the computational overhead (in FLOPs or training time) introduced by optimizing the rotation matrices and fusion coefficients? Would a simplified or approximate alignment (e.g., diagonal or low-rank rotations) retain most of the benefit?
- Does MDA remain stable when merging models fine-tuned on dissimilar modalities (e.g., text vs. image encoders), or does it assume shared architectures and token embeddings?
- The low-rank truncation in Algorithm 1 is crucial; how does varying the rank k affect both computational efficiency and accuracy?
- Since many modern fine-tuning schemes are parameter-efficient, can the ETF alignment principle apply to merging low-rank adapters instead of full models?

---

> ### Author Response · Authors · 2025-11-19
>
> **Response to W1&Q1:** Thanks for the useful comments. We will incorporate references to some neuroscience-inspired works in our study and conduct comparative analyses accordingly.
>
> To investigate the relationship between geometric alignment and model performance, we analyzed the correlation between the cosine-based ETF deviation ($\Delta_{\text{ETF}}$) and the corresponding performance difference ($\Delta_{\text{diff}}$) across different architectures and task configurations, **$\Delta_{\text{Ours}}$ denotes the cosine distance between the direction-aligned representation and the standard ETF, while $\Delta_{\text{TSV}}$ denotes the cosine distance between the pre-alignment representation and the standard ETF.**  As shown in the Figure1 (Right) and summarized in the table below, **larger $\Delta_{\text{ETF}}$, indicating weaker directional alignment of original parameter space (before alignment), consistently leads to higher $\Delta_{\text{diff}}$,** This clear trend demonstrates that **directional coherence, quantified by cosine similarity relative to the ETF geometry, is strongly predictive of the merged model’s performance**.
>
> | Architecture | 8 tasks | 14 tasks | 20 tasks |
> | --- | --- | --- | --- |
> | **$\Delta_{\text{ETF}}$ (cosine similarity gap)** |  |  |  |
> | ViT-B-32 | 0.17 | 0.20 | 0.14 |
> | ViT-B-16 | 0.09 | 0.12 | 0.15 |
> | ViT-L-14 | 0.07 | 0.11 | 0.14 |
> | **$\Delta_{\text{diff}}$ (performance gap)** |  |  |  |
> | ViT-B-32 | 1.2% | 1.6% | 0.9% |
> | ViT-B-16 | 0.9% | 1.3% | 2.2% |
> | ViT-L-14 | 0.7% | 1.0% | 1.3% |
>
> Overall, these results confirm a strong empirical coupling between geometric deviation and performance loss: **the smaller the deviation from ETF (higher directional consistency), the better the merged model performs.**
>
> **Response to W2:** Thanks for your questions. We would like to explain this question as follows:
>
> While previous works [1，2，3] primarily focus on reducing task interference through parameter subspace, we explicitly emphasize the **importance of direction**, formulating model merging as the problem of achieving **global directional coherence** across tasks.
> This coherence is established through projection onto a **simplex Equiangular Tight Frame (ETF)** basis, which provides the theoretically optimal configuration for balanced and maximally separated directions.
>
> 1. **Parameter-space alignment (Algorithm 1)**
> In the parameter space, both [1,2,3] and MDA employ SVD to extract low-rank task representations. However, while these methods **decompose the parameter space into *shared* and *task-specific* subspaces and optimize them separately to reduce task interference,** MDA instead achieves this goal through **directional alignment**. **Rather than dividing the space, we align all task directions within a common geometric basis—the simplex ETF, which enforces global directional coherence and captures shared information implicitly.** This formulation avoids the need for explicit subspace partitioning and provides a more unified and theoretically grounded approach to parameter-space consistency.
> 2. **Feature-space alignment (Algorithm 2)**
> Beyond parameter-space alignment, MDA further extends the same geometric intuition into the **feature space**, where it learns task-specific rotation matrices and fusion coefficients that align feature representations of each task with the ETF geometry predicted by neural collapse. In contrast, prior methods [1,2,3] restrict their geometric operations to the parameter space. This stage ensures that the merged model maintains consistent feature directions across tasks, bridging the theoretical alignment in the parameter space with the emergent geometry observed in the feature space.
>
> [1] Yang, Enneng, et al. Adamerging: Adaptive model merging for multi-task learning, 2023
>
> [2] Marczak, Daniel, et al. No task left behind: Isotropic model merging with common and task-specific subspaces, 2025
>
> [3] Gargiulo, Antonio Andrea, et al. Task singular vectors: Reducing task interference in model merging, 2025

---

> > ### Author Response · Authors · 2025-11-19
> >
> > **Response to W3 & Q2:** Thanks for the very constructive comments. We would like to clarify it as follows:
> > Optimizing rotation matrices and fusion coefficients adds a *small* wall-clock overhead per epoch (≈1–3%), while the **Procrustes step is the dominant additional FLOPs cost**.
> >
> > **Per-epoch timing** (averaged across architectures):
> >
> > | Model | model\_fwd (s) | backward (s) | procrustes (s) | epoch total (s, partial) | Procrustes time ratio |
> > |:------|---------------:|--------------:|----------------:|--------------------------:|----------------------:|
> > | **ViT-L/14** | 10.82 | 3.33 | 0.24 | ≈14.47 | **1.7%** |
> > | **ViT-B/32** | 4.18 | 0.87 | 0.16 | ≈5.27 | **3.0%** |
> > | **ViT-B/16** | 6.12 | 1.12 | 0.16 | ≈7.48 | **2.2%** |
> >
> > **Per-epoch FLOPS**:
> >
> > | Model | rot\_forward | rot\_total | procrustes |
> > |:------|--------------:|-------------:|-------------|
> > | **ViT-L/14** | 3.78×10⁷ | 1.13×10⁸ | 9.06×10⁸ |
> > | **ViT-B/32** | 1.68×10⁷ | 5.03×10⁷ | 2.68×10⁸ |
> > | **ViT-B/16** | 1.68×10⁷ | 5.03×10⁷ | 2.68×10⁸ |
> >
> > The alignment step (rotation application + Procrustes computation) therefore introduces **negligible wall-clock cost** and a **manageable FLOPs increase** dominated by the Procrustes step. It is important to note that the reported FLOPs counters (`rot_forward`, `rot_total`, and `procrustes`) measure **computational cost**, not loss components:
> >
> > | Counter | Meaning | Training stage |
> > |:---------|:--------|:---------------|
> > | `rot_forward` | FLOPs of applying rotation matrices ($\mathbf{h}' = \mathbf{R}\mathbf{h}$) during forward pass | Forward |
> > | `rot_total` | Approximate total FLOPs of rotation (forward + backward) | Forward + Backward |
> > | `procrustes` | FLOPs of solving for optimal rotation via SVD | Alignment update |
> >
> > We conducted the corresponding evaluation using diagonal based on ViT-B-32.
> > | **Method** | **SUN397** | **Cars** | **RESISC45** | **EuroSAT** | **SVHN** | **GTSRB** | **MNIST** | **DTD** | **Avg.** |
> > |:------------|:----------:|:----------:|:----------:|:----------:|:----------:|:----------:|:----------:|:----------:|:----------:|
> > | TSV AM | 68.1 | **72.1** | 87.7 | 96.4 | 92.3 | 93.3 | **99.4** | 88.9 | 87.3 |
> > | MDA AM Diagonal | **70.1** | 71.6 | **89.4** | **98.1** | **95.4** | **96.8** | 87.8 | **92.6** | 87.7 |
> > | **MDA AM** | 70.0 | 72.0 | 89.1 | 97.6 | 93.7 | 95.1 | **99.4** | 91.8 | **88.6** |
> >
> > We found that using diagonal alignment method can lead to certain performance improvements while reducing complexity. This represents a trade-off process between performance and efficiency.

---

> > > ### Author Response · Authors · 2025-11-19
> > >
> > > **Response to W4:** Thank you for raising this. We would like to clarify it as follows:
> > > We agree that **Neural Collapse (NC)**, in its strict theoretical form, requires strong assumptions—overparameterization, balanced data, and vanishing training error—that fine-tuned or merged models do not generally satisfy.  However, our method does **not assume the full emergence of Neural Collapse**. Instead, we adopt **ETF geometry as a geometric prior** to encourage directional consistency and balanced separation, applied differently in the parameter and feature spaces as detailed below.
> > > 1. In the parameter space: ETF as a geometric prior
> > > In the parameter space, we treat the ETF structure as a **geometric prior** that provides an idealized target for directional alignment. By projecting the shared parameter subspace $\tau_{\text{share}}$ onto the ETF basis, we align task directions toward an equiangular and balanced configuration—achieving a more uniform partition of the parameter space. This step ensures that different task updates interact coherently within a common geometric framework.
> > > Recent empirical evidence shows that the geometric regularities associated with Neural Collapse are not confined to the final classifier layer but emerge progressively throughout the network. As depth increases, intermediate representations exhibit decreasing intra-class variability and increasing inter-class angular separation, **forming an increasingly pronounced simplex ETF–like structure across intermediate hidden layers [1].** Complementary to these empirical findings, recent theoretical analyses reveal that feature learning itself can be interpreted as an alignment process between the weight matrices and the evolving feature covariance. [2] demonstrates that during gradient-based optimization, the left singular structure of the weights becomes aligned with the pre-activation tangent kernel of the corresponding layer—a phenomenon formalized as the Neural Feature Ansatz (NFA). This alignment arises naturally from the dynamics of stochastic gradient descent rather than from explicit regularization, **implying that the geometric organization of weights and features co-evolves throughout training.**
> > > Building on these insights, we posit that enforcing ETF-aligned configurations in the shared low-rank parameter subspace provides an explicit and theoretically grounded means to promote this intrinsic alignment. By applying ETF regularization on the principal task directions extracted via SVD, our approach makes the emergent weight–feature alignment more structured and balanced, thereby encouraging maximally separated and decorrelated representations in the shared parameter space.
> > > 2. In the feature space: rotation-induced ETF alignment
> > > In the feature space, we acknowledge that ETF geometry—closely related to Neural Collapse—typically appears only under strong training conditions.  Therefore, rather than assuming it to emerge naturally, we **explicitly induce ETF-like structure** through learnable rotation matrices in Algorithm 2. Each rotation $\mathbf{R}^t$ gradually aligns the empirical class means with the ETF targets, serving as a practical mechanism to **encourage feature-space directional alignment** without relying on the strict theoretical premises of Neural Collapse. This design bridges the idealized NC geometry and realistic multi-task merging scenarios.
> > >
> > > In summary, MDA leverages the ETF geometry as a flexible geometric principle, not as a strict assumption, allowing consistent directional alignment across both parameter and feature spaces, even when the precise conditions for Neural Collapse are not satisfied.
> > >
> > > [1] Parker, Liam, et al. Neural collapse in the intermediate hidden layers of classification neural networks, 2023.
> > >
> > > [2] Beaglehole, Daniel, Ioannis Mitliagkas, and Atish Agarwala. Feature learning as alignment: a structural property of gradient descent in non-linear neural networks, 2024.

---

> > > > ### Author Response · Authors · 2025-11-19
> > > >
> > > > **Response to Q3:** We thank the reviewer for this valuable comment. We would like to clarify it as follows:
> > > >
> > > > We thank the reviewer for this insightful question. In our current experiments, MDA is evaluated on models that share a common architecture and compatible embedding spaces  (e.g., CLIP-based vision encoders or T5-based language models), which ensures consistent parameterization for alignment. However, recent work [1] has shown that **Neural Collapse–like geometry can also emerge in multimodal models such as CLIP**, where it has been successfully used to **align representations across text and image modalities** within a shared latent space. These findings suggest that the ETF-based directional alignment principle of MDA could naturally extend to multimodal merging scenarios as well.
> > > > Exploring this direction—adapting MDA for merging models fine-tuned on *heterogeneous modalities* (e.g., text vs. image encoders)—is an exciting avenue for future research, and we plan to investigate this in our subsequent work.
> > > >
> > > > [1] Zhu, Didi, et al. Neural collapse anchored prompt tuning for generalizable vision-language models, 2024.
> > > >
> > > > **Response to Q4:** We thank the reviewer for raising this point, which is very helpful for improving our manuscript. We would like to clarify it as follows:
> > > >
> > > > When $k = d_\text{out}/T$, the concatenated dimension $kT$ naturally equals $d_\text{out}$, so the resulting shared representation $\tau_{\text{share}}^{(l)}$ directly matches the layer’s output dimension without requiring any truncation. More generally, when $k$ varies across tasks or $kT \neq d_\text{out}$, the formulation still holds:
> > > >     - If $kT > d_\text{out}$, we truncate $U_\text{share}^{(l)}$ to its top $d_\text{out}$ singular components, preserving the most informative shared directions that capture the highest variance across tasks.
> > > >     - If $kT < d_\text{out}$, the remaining dimensions can be completed either by zero-padding or implicitly through the subsequent ETF projection, which fills the residual subspace.
> > > > However, both truncation and zero-padding are approximate adjustments that either discard useful information or introduce artificial components.
> > > > By setting $k = d_\text{out}/T$, the concatenated dimension $kT$ naturally matches $d_\text{out}$, ensuring a balanced contribution from each task without additional correction.  We also conducted an ablation study on the hyperparameter $k$ to examine its effect on the overall model performance. We found that in some cases there are better results around $k = d_\text{out}/T$, but we followed the setting in [1] and reported $k = d_\text{out}/T$ as our result. **MDA TA 0.01** means that we assign the top $1\%$ proportion of feature dimensions $d_\text{out}$ as the principal components for each task.
> > > >
> > > > | **Method** | **ViT-B-32** 8 tasks | 14 tasks | 20 tasks | **ViT-B-16** 8 tasks | 14 tasks | 20 tasks | **ViT-L-14** 8 tasks | 14 tasks | 20 tasks |
> > > > |:------------|:------------------:|:----------:|:----------:|:------------------:|:----------:|:----------:|:------------------:|:----------:|:----------:|
> > > > | **MDA TA** | 86.4 | 81.4 | **77.3** | 89.9 | 85.8 | **84.7** | 92.0 | **89.4** | **88.4**|
> > > > | MDA TA 0.01|  77.9|  76.3| 72.6 |  82.0| 80.5|  78.2| 87.8  | 86.6 |85.4|
> > > > | MDA TA 0.06|  86.1|  **81.9**| 77.1 |  90.1| **86.1**|  82.5| 91.9|  89.3|  88.2|
> > > > | MDA TA 0.1|  **86.7**|  81.2| 75.9 |  **90.6**| 85.5|  81.1| **92.1**|  89.3|  87.1|
> > > > | MDA TA 0.5|  77.8|  70.6| 66.7 | 83.4 | 76.3|  73.0| 87.9|  82.9|  79.5|
> > > > | MDA TA 0.9|  70.5|  66.8|  64.3|  78.3| 72.9|  70.5| 84.2|  79.8|  76.9|
> > > >
> > > > [1] Gargiulo, Antonio Andrea, et al. Task singular vectors: Reducing task interference in model merging, 2025

---

> > > > > ### Author Response · Authors · 2025-11-19
> > > > >
> > > > > **Response to Q5:** Thanks for the useful comments. We would like to clarify it as follows:
> > > > >
> > > > > We appreciate the reviewer’s question regarding parameter-efficient fine-tuning. The ETF alignment principle is not limited to full-model merging;  it can also be applied to **low-rank adapters** since the core idea focuses on *directional alignment* among task-specific updates, independent of their dimensionality or sparsity.
> > > > >
> > > > > We conducted additional experiments on **Flan-T5-base** where each task was fine-tuned using **LoRA**, and the corresponding low-rank updates were merged under the same MDA framework. The results demonstrate that **ETF alignment remains stable and effective** in this setting, achieving performance improvements comparable to those observed in full-parameter merging.
> > > > >
> > > > > | Method | CoLA | MNLI | MRPC | QNLI | QQP | RTE | SST2 | STSB | Avg. |
> > > > > |:--|:--:|:--:|:--:|:--:|:--:|:--:|:--:|:--:|:--:|
> > > > > | Individual | 69.1 | 82.7 | 85.5 | 90.9 | 84.0 | 84.4 | 92.9 | 87.4 | 84.6 |
> > > > > | Weight Averaging | **69.7** | 59.7 | 78.9 | 90.1 | 83.3 | 80.5 | 91.2 | 72.0 | 78.2 |
> > > > > | Task Arithmetic | 68.8 | 55.2 | 78.7 | 89.4 | 83.3 | 79.1 | 91.5 | 72.4 | 77.4 |
> > > > > | Ties-Merging | 68.3 | 56.3 | 79.2 | 89.3 | 83.7 | 79.6 | 91.6 | 71.2 | 77.5 |
> > > > > | Concrete TA | 69.1 | 58.1 | 78.4 | 89.4 | 83.5 | 79.0 | 91.6 | 73.4 | 78.1 |
> > > > > | TSV TA | 69.3 | 77.1 | 79.4 | 89.7 | 84.0 | 80.8 | 92.5 | 83.1 | 82.0 |
> > > > > | ISO TA | 69.1 | 57.4 | 76.7 | 88.6 | 82.7 | 80.1 | 91.3 | 63.3 | 76.2 |
> > > > > | DoGE TA | 69.1 | 71.9 | **80.9** | **90.3** | 83.7 | **82.5** | **92.5** | 77.1 | 79.9 |
> > > > > | **MDA TA** | 69.5 | **76.9** | 76.9 | 89.6 | **83.9** | 82.4 | **92.5** | **84.7** | **82.1** |

---

### Official Review · Reviewer_BhG3 · 2025-11-01

**Soundness:** 3
**Presentation:** 3
**Contribution:** 3
**Rating:** 4
**Confidence:** 4

**Summary:**

This paper theoretically establishes the importance of directional alignment in model merging, and proposes a unified framework, Merging with Directional Alignment (MDA), which aligns direction in both parameter and feature spaces. Extensive experiments on vision (ViT-B/32, ViT-B/16, ViT-L/14 across 8, 14, 20 tasks) and NLP (Flan-T5-base on GLUE) benchmarks show consistent improvements over strong baselines, with ablations validating each component.

**Strengths:**

- **Well-motivated perspective:** This paper formulates model merging as a problem of geometric alignment (direction) rather than just coefficient optimization (magnitude). Moreover, the paper provides a solid theoretical grounding for why directional alignment should matter by the connection to neural collapse and the simplex ETF.

- **Comprehensive evaluation and theoretical analysis:** The experiment is well-designed and comprehensive. Its evaluation spans multiple architectures, diverse tasks, and many baselines. The consistency of improvements strengthens the initial claims. Moreover, this paper provides a theoretical analysis to demonstrate that directional alignment improves structural coherence and tightens generalization bounds.

**Weaknesses:**

- **Questionable assumption about neural collapse in merged models:** The paper assumes that multi-class joint training induces simplex ETF geometry and **that merged models should approximate this structure**. Why? In fact, merged models fundamentally differ from jointly trained models: they aggregate task-specific adaptations rather than training jointly from scratch. No evidence is provided showing that merged models actually violate ETF structure or that imposing ETF structure better approximates joint training than alternatives. Why is ETF specifically the right target structure for merged models? How about other geometric structures (e.g., orthogonal subspaces, common low-rank structure)?

- **Modest performance gains on LLMs:** The NLP results in Table 7 seem to underperform relative to the claims, as MDA TA (82.1%) only marginally outperforms TSV TA (81.8%) and falls short compared to the gains on vision tasks. This raises questions about whether directional alignment via ETF is as beneficial for LLMs as it is for vision models. Why might ETF alignment work better for vision than for NLP? There is a lack of discussion on this point.

**Questions:**

1. The paper claims the method works in both data-free and data-based regimes, but aren't the two settings quite different in terms of optimization?
2. How sensitive are results to the rank k in SVD-based decomposition? Is k fixed or task/layer-specific?

---

> ### Author Response · Authors · 2025-11-19
>
> **Response to W1:** We sincerely thank the reviewer for the insightful feedback regarding assumption about neural collapse in merged models.
> 1. Why merged models should approximate ETF geometry
> - Our framework does **not** assume that merged models inherently satisfy ETF geometry. The simplex ETF structure was first observed in the **feature space** under Neural Collapse [1], and we extend this geometric prior to the **parameter space** following both theory and empirical evidence. In linear classifiers, class-mean features and classifier weights are **collinear**, so the ETF configuration in feature space is directly mirrored in the parameter matrix.  Recent empirical finding [2] provides strong support for the assumption that **ETF-like geometry emerges progressively across layers** rather than appearing only in the final representation space.
> - Recent empirical evidence [2] shows that the geometric regularities associated with Neural Collapse are not confined to the final classifier layer but emerge progressively throughout the network. As depth increases, intermediate representations exhibit decreasing intra-class variability and increasing inter-class angular separation, **forming an increasingly pronounced simplex ETF–like structure across intermediate hidden layers [2].** Complementary to these empirical findings, recent theoretical analyses reveal that feature learning itself can be interpreted as an alignment process between the weight matrices and the evolving feature covariance. [3] demonstrates that during training process, the left singular structure of the weights becomes aligned with the pre-activation tangent kernel of the corresponding layer—a phenomenon formalized as the Neural Feature Ansatz (NFA). This alignment arises naturally from the dynamics of stochastic gradient descent rather than from explicit regularization, **implying that the geometric organization of weights and features co-evolves throughout training.**
> Building on these insights, we posit that enforcing ETF-aligned configurations in the shared low-rank parameter subspace provides an explicit and theoretically grounded means to promote this intrinsic alignment. By applying ETF regularization on the principal task directions extracted via SVD, our approach makes the emergent weight–feature alignment more structured and balanced, thereby encouraging maximally separated and decorrelated representations in the shared parameter space.
> - To further investigate the relationship between geometric alignment and model performance, we analyzed the correlation between the cosine-based ETF deviation ($\Delta_{\text{ETF}}$) and the corresponding performance difference ($\Delta_{\text{diff}}$) across different architectures and task configurations.  As shown in the Figure1 (Right) and summarized in the table below, larger $\Delta_{\text{ETF}}$, indicating weaker directional alignment of original parameter space (before alignment), consistently leads to higher $\Delta_{\text{diff}}$. This clear trend demonstrates that **directional coherence, quantified by cosine similarity relative to the ETF geometry, is strongly predictive of the merged model’s performance**.
>
> | Architecture | 8 tasks | 14 tasks | 20 tasks |
> | --- | --- | --- | --- |
> | **$\Delta_{\text{ETF}}$ (cosine similarity gap)** |  |  |  |
> | ViT-B-32 | 0.17 | 0.20 | 0.14 |
> | ViT-B-16 | 0.09 | 0.12 | 0.15 |
> | ViT-L-14 | 0.07 | 0.11 | 0.14 |
> | **$\Delta_{\text{diff}}$ (performance gap)** |  |  |  |
> | ViT-B-32 | 1.2% | 1.6% | 0.9% |
> | ViT-B-16 | 0.9% | 1.3% | 2.2% |
> | ViT-L-14 | 0.7% | 1.0% | 1.3% |
>
> Overall, these results confirm a strong empirical coupling between geometric deviation and performance loss: **the smaller the deviation from ETF (higher directional consistency), the better the merged model performs.**
>
> 2. Why ETF rather than orthogonal or low-rank structures
> When the number of classes $C$ exceeds the feature dimension $d_{\text{out}}$, assigning perfectly orthogonal class directions becomes mathematically impossible. Forcing strict orthogonality in this setting distorts useful inter-class relations and wastes representational capacity. The ETF structure provides an **optimal and milder compromise** between separation and dimensional constraints:
> - ETF is the unique unit-sphere configuration with **equal pairwise angles and balanced norms**;
> - When $C > d_{\text{out}}$, orthogonality fails, but ETF maintains constant pairwise inner products $-1/(C-1)$. In this situation, the ETF configuration maximizes the minimum distance between all pairs of class vectors within the given dimensional space. It achieves the largest possible angular separation when true orthogonality is not an option;
> - This property directly matches neural collapse theory, where class means form a centered simplex ETF at convergence.

---

> > ### Author Response · Authors · 2025-11-19
> >
> > 3. Comparison with orthogonal and low-rank alternatives
> > Following the reviewer’s suggestion, we conducted additional experiments using **Low-rank alignment and Orthogonal** as baselines, where the rank was set to $d/4$. Low-rank alignment preserved variance but failed to maintain balanced angular separation, leading to weaker class discrimination.  The results show that this low-rank and orthogonal approaches perform notably worse than the ETF-based alignment,  indicating that enforcing the full ETF geometric structure provides more effective directional consistency across tasks.
> >
> > | **Method** | **ViT-B-32** 8 tasks | 14 tasks | 20 tasks | **ViT-B-16** 8 tasks | 14 tasks | 20 tasks | **ViT-L-14** 8 tasks | 14 tasks | 20 tasks |
> > |-|-|-|-|-|-|-|-|-|-|
> > | **MDA TA** | **86.4** | **81.4** | **77.3** | **89.9** | **85.8** | **84.7** | **92.0** | **89.4** | **88.4**|
> > | MDA TA Low-rank| 79.7 | 74.0 | 68.1 | 85.6 | 79.8 | 75.6 |90.6  |  87.1 | 85.8 |
> > | MDA TA Orthogonal|  84.7|  79.7|  75.0| 88.6| 84.2| 80.5| 91.2| 88.2| 87.0|
> >
> > In conclusion, these points justify why ETF is the appropriate target structure for merged models in our work.
> >
> > [1] Papyan, Vardan, X. Y. Han, and David L. Donoho. Prevalence of neural collapse during the terminal phase of deep learning training, 2020.
> >
> > [2] Parker, Liam, et al. Neural collapse in the intermediate hidden layers of classification neural networks, 2023.
> >
> > [3] Beaglehole, Daniel, Ioannis Mitliagkas, and Atish Agarwala. Feature learning as alignment: a structural property of gradient descent in non-linear neural networks, 2024.
> >
> > **Response to W2:** Thanks for pointing out the issue. We would like to explain it as follows.
> >
> > 1. Difference in class granularity and dimensional ratios
> > In vision experiments, each benchmark suite contains **8 （758 classes）, 14 (1016 classes), or 20 (1306 classes) tasks**, with the corresponding number of categories ranging from approximately **758 to over 1,300 classes in total**.
> > For models such as **ViT-B/32 ($d_{\text{out}}$ is typically equal to 768)**, **ViT-B/16 ($d_{\text{out}}$ is typically equal to 768)**, and **ViT-L/14 ($d_{\text{out}}$ is typically equal to 1024)**, the total number of classes *exceeds* the feature dimensionality. In this regime  $(C > d_{\text{out}})$, enforcing balanced angular separation between classes becomes geometrically nontrivial, and **directional alignment via ETF** plays a critical role in achieving near-uniform separation across class directions.
> > Therefore, ETF alignment provides clear structural and generalization benefits for vision models, where the class-to-dimension ratio is high.
> > 2. Smaller class space in NLP tasks
> > In contrast, the NLP setting (based on **eight GLUE tasks**) involves far fewer labels—mostly binary or three-way classification problems (e.g., SST-2, MRPC, QQP, RTE).
> > Even across all tasks combined, the total number of categories remains below **20**, which is *much smaller* than the output feature dimension of **Flan-T5-base ($d_\text{out}$ is typically equal to 2048)**. In this high-dimensional, low-class regime $(C \ll d_{\text{out}})$, class vectors are already well separated without explicit geometric regularization. Thus, ETF alignment provides **limited additional improvement**, as the geometry is already near-orthogonal.
> >
> > In short, the benefit of ETF alignment increases when the number of classes is large relative to the embedding dimension—precisely the case in multi-task vision settings but not in small-class NLP benchmarks.  We have included this quantitative analysis in Section G.2.4 in the appendix of the revision to clarify why directional alignment yields more pronounced gains for vision models than for LLM-based NLP tasks.

---

> ### Author Response · Authors · 2025-11-19
>
> **Response to Q1:**  Thank you for pointing out the limitations. We indeed omitted the comparisons in our manuscript.
>
> We agree that the **data-free** and **data-based** regimes differ in their optimization procedures—one operates solely on model parameters, while the other leverages data to refine feature representations. However, our goal is not to propose two unrelated algorithms, but to offer a **unified geometric perspective** centered on the *importance of directional alignment* across both settings.
> 1. Unified principle
> Both variants are grounded in the same insight: effective model merging depends not only on the magnitude of task updates but also on their **directional consistency** in both *parameter space* and *feature space*. This directional alignment acts as a common geometric foundation, bridging the two regimes.
> 2. Implementation differences, same geometric goal
> - **Data-free regime (parameter-space alignment):**  We align task vectors via **low-rank SVD decomposition** followed by **ETF-based directional projection**, achieving coherent geometry *without using any data*.
> - **Data-based regime (feature-space alignment):**  When task data are available, we extend the same idea to learn **rotation matrices** and **fusion coefficients** that maintain directional consistency among feature representations.
>
> Although the optimization formulations differ, both pursue the same objective—preserving a coherent global structure aligned with the ETF basis. Thus, the data-free and data-based variants are not separate algorithms but complementary realizations of the same geometric principle.
>
> **Response to Q2:**  Thank you for pointing this out. In our manuscript, $k$ is task-specific, and we set it as $k = d_{\text{out}} / T$, $T$ is the number of tasks.
> - Truncating to $d_{\text{out}}$ is not merely for dimensional matching but serves to retain the
> dominant shared directions across tasks that explain the largest variance in the concatenated subspace. This step ensures that the shared representation remains compact and captures the most informative geometric structure before ETF alignment.
> - When $k = d_\text{out}/T$, the concatenated dimension $kT$ naturally equals $d_\text{out}$, so the resulting shared representation $\tau_{\text{share}}^{(l)}$  directly matches the layer’s output dimension without requiring any truncation. More generally, when $k$ varies across tasks or $kT \neq d_\text{out}$, the formulation still holds:
>     - If $kT > d_\text{out}$, we truncate $U_\text{share}^{(l)}$ to its top $d_\text{out}$ singular components, preserving the most informative shared directions that capture the highest variance across tasks.
>     - If $kT < d_\text{out}$, the remaining dimensions can be completed either by zero-padding or implicitly through the subsequent ETF projection, which fills the residual subspace.
> However, both truncation and zero-padding are approximate adjustments that either discard useful information or introduce artificial components.  By setting $k = d_\text{out}/T$, the concatenated dimension $kT$ naturally matches $d_\text{out}$, ensuring a balanced contribution from each task without additional correction.
>
> We also conducted an ablation study on the hyperparameter $k$ to examine its effect on the overall model performance. We found that in some cases there are better results around $k = d_\text{out}/T$, but we followed the setting in [1] and reported $k = d_\text{out}/T$ as our result. **MDA TA 0.01** means that we assign the top $1\%$ proportion of feature dimensions $d_\text{out}$ as the principal components for each task.
>
> | **Method** | **ViT-B-32** 8 tasks | 14 tasks | 20 tasks | **ViT-B-16** 8 tasks | 14 tasks | 20 tasks | **ViT-L-14** 8 tasks | 14 tasks | 20 tasks |
> |:------------|:------------------:|:----------:|:----------:|:------------------:|:----------:|:----------:|:------------------:|:----------:|:----------:|
> | **MDA TA** | 86.4 | 81.4 | **77.3** | 89.9 | 85.8 | **84.7** | 92.0 | **89.4** | **88.4**|
> | MDA TA 0.01|  77.9|  76.3| 72.6 |  82.0| 80.5|  78.2| 87.8  | 86.6 |85.4|
> | MDA TA 0.06|  86.1|  **81.9**| 77.1 |  90.1| **86.1**|  82.5| 91.9|  89.3|  88.2|
> | MDA TA 0.1|  **86.7**|  81.2| 75.9 |  **90.6**| 85.5|  81.1| **92.1**|  89.3|  87.1|
> | MDA TA 0.5|  77.8|  70.6| 66.7 | 83.4 | 76.3|  73.0| 87.9|  82.9|  79.5|
> | MDA TA 0.9|  70.5|  66.8|  64.3|  78.3| 72.9|  70.5| 84.2|  79.8|  76.9|
>
> [1] Gargiulo, Antonio Andrea, et al. Task singular vectors: Reducing task interference in model merging, 2025

---

### Meta-Review · Area_Chair_kChT · 2026-01-08

**Summary:**

A novel method for model merging is proposed based on intuitions from neural collapse literature. Overall the reviewers praised the evaluation and found the idea original. A significant concern however was the validity of the underlying assumptions of the method and its motivation as well as some concerns regarding the baselines and ablations. These were partially addressed in the rebuttal but further addressing these points is needed before publication.

**Reviewer Concerns:**

- Baselines, reproducibility, ablations - partially addressed in rebuttal
- Concerns related to ETF alignment - partially addressed in rebuttal

**Reviewer Scores:**

6nx2 indicated they would keep the score. Other reviewers had similar concerns and thus might also maintain their scores.

---

### Decision · Program_Chairs · 2026-01-26

Reject